# Coordinated regulation of pH alkalinization by two transcription factors promotes fungal commensalism and pathogenicity

Xinhua Huang[1,8] ✉, Guangsheng Chen[2,8], Lei Wu[1,3,8], Yun Zou[1,3,8], Luyao Zhang[1,3], Shanshan Li[1], Kunlin Li[1,3], Zaijie Jiang[1], Yuping Zhang[1], Xiaoqing Chen[1,3], Winnie Shum [4], Jianbiao Dai[4], Huichang Huang[1], Munika Moses[1,3], Xianwei Wu[1,3], Yuanyuan Wang [1], Tong Jiang[1,3], Zhiyi He[2], Qing Guo[5], Wenwen Xue[6], Hao Li[7] ✉ & Changbin Chen [1] ✉

As a clinically relevant opportunistic human fungal pathogen, *Candida albicans* is able to rapidly sense and adapt to changing microenvironments within the host, a process that is essential for its successful invasion and survival. Although studies have shown that the transcription factor Stp2 is the master regulator of environmental alkalinization, accumulating evidence supports a clear involvement of other participants in this adaptation process. Here, following a large-scale genetic screen, we identify the transcription factor Dal81 as an uncharacterized positive regulator of pH alkalinization in *C. albicans*. Dal81 influences the protein levels of Stp2. A mutant lacking *DAL81* also fails to alkalinize both in vitro and in the phagolysosome, and this defective phenotype is further enhanced by deleting both factors in most cases. Notably, our results demonstrate that Dal81 physically interacts with Stp2 to co-regulate the expression of a broad set of downstream target genes related to metabolism of organic acids, oxoacids, carboxylic acids and amino acids. This coordinated regulation mode is required for the alkalinization process and plays a role in modulating commensalism and pathogenicity of *C. albicans*. Taken together, our findings elucidate the cooperative function of Dal81 with Stp2 in the nucleus to orchestrate the expression of downstream genes required for the survival and propagation of *C. albicans* in the host.

Microorganisms are capable of adapting their cells to almost every biological and physicochemical environment. Any deviation from homeostatic conditions can trigger stress reactions in these microbes. Within the spectrum of environmental factors that influence microbial life, pH stands out as one of the most important variables shaping the dynamics of microcosm ecosystems[1]. Changes in pH can induce diverse stresses on cellular functions, including the availability of essential micronutrients, protein functions, and membrane potentials. Importantly, microbes in the human body must adapt to a variety of pHs in different anatomical sites, ranging from acidic human vagina (pH 4.0) to neutral or slightly alkaline blood and tissue (pH 7.4). In particular, pH along the digestive tract varies considerably, ranging from pH 2.0 to 8.0[2]. To survive and grow, microorganisms evolve the ability to rapidly sense and respond to potentially lethal changes in environmental pH, and multiple important signaling pathways are involved. For example, adaptation to extracellular pH is critical for numerous fungal species, primarily mediated by the conserved PacC/Rim101 pathway[3,4]. Moreover, both bacterial and fungal pathogens have been shown to actively modify, either lowering or raising, the pH of their surrounding environment to support their survival and

enhance virulence. Certain pathogenic fungi, such as *Sclerotinia sclerotiorum* and *Aspergillus spp.*, secrete acids to acidify the environment and damage host tissues[5-7]. In contrast, *Legionella pneumophila* can raise the pH of its phagocytic vacuole, a process that may be critical for the intracellular survival and multiplication of this and other intracellular pathogens[8]. Similarly, the gastric pathogen *Helicobacter pylori* employs a urease system to neutralize acidic conditions: it produces ammonia, which enables the bacterium to colonize the highly acidic surface of the stomach[9]. Some phytopathogenic fungi, including *Alternaria alternat, Colletotrichum gleosporioides,* and *Fusarium oxysporum,* also secrete ammonia to alkalinize the acid environment[5]. Thus, the capacity to modulate environmental pH appears to be a general feature among most microorganisms during their interactions with hosts. However, the precise mechanisms behind pH acidification or alkalinization remain largely unknown in fungi.

*Candida albicans* is a common opportunistic fungal pathogen in humans, capable of causing superficial infections at mucosal surfaces as well as systemic infections that can disseminate to nearly all major organs. Such disseminated infections, known as candidemia, typically arise in individuals encountering specific risk factors, including prolonged hospitalization in intensive care units, gastrointestinal surgery, the presence of central venous catheters, polytrauma, extremes of age, severe immunosuppression, neutropenia, solid tumors, and hematological malignancies[10-12]. Owing to the challenges in eradication and treatment, as well as a substantial risk of morbidity and mortality, this fungus was recently classified in the critical priority fungal pathogen group by the World Health Organization (WHO)[13]. A growing body of studies have demonstrated that *C. albicans* raises extracellular pH, both in vitro and in the macrophage phagosome, when nutrients such as fatty acids, N-acetylglucosamine (GlcNAc), carboxylic acids (*e.g.,* lactate), amino acids, peptides, and proteins are available in distinct microenvironments[14-19]. For example, *C. albicans* can avidly alkalinize environmental pH by releasing amine groups in the form of ammonia, particularly when amino acids or carboxylic acids serve as its sole carbon source[16]. Of significance, pH alkalinization has been identified as a key factor linked to infection-associated morphogenetic changes in *C. albicans*. A rise in environmental pH promotes the neutralization of the macrophage phagosome, triggers the critical yeast-to-hyphal transition, and facilitates the escape from macrophages, all processes that significantly contribute to the fitness and pathogenicity of *C. albicans* within the host[15,16,20]. Further mechanistic investigations have revealed that *C. albicans* primarily senses extracellular amino acids via the SPS (Ssy1, Ptr3, and Ssy5) sensor system, followed by the activation of transcription factor Stp2 in a SPS-dependent manner and induction of amino acid influx[21]. Meanwhile, Stp2 is recognized as the master regulator of environmental alkalinization, playing an essential role in amino acid utilization by predominantly controlling the expression of genes encoding amino acid permeases, Ato transporters, and catabolic enzymes, such as *ACH1, CSH3, SSY1, PTR3, SSY5, DUR1,2, DUR31, ATO1,* and *ATO5*[16,22,23]. Interestingly, accumulating evidence suggests that the regulation of pH alkalinization may involve additional players beyond transcription factor Stp2. Mutant cells lacking *STP2*, when grown on both vaginal simulating fluid (VSF) and artificial saliva (AS), retain the ability to elevate the external environmental pH to levels comparable to those observed in wild-type cells[15]. More importantly, *C. albicans* cells cultured on organic acids do not generate ammonia, and genes required for amino acid-induced pH alkalinization have no impact on growth or pH changes when organic acids or N-acetylglucosamine (GlcNAc) are present[19]. These experimental findings strongly indicate that *C. albicans* regulates environmental alkalinization through multiple distinct mechanisms, and that a range of transcriptional regulators may be involved in this process.

Here, we performed a large-scale genetic screen and identified an uncharacterized regulator of pH alkalinization in *C. albicans*. Our data strongly suggest that the transcription factor Dal81 physically interacts with the known pH-regulator Stp2, and together they co-regulate the expression of a broad set of downstream target genes necessary for the metabolic engineering of organic acids, oxoacids, carboxylic acids, and amino acids. This coordinated regulation mode not only facilitates the alkalinization process but also enhances the commensalism and pathogenicity of *C. albicans*.

## Results

### Dal81 is an uncharacterized regulator of pH alkalinization in *C. albicans*

A standard set of in vitro alkalinization-inducing condition was used to expose the *C. albicans* cells on medium 199 (initial pH 4.0) supplemented with phenol red as the pH indicator and incubated for 24 hours at 37 °C[16]. Using this method (Fig. 1A), we screened the homozygous gene deletion library[24] for mutants defective in pH alkalinization. This screen yielded 23 candidate mutants with reduced alkalinization capacity (Fig. S1). Notably, both the *grr1Δ/Δ* and *dal81Δ/Δ* mutants exhibited a complete loss of alkalinization ability, suggesting that Grr1 and Dal81 are potential key regulators of this process. Among these candidates, Dal81 was chosen for further analysis since it encodes a Zn (II)2Cys6 transcription factor and mutant cells lacking *GRR1* showed significant defects in growth (Fig. S2) and a highly filamentous morphology under non-hypha-inducing conditions[25]. Clearly, Fig. 1B showed that deletion of *DAL81* severely abolished the ability of *C. albicans* to alkalinize extracellular environment, phenocopying the defects observed in *stp2Δ/Δ* mutant. Given that both the *dal81Δ/Δ* mutant strain and its isogenic wild-type control strain (SN250) are auxotrophic for arginine (Arg-), culturing these strains on an amino acid-rich medium (*e.g.,* medium 199)[16] could introduce confounding effects on the alkalinization phenotype. To address this, we generated a new *dal81Δ/Δ* mutant strain (*dal81Δ/Δ* M1) in a prototrophic background (SC5314) using the SAT-flipper method[26]. We verified through colony PCR and RT-qPCR that the open reading frame (ORF) of the *DAL81* gene was successfully deleted in the *dal81Δ/Δ* M1 strain (Fig. 1C) and that *DAL81* expression was undetectable in this mutant (Fig. 1D). Importantly, the alkalinization phenotype of the *dal81Δ/Δ* M1 strain was consistent with that of the previously constructed *dal81Δ/Δ* mutant (Fig. 1E). Given this phenotypic concordance, we proceeded to use the original *dal81Δ/Δ* mutant for subsequent analysis. A time-course assay further confirmed the observed phenotypes, as we found that when grown in alkalinization-inducing medium (YNB + 1% CAA, pH 4.0), cultures of both wild-type and complemented strains rapidly raised the medium pH from an initial pH of 4.0 to about 5.5 within several hours and reached approximately 6.5 within 24 hours, whereas cultures of the *dal81Δ/Δ* mutant significantly delayed the alkalinization process, reaching only pH 4.7 at 24 hours (Fig. 1F). Notably, in comparison to the wild-type strain, both the *dal81Δ/Δ* and *stp2Δ/Δ* mutants exhibited only mild growth defects in YPD medium (Fig. S3A, B) but showed significantly impaired growth in YNB medium supplemented with 1% casamino acids (CAA) (Fig. S3C). This growth impairment in YNB + 1% CAA may be due to the inability of the mutants to efficiently catabolize amino acids, particularly CAA, thereby contributing to the observed alkalinization defect. Moreover, the alkalinization defect of the *dal81Δ/Δ* mutant was also observed on solid GM-BCP or GM-BCG medium (initial pH 4.0; Fig. 1G). Reintroducing an ectopic copy of the wild-type allele back into the *dal81Δ/Δ* mutant rescued the alkalinization defect (Fig. 1F, G), confirming that the phenotype is specifically attributable to the deletion of the *DAL81* gene. Like the *stp2Δ/Δ* mutant, the *dal81Δ/Δ* strain released significantly less ammonia than the wild type (Fig. 1H). This finding suggests that reduced ammonia release may be a primary factor contributing to the failed alkalinization phenotype of the *dal81Δ/Δ* mutant. Indeed, consistent with previous findings in the *stp2Δ/Δ* mutant[23], we observed that the growth of the *dal81Δ/Δ* mutant was completely inhibited by the sulfonylurea herbicide metsulfuron methyl (MM), which targets acetohydroxyacid synthase to

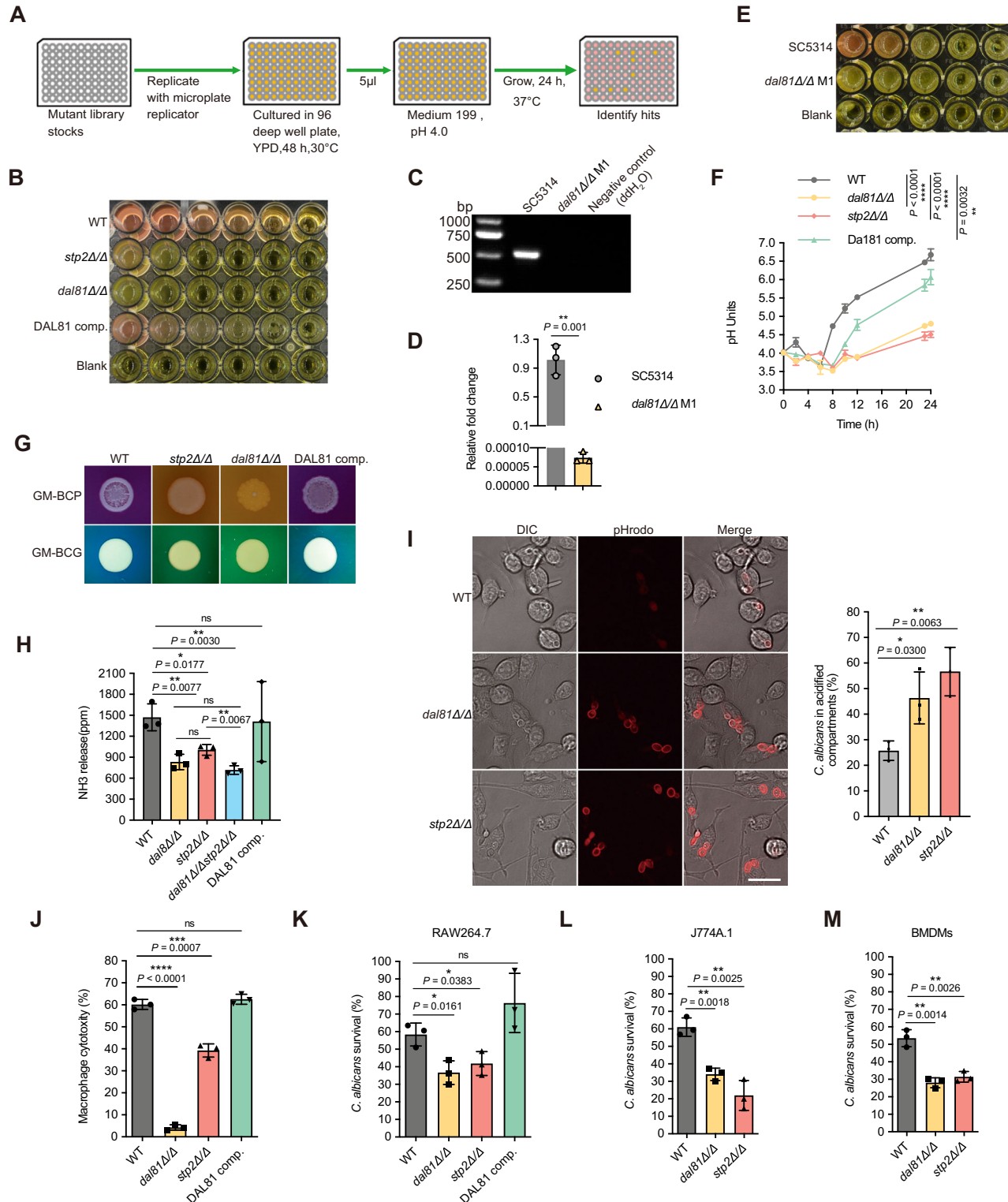

block branched-chain amino acid biosynthesis (Fig. S4). As expected, the complemented strain exhibited growth nearly indistinguishable from the wild type in the presence of MM (Fig. S4). These results indicate that knocking out either *STP2* or *DAL81* yields identical phenotypes, characterized by severe inhibition of amino acid uptake and impaired pH alkalinization. This phenotypic convergence strongly suggests that Stp2 and Dal81 may act in concert or via parallel pathways to regulate this critical process.

The in vitro evidence demonstrating the inability of the *dal81Δ/Δ* mutant to alkalinize environmental pH was further validated in the phagosomal context. Log phase wild-type cells, *stp2Δ/Δ*, and *dal81Δ/Δ* mutant cells were co-cultured with RAW264.7 macrophages cell lines, and the capacity of *C. albicans* cells to modulate phagosomal pH was assessed using the pH-sensitive dye pHrodo, a probe that fluoresces inside acidic phagosomes[27,28]. In this assay, only 24% of wild-type *C. albicans* cells were localized to acidified phagolysosomes, whereas over 50% of both *stp2Δ/Δ* and *dal81Δ/Δ* mutant cells were found in these acidic compartments (Fig. 1I). These findings indicate that deletion of either *STP2* or *DAL81* impairs the ability of fungal cells to neutralize the phagosomal environment. A similar pattern was observed in

**Fig. 1 | C. albicans Dal81 is required for pH alkalinization, hyphal growth, and macrophage escape. A** Schematic outlining the in vitro pH alkalinization screening workflow. The figure was created in BioRender[84]. **B** Similar to *stp2Δ/Δ*, *dal81Δ/Δ* cells displayed alkalinization defects in liquid assays. Wild-type (WT), *dal81Δ/Δ*, *stp2Δ/Δ* and the complemented strain (DAL81 comp.) were grown overnight in YPD, centrifuged, washed with water, and resuspended in medium 199 (initial pH 4.0) at an OD$_{600}$ of 1.0. Serial 1:5 dilutions were prepared in 96-well plates and alkalinization was visualized colorimetrically by the development of an orange-red medium color after 24 h incubation at 37 °C. **C** Colony PCR confirmed *DAL81* deletion. The *DAL81* ORF was present in WT (SC5314) but absent in the *dal81Δ/Δ* M1 mutant. **D** RT-qPCR analysis of *DAL81* expression in SC5314 and *DAL81* knockout strains following 5 h culture in medium 199 (initial pH 4.0) at 37 °C. **E** Under the same conditions as (B), the *dal81Δ/Δ* M1 mutant showed comparable alkalinization defects. **F** *DAL81*-deficient mutants exhibited delayed pH elevation in YNB + 1% CAA (initial pH 4.0), with medium pH measured at indicated time points. **G** Alkalinization defects in *stp2Δ/Δ* and *dal81Δ/Δ* mutants were also observed on solid medium. Cells were spotted onto GM-BCP or GM-BCG solid medium (initial pH 4.0) and incubated at 37 °C for 3 days. **H** Dal81 and Stp2 are required for ammonia release, quantified from GM-BCP agar cultures. **I** *DAL81*-deficient cells failed to alkalinize the macrophage phagosome. Intracellular compartment acidification was measured using pHrodo. Left panel shows a representative image of phagosomal pH tracking. Scale bar, 20 μm. The right panel quantified acidified compartments. **J** *DAL81* deletion reduced RAW264.7 macrophage killing, measured by LDH release after 5 h infection at MOI = 3. **K–M** *DAL81*-deficient cells showed decreased survival in RAW264.7 (**K**), J774A.1 (**L**), and BMDMs (**M**). Data in (**D, F, H–M**) are mean ± SD of three biological replicates. Statistical significance was assessed via one-way ANOVA with Tukey's test (**F**, 24 h) and unpaired two-tailed Student's *t*-test (**D, H–M**). ns, not significant. Source data for (**D, F, H–M**) are provided in the Source Data file.

J774A.1 macrophages (Fig. S5A, B). Time-course experiments further confirmed this trend: phagosomes containing wild-type cells in J774A.1 macrophages exhibited a marked reduction in acidification over time, while those harboring *dal81Δ/Δ* or *stp2Δ/Δ* mutants remained consistently acidic (Fig. S5A, B). Taken together, our data strongly support that transcription factor Dal81 appears to be a new regulator required for pH alkalinization of *C. albicans*.

## Dal81 is required for hyphal growth and escape from macrophages

In *C. albicans*, neutral to alkaline pH environments drive the yeast-to-hyphal morphogenetic switch, a critical adaptive strategy that enables the fungus to evade macrophage-mediated killing[2,16,29]. The transcription factor Stp2 has been identified as a key regulator in this process[15]. Given that Dal81 regulates pH alkalinization, we asked whether Dal81 is also required for hyphal induction and phagosomal evasion. To address this, we investigated the hyphal formation ability of the wild-type, *dal81Δ/Δ*, and the *DAL81*-complemented strain in an alkalinization-inducing medium (YNB + 1% CAA, initial pH 4.5), using the *stp2Δ/Δ* mutant as a control. Similar to the *stp2Δ/Δ* mutant, the *dal81Δ/Δ* mutant cells exhibited a defect in the yeast-to-hyphal transition (Fig. S6A). In contrast, both the wild-type and *DAL81*-complemented strains formed normal hyphae under the same conditions. Consistently, in co-cultures with macrophages, *dal81Δ/Δ* mutant cells were primarily arrested in the yeast form, whereas wild-type cells robustly developed hyphae in the phagosome (Fig. 1I). Importantly, the failure of the *dal81Δ/Δ* mutant to form hyphae in this context is not due to an intrinsic inability to filament, as hyphal growth of the *dal81Δ/Δ* mutant was indistinguishable from that of the wild type when strains were grown overnight in YPD and then transferred to medium 199 buffered at pH either 4.5 or 7.5. (Fig. S6B). These findings indicate that Dal81 plays a specific role in promoting *C. albicans* hyphal formation within macrophages.

To investigate whether Dal81 acts to enhance the ability of *C. albicans* to survive within the phagocyte, and ultimately, contributes to host cell death, we examined the cytotoxicity of fungal cells in RAW264.7 macrophages by measuring the lactate dehydrogenase (LDH) release following a 5-hour co-culture. We found that the *dal81Δ/Δ* mutant cells exhibited significantly lower cytotoxicity compared to both the wild-type and *DAL81*-complemented strains, and even lower than the *stp2Δ/Δ* strain (Fig. 1J). Moreover, we used an endpoint dilution assay to compare the survival of different *C. albicans* strains across three macrophage types, including RAW264.7, J774 A.1, and bone marrow-derived macrophages (BMDMs). The results revealed that both the *dal81Δ/Δ* and *stp2Δ/Δ* strains displayed impaired survival relative to the wild-type when challenged with these macrophages (Fig. 1K–M).

Collectively, these findings affirm the crucial role of Dal81 in modulating environmental alkalinization, hyphal formation, and the escape from innate immune cells such as macrophages.

## Dal81 exerts a superposition effect on Stp2-dependent alkalinization

The data presented above strongly suggested that Dal81 behaves similarly to Stp2 in mediating *C. albicans*-induced alkalinization of the extracellular environment, prompting us to test whether these two transcription factors exhibit functional redundancy in regulating pH alkalinization. We first analyzed the expression patterns of *DAL81* and *STP2* in *C. albicans* during alkalinization in liquid culture (YNB + 1% CAA, initial pH 4.5). Using RT-qPCR, we measured the relative transcription levels of *DAL81* and *STP2* following the rise of culture pH. The results revealed that the expression of both genes was significantly upregulated during the alkalinization process (Fig. 2A), highlighting their shared relevance to extracellular pH elevation.

To fully understand the contributions of Dal81 and Stp2 to environmental alkalinization by *C. albicans*, we generated a *dal81Δ/Δstp2Δ/Δ* double mutant strain. As shown in Fig. S7, the double mutant exhibited similar phenotypes comparable to those of the single mutants, including reduced amino acids uptake capacity (Fig. S7A), impaired hyphal formation under alkalinization-inducing conditions (Fig. S7B), diminished macrophage damage (Fig. S7C), and decreased survival in macrophages (Fig. S7D). However, relative to the single mutants, *C. albicans* cells lacking both *DAL81* and *STP2* displayed more severe alkalinization defects when grown in YNB + 1% CAA (initial pH 4.5; Fig. 2B) and in artificial saliva (AS; Fig. 2C). Notably, this enhanced defect was independent of inoculum size, as it persisted across varying cell densities (Fig. 2E, F). Biochemically, the *dal81Δ/Δstp2Δ/Δ* double mutant showed a more pronounced reduction in ammonia release compared to the *stp2Δ/Δ* single mutant, though its ammonia levels were indistinguishable from those of the *dal81Δ/Δ* mutant (Fig. 1E), suggesting that the severe alkalinization defect in the double mutant is likely driven by impaired ammonia production, and additional Dal81 or Stp2-dependent metabolic endpoints may also contribute to this phenotype. Interestingly, we observed niche-specific differences in their roles. In AS, the *dal81Δ/Δ* mutant behaved like the wild-type, while in vaginal simulating fluid (VSF), the *stp2Δ/Δ* mutant resembled the wild-type, and the *dal81Δ/Δ* mutant phenocopied the double mutant (Fig. 2D). These results reveal a context-dependent cooperative relationship between Dal81 and Stp2 in regulating pH alkalinization, where their functional importance shifts based on the environmental niche.

In addition to their roles in pH regulation, colony morphology varied significantly across strains, reflecting differences in hyphal growth and morphological switching. As shown in Fig. 2F, the wild-type and Dal81-complemented strains formed wrinkled colonies, a hallmark of robust hyphal development, whereas the *stp2Δ/Δ* mutant and *dal81Δ/Δstp2Δ/Δ* double mutant produced smooth colonies, indicative of a yeast-dominant state with no detectable hyphal growth. The *dal81Δ/Δ* mutant displayed an intermediate phenotype, characterized by limited colony wrinkling. Microscopic examination confirmed these morphological differences: wild-type and complemented strains displayed elongated, branching hyphae, while *dal81Δ/Δ* cells showed

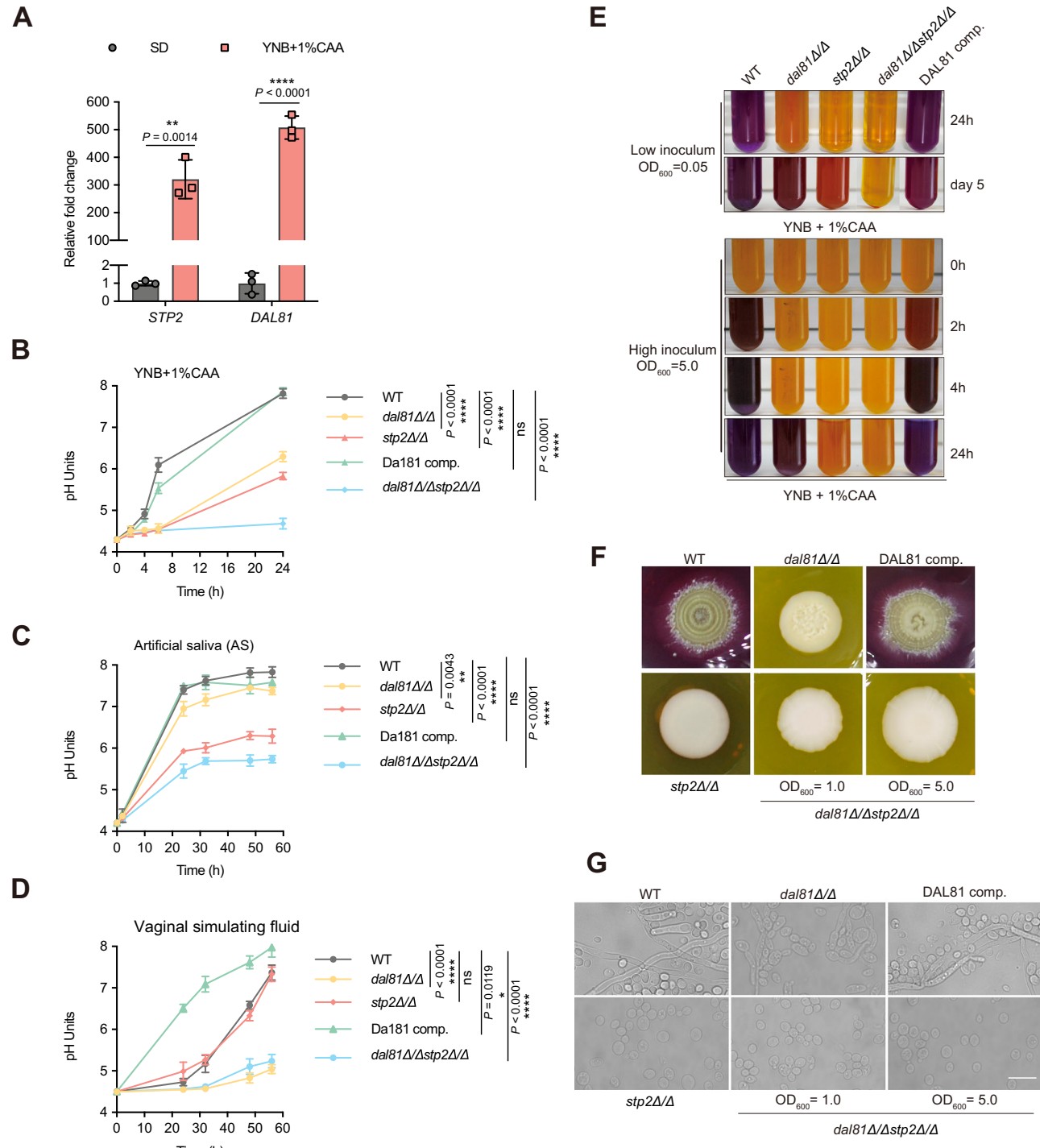

**Fig. 2 | Fungal cells lacking both *DAL81* and *STP2* exhibit more severe pH alkalinization defects than either single mutant. A** RT-qPCR analysis revealed that both *DAL81* and *STP2* were highly induced during pH alkalinization in wild-type cells grown in synthetic dextrose minimal medium (SD) or YNB + 1% CAA medium (initial pH 4.5) for 6 h. **B–F** The *dal81Δ/Δstp2Δ/Δ* double mutant showed significant alkalinization defects. Indicated strains were grown in YNB + 1% CAA medium (initial pH 4.3) **B**, artificial saliva medium (initial pH 4.2) **C** or vaginal simulating fluid medium (initial pH 4.5) **D** at 37 °C, with pH measured at indicated time points. **E** Colorimetric assay. Overnight YPD cultures were washed, diluted to OD$_{600}$ = 0.05

(top) or 5.0 (bottom) in YNB + 1% CAA (initial pH 4.5) with bromocresol purple, and incubated at 37 °C under aeration. **F** Representative images. Overnight YPD cultures were washed, resuspended in water at OD$_{600}$ = 1.0 or 5.0, and 2 µl spotted onto GM-BCP medium (pH 4.0) at 37 °C. **G** Microscopic morphology of colony center from (**F**). Scale bar, 10 µm. Data are mean ± SD of three biological replicates. Statistical significance was assessed by an unpaired two-tailed Student's *t*-test (**A**) and one-way ANOVA with Tukey's test (24 h for **B**, 56 h for **C** and **D**). *$P < 0.05$, ** $P < 0.01$, **** $P < 0.0001$; ns, not significant. Source data for (**A–D**) are provided in the Source Data file.

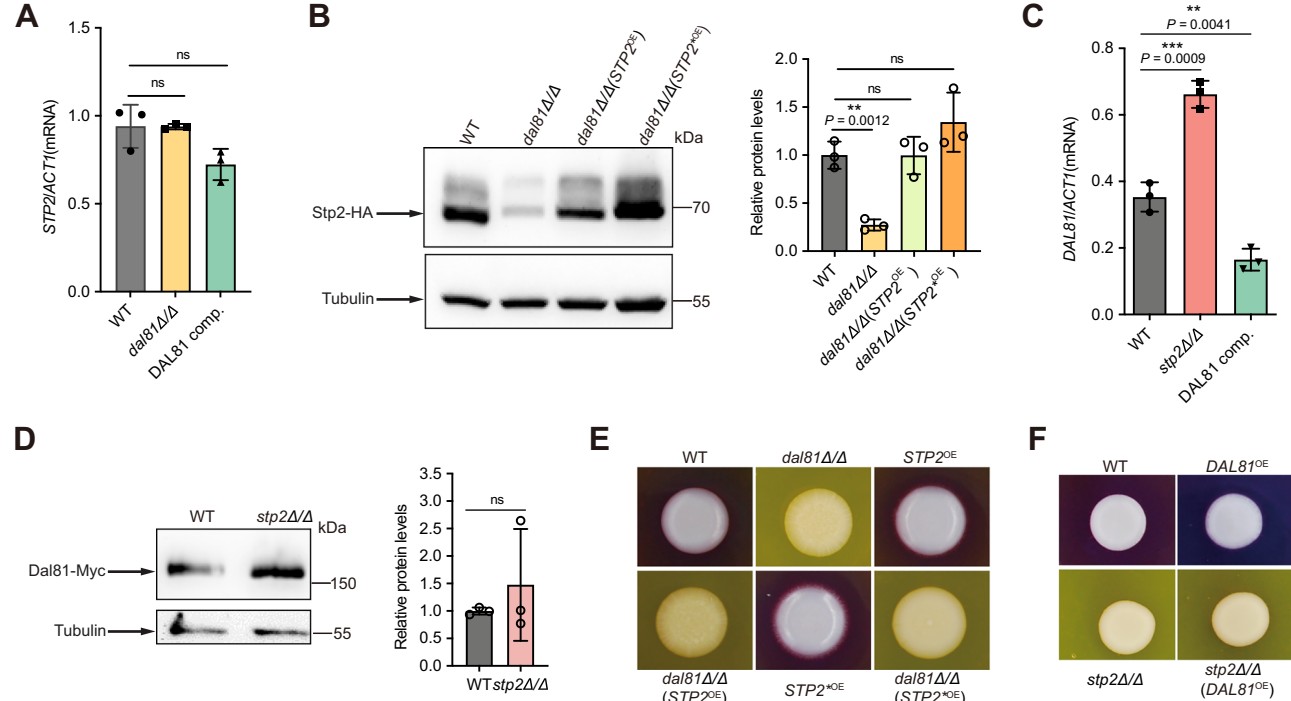

**Fig. 3 | pH alkalinization in *C. albicans* requires the transcriptional activities of both Dal81 and Stp2. A** Dal81 had no effect on the mRNA level of *STP2*, assayed by RT-qPCR analyses. Indicated strains were grown in YNB + 1% CAA (initial pH 4.5) for 6 h. **B** Dal81 influenced Stp2 protein level but not its proteolysis. Immunoblotting assessed Stp2-HA levels in different strains after 6 h in YNB + 1% CAA (initial pH 4.5). Quantification via ImageJ was shown. **C** Stp2 influenced the mRNA level of *DAL81*, assayed by RT-qPCR under the same growth conditions described in (**B**). **D** Immunoblotting showed Stp2 did not alter Dal81 protein level. Dal81-Myc levels were quantified using ImageJ. Indicated strains were grown as in (**B**). **E, F** Overexpression of *STP2* in *dal81Δ/Δ* or *DAL81* in *stp2Δ/Δ* failed to rescue alkalinization defects. Overnight YPD cultures were washed, resuspended to OD$_{600}$ = 1.0, and 2 µl spotted onto GM-BCP medium (pH 4.0) at 37 °C. Plates were photographed after 3 days. Data are mean ± SD of three biological replicates. Statistical significance for (**A**–**D**) was determined by unpaired two-tailed Student's *t*-test. * $P < 0.05$, ** $P < 0.01$, *** $P < 0.001$; ns: not significant. Source data for (**A**–**D**) are provided in the Source Data file.

defective filamentation. In contrast, *stp2Δ/Δ* and the double mutant remained entirely in the yeast form, with no evidence of hyphal differentiation (Fig. 2G). These observations indicate that impaired hyphal development directly contributes to the altered colony morphology and further highlight the coordinated roles of Dal81 and Stp2 in regulating both alkalinization and morphogenesis in *C. albicans*.

### The process of pH alkalinization requires the co-participation of Dal81 and Stp2

The observation that both Dal81 and Stp2 significantly influence pH alkalinization prompted us to investigate their functional relationship. Knocking out *DAL81* had no effect on the mRNA levels of *STP2* but led to a significant reduction in Stp2 protein levels (Fig. 3A, B). Importantly, the N-terminal proteolytic processing of Stp2, a critical post-translational step for its amino acid-induced transcriptional activation[23], remained unaffected in the *dal81Δ/Δ* mutant (Fig. 3B). These results strongly suggest that Dal81 is necessary for maintaining maximal Stp2 protein levels but is not involved in its proteolytic processing. Interestingly, while *DAL81* mRNA levels were significantly upregulated in the *stp2Δ/Δ* mutant compared to the wild-type (Fig. 3C), Dal81 protein levels remained unchanged (Fig. 3D). This discrepancy suggests a decoupling between transcriptional and translational regulation of Dal81 in the absence of Stp2.

Given the decreased mRNA and protein levels of Stp2 in *C. albicans* cells lacking *DAL81*, we asked whether overexpressing *STP2* in the *dal81Δ/Δ* mutant could restore the capacity of alkalinizing the extracellular pH. We therefore generated two overexpression strains: one driving expression of full-length, unprocessed Stp2, and another expressing a truncated, processed form (*STP2**) under the control of

the strong constitutive *TDH3* promoter. The *STP2** variant lacks the N-terminal nuclear exclusion domain (amino acids 1–99) and mimics the proteolytically activated form of Stp2[21]. Notably, the strain (*STP2*^OE^) exhibited a markedly enhanced pH alkalinization phenotype compared to the wild-type (Fig. S8A). To our surprise, overexpression of either the full-length unprocessed or the truncated version of *STP2* failed to rescue the alkalinization defect caused by *DAL81* deletion (Fig. 3E and Fig. S8A, B). Consistently, overexpression of *DAL81* in the *stp2Δ/Δ* background also did not restore alkalinization capacity (Fig. 3F). These results strongly suggest that the relationship between Dal81 and Stp2 does not follow a classic epistatic transcriptional hierarchy. Instead, amino acid-induced pH alkalinization in *C. albicans* appears to require the transcriptional activities of both Dal81 and Stp2, with neither factor alone sufficient to compensate for the loss of the other.

### Dal81 has no effect on nuclear localization of Stp2

Previous studies have demonstrated that under inducing conditions, N-terminally processed Stp2 efficiently enters the nucleus, binds to target gene promoters, and activates their expression[21]. The above-mentioned findings have shown that overexpression of *STP2* in the *dal81Δ/Δ* mutant failed to rescue the alkalinization defect. Moreover, we performed RT-qPCR analyses and found that the mRNA levels of three major Stp2 target genes (*CAN2*, *HGT2*, and *ACS1*) were significantly decreased in the *dal81Δ/Δ* mutant, a pattern mimicking the downregulation observed in the *stp2Δ/Δ* mutant (Fig. S9). Because *DAL81* deletion had no effect on the proteolytic cleavage of Stp2, we asked whether the loss of Dal81 activity could interfere with Stp2 nuclear localization or promoter transactivation and DNA binding, or

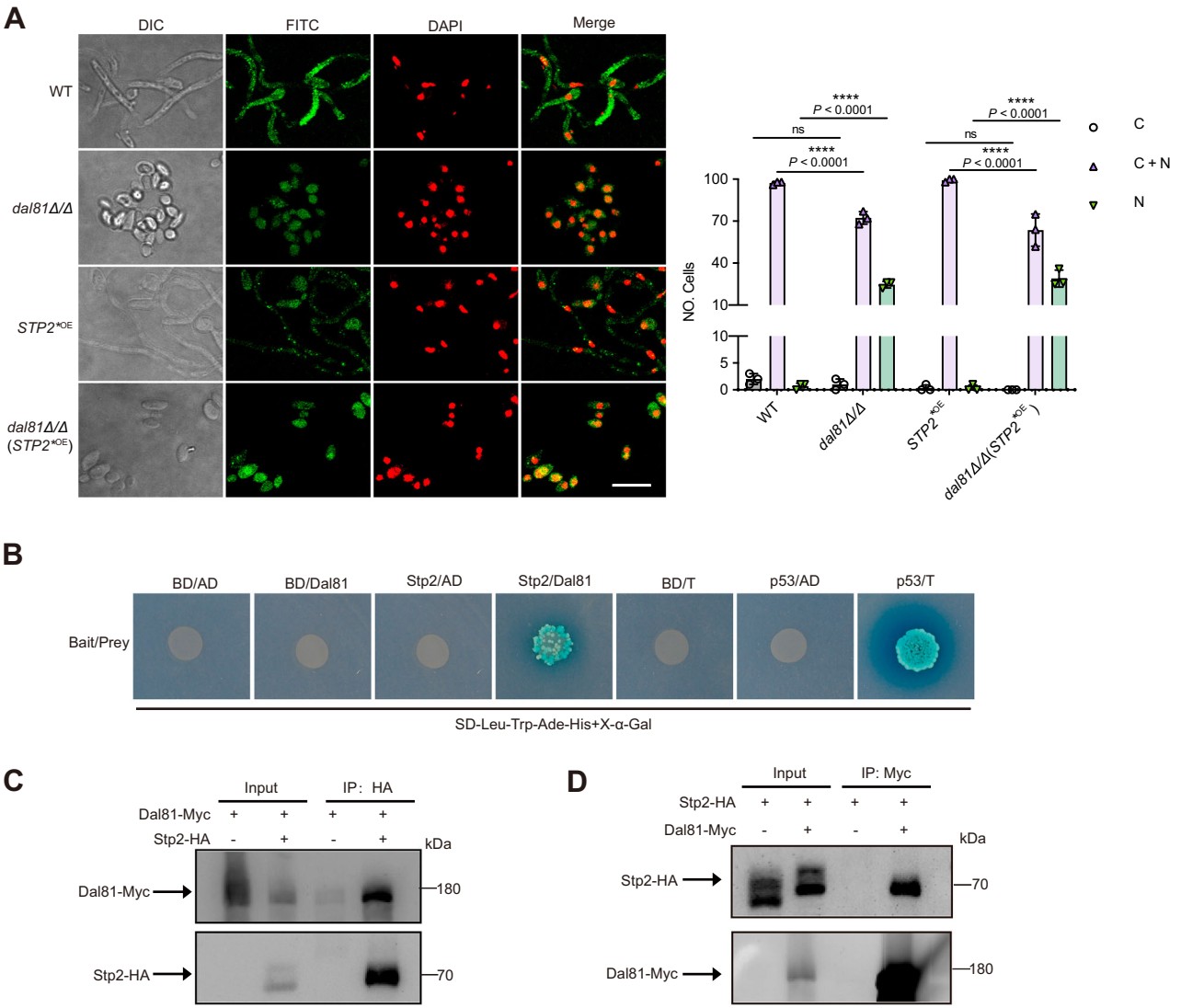

**Fig. 4 | Dal81 physically interacts with Stp2. A** Dal81 had no effect on the nuclear localization of Stp2. Indirect immunofluorescence of strains under alkalinization conditions. DIC (phase images), FITC (Stp2-HA staining), DAPI (DNA staining), and Merge (Stp2-HA/DNA overlay). Right panel quantifies 100 cells/experiment. "C" (≥90% cytoplasmic), "N" (≥90% nuclear), and "C + N" (mixed). Scale bar, 10 µm (all images same magnification). Data are mean ± SD (three biological replicates). Statistical significance via two-way ANOVA with Tukey's test. **** $P < 0.0001$; ns: not significant. **B** Yeast two-hybrid analysis of Dal81-Stp2 interaction. Stp2 was fused to the bait vector, Dal81 to the prey vector. Positive control: Interaction of pGADT7-T with pGBKT7-53. Strains were grown on SD medium lacking leucine, tryptophan, adenine, histidine, plus 40 µg/ml X-α-Gal (SD-Leu-Trp-Ade-His+X-α-Gal). **C, D** Co-immunoprecipitation of Stp2-HA with Dal81-Myc. Strains expressing Stp2-HA or Dal81-Myc were grown in YNB + 1% CAA medium (initial pH4.5). Whole cell extracts were prepared under nondenaturing conditions and were immunoprecipitated with anti-HA (**C**) or anti-c-Myc (**D**) beads. Precipitated proteins were separated by SDS-PAGE and probed with anti-Myc or anti-HA monoclonal antibodies. Experiments were independently replicated twice, with consistent results obtained in each repetition. Source data for (**A, C, D**) are provided as a Source Data file.

both. To explore this, we examined the cellular localization of wild-type and mutant version of Stp2-3xHA by immunofluorescence in amino acids-inducing media. Interestingly, Stp2 retains the ability to translocate to the nucleus independent of Dal81. In wild-type, *dal81Δ/Δ*, and *STP2*\*OE mutant strains, Stp2 localized to both the cytoplasm and nucleus in most cells (Fig. 4A). While the *dal81Δ/Δ* mutant exhibited a slight reduction in cells with dual cytoplasmic-nuclear Stp2 localization compared to the wild-type, this was offset by an increased proportion of cells with exclusively nuclear Stp2. As a result, the total percentage of cells with nuclear Stp2 was comparable between *dal81Δ/Δ* and wild-type strains. Additionally, in YNB + 1% CAA medium, the *dal81Δ/Δ* mutant expressed both full-length and cleaved (active) forms of Stp2 (Fig. 3B), confirming that post-translational processing and activation of Stp2 occur normally in the absence of Dal81. Together, these findings suggest that Dal81 is not required for the nuclear localization or functional activation of Stp2.

## Dal81 physically interacts with Stp2

To investigate whether Dal81 influences the promoter transactivation and DNA binding capacity of Stp2, we first tested for physical interaction between the two transcription factors. A yeast two-hybrid analysis revealed a robust interaction between Stp2 and Dal81. When Stp2 was fused to the Gal4 DNA binding domain (BD-Stp2) and Dal81 to the Gal4 activation domain (AD-Dal81), a strong reporter signal was detected (Fig. 4B). No interaction was observed when BD-Stp2 or AD-Dal81 was co-expressed with empty vectors, confirming specificity. To validate this interaction in a physiological context, we performed co-immunoprecipitation (Co-IP) experiments. Stp2 was tagged with a C-terminal hemagglutinin (HA) epitope and co-expressed with Dal81-Myc in *C. albicans*. Importantly, strains expressing HA-tagged Stp2, Myc-tagged Dal81, or both tagged proteins exhibited growth and alkalization phenotypes indistinguishable from the untagged wild-type strain when cultured in YNB + 1% CAA medium (Fig. S10), confirming

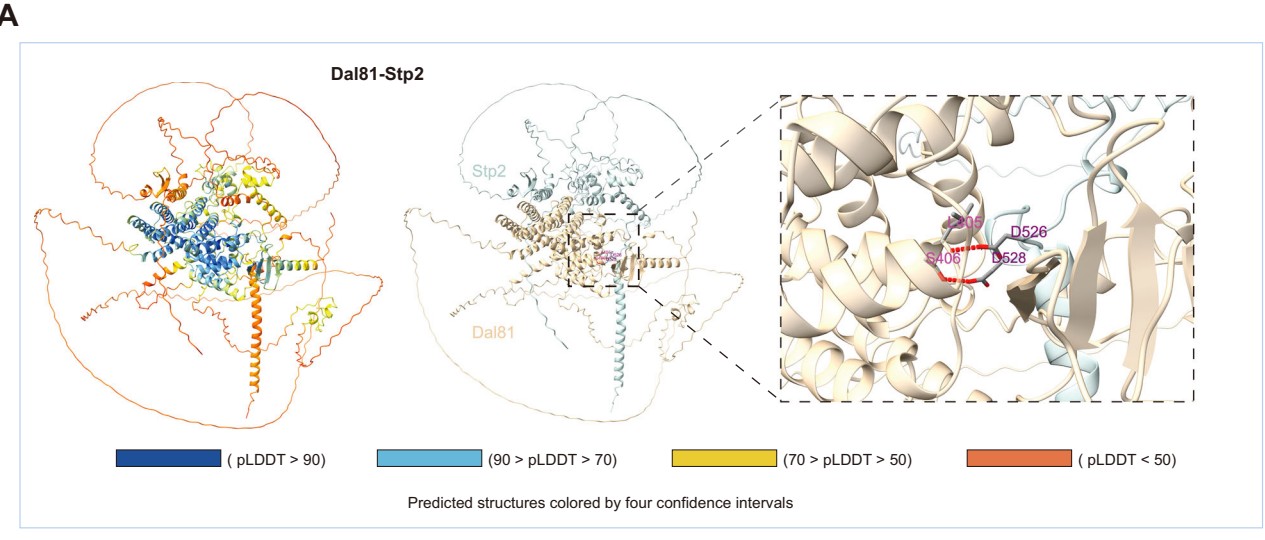

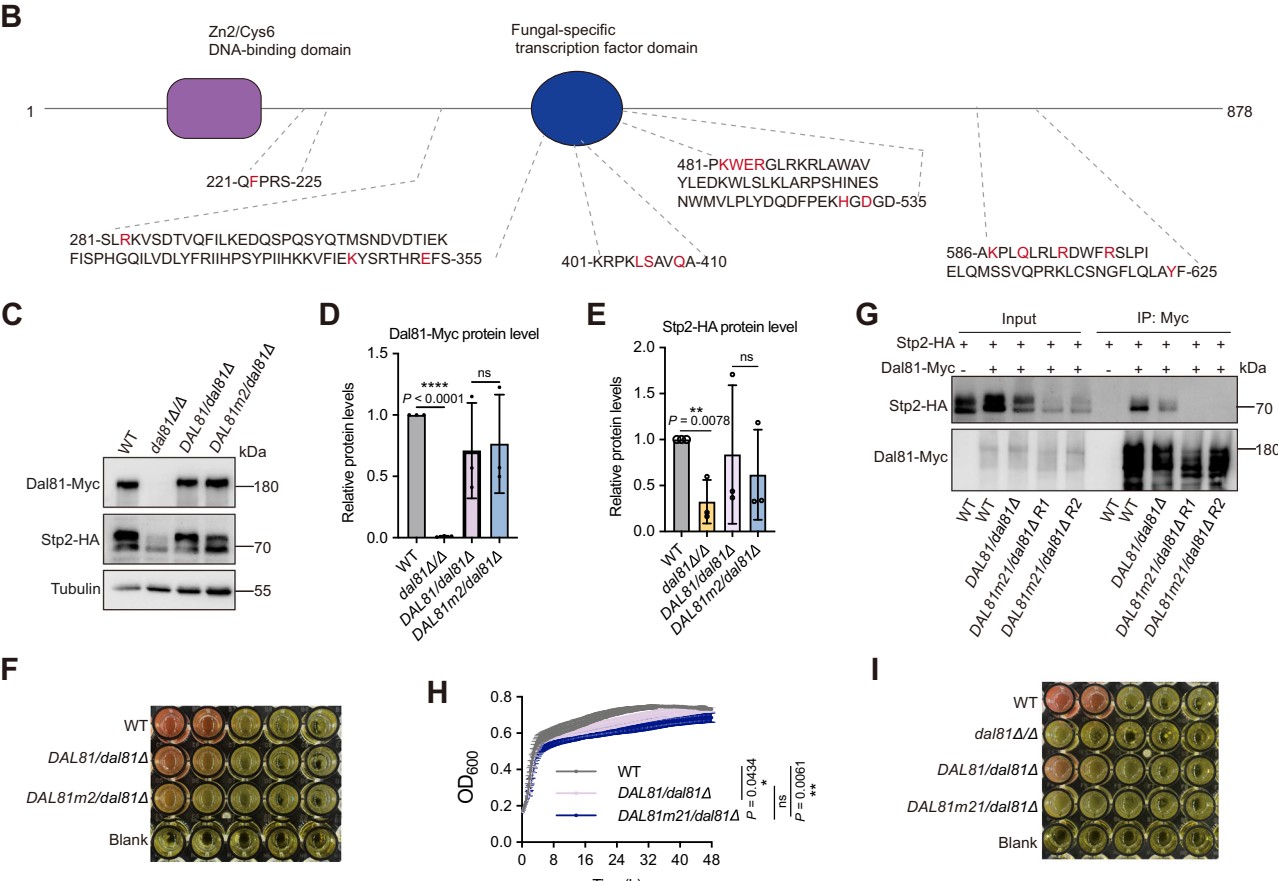

**Fig. 5 | Identification of critical interaction residues between Dal81 and Stp2 responsible for pH alkalinization. A** AlphaFold 3 in silico docking simulations revealed that Dal81 and Stp2 formed two important hydrogen bonds (Dal81[L405] - Stp2[D526] and Dal81[S406] - Stp2[D528]). Dal81 is visualized as yellowish brown cartoons, Stp2 as light blue. Predicted interaction structures are categorized by confidence intervals. **B** Schematics of *C. albicans* Dal81, with 21 AlphaFold3-predicted Stp2-interacting residues highlighted in red. **C–E** Immunoblots (**C**) of Dal81-Myc and Stp2-HA expression in indicated strains after 6 h in YNB + 1% CAA (initial pH 4.5). ImageJ quantification of Dal81-Myc (**D**) and Stp2-HA (**E**) band intensities. Data are mean ± SD (three biological replicates). Statistical significance via unpaired two-tailed Student's *t*-test. * $P < 0.05$, ** $P < 0.01$; ns, not significant. **F** Alkalinization analysis of the specified strains was conducted following the procedures described in Fig. 1B. **G** Stp2-HA failed to co-immunoprecipitate with Dal81m21-Myc. Strains co-expressing Stp2-HA and Dal81m21-Myc were grown in YNB + 1% CAA (initial pH 4.5). Whole cell extracts were prepared under nondenaturing conditions, and immuno-precipitated with anti-Myc beads. Precipitates were immunoblotted with anti-HA or anti-Myc antibodies. R1 and R2 denote biological replicates. **H** Growth curves of indicated strains in medium 199 (initial pH 4.0) at 37 °C, with $OD_{600}$ measured every 15 min (BioTek Synergy 2 Multi-mode Microplate Reader). Data are mean ± SD (three independent biological replicates). Statistical analysis via one-way ANOVA with Tukey's multiple comparison test at 24 h. * $P < 0.05$, ** $P < 0.01$. **I** Alkalinization assay (as in Fig. 1B) showing pH modulation defects in *DAL81m21/dal81Δ*. Source data for (**C–E, G, H**) are provided in the Source Data file.

that the epitope tags do not disrupt Dal81 or Stp2 function. In Co-IP assays, immunoprecipitation of Stp2-HA using an anti-HA antibody co-precipitated Dal81-Myc, as detected by Western blot (Fig. 4C). Reciprocally, immunoprecipitation of Dal81-Myc with an anti-Myc antibody co-precipitated Stp2-HA (Fig. 4D). These results provide direct evidence that Dal81 and Stp2 physically interact in vivo.

## The physical interaction between Dal81 and Stp2 is required for pH alkalinization of *C. albicans*

Building on our earlier findings of a physical interaction between Dal81 and Stp2, we next aimed to identify the specific residues mediating this interaction. A sequence analysis using InterPro (https://www.ebi.ac.uk/interpro/) revealed that *C. albicans* Dal81 contains a Zn2/Cys6 DNA-binding domain (amino acids 110-159) and a fungal-specific transcription factor domain (amino acids 325-597) (Fig. S11). The Zn2/Cys6 domain is characterized by alpha-helical structures that coordinate zinc atoms for DNA binding, while the fungal-specific domain comprises several structural motifs implicated in transcriptional regulation.

To investigate the interaction interface, we predicted the structure of the Dal81-Stp2 complex using AlphaFold 3 by inputting the full-length amino acid sequences of both proteins. Strikingly, the predicted protein model suggested a lock-and-key configuration, wherein Stp2 fits into a complementary binding groove on Dal81 (Fig. 5A). Based on the AlphaFold-predicted model and structural analysis using ChimeraX[30], we identified 21 candidate residues on protein Dal81 that are potentially critical for its interaction with protein Stp2 (Fig. 5B). To experimentally validate these in silico predictions, we generated site-directed mutants in which the two top-ranked predicted interaction residues, Dal81$^{L405}$ and Dal81$^{S406}$, were substituted with alanine (Fig. 5B). Using the Stp2-HA/*dal81Δ/Δ* background strain described in Fig. 3B, we constructed a Myc-tagged Dal81 mutant strains (*DAL81m2/dal81Δ*) harboring both point mutations, alongside a Myc-tagged wild-type *DAL81* control strain (*DAL81/dal81Δ*). Immunoblot analysis of cell lysates from these strains demonstrated that the alanine substitutions had only a minor impact on the protein levels of Dal81 and Stp2 (Fig. 5C–E), suggesting that the mutations did not impair protein stability.

Unexpectedly, we observed that the *DAL81m2/dal81Δ* mutant displayed normal alkalinization activity on medium 199 (Fig. 5F), indicating that substitution of Dal81$^{L405}$ and Dal81$^{S406}$ alone is insufficient to disrupt the Dal81-Stp2 interaction or its downstream functional output. These results suggest that additional residues likely contribute to the interaction interface required for pH alkalinization. To assess the functional importance of the predicted interaction interface between Dal81 and Stp2, we constructed a mutant strain (*DAL81m21/dal81Δ*) in which all 21 putative binding residues identified from the docking analysis (Fig. 5B) were substituted with alanine. Co-immunoprecipitation (Co-IP) analysis revealed that the physical interaction between Dal81 and Stp2 was abolished in the *DAL81m21/dal81Δ* strain (Fig. 5G), confirming the critical role of these residues in mediating protein–protein interaction. Interestingly, these amino acid substitutions did not affect fungal growth, as the *DAL81m21/dal81Δ* mutant exhibited growth comparable to its wild-type control strain (*DAL81/dal81Δ*) (Fig. 5H). However, *DAL81m21/dal81Δ* displayed a severe defect in extracellular pH alkalinization when cultured in medium 199 (Fig. 5I), demonstrating that the physical interaction between Dal81 and Stp2 is functionally required for this phenotype.

## The physical interaction between Dal81 and Stp2 promotes target gene expression

To explore the functional relationship between Dal81 and Stp2 in regulating pH alkalinization of *C. albicans*, we used RNA-seq to determine the transcriptional profiles and pH alkalinization-responsive gene networks that were dependent on the two transcription regulators Dal81 and Stp2. Three independent RNA-seq analyses were performed to profile the *dal81Δ/Δ* and *stp2Δ/Δ* mutant strains, alongside two

reference strains (SC5314 and SN250), under two culture conditions: SD or YNB + 1% CAA (YC, initial pH 4.5) with a 6-hour incubation period. The inclusion of two reference strains was necessitated by the auxotrophic nature of the SN250 strain for arginine, which prevents its growth in SD medium. Principal component analysis (PCA) (Fig. 6A) and heat map visualization (Fig. 6B) revealed a high correlation in gene expression profiles between the *dal81Δ/Δ* and *stp2Δ/Δ* mutant strains, suggesting comparable transcriptional responses under the tested conditions. Using a stringent cutoff (at least a 2-fold change between baseline and treatment samples, FDR ≤ 0.1), we identified 2516 differentially expressed genes (DEGs) in wild-type cells upon alkaline pH induction, defined as "alkalinization-responsive genes" (Data S1). Among these, 1280 alkalinization-responsive genes were regulated by Dal81, 1277 genes by Stp2, and 1120 were coregulated by both factors (Fig. S12A, B, Fig. 6C, Data S1). Gene Ontology (GO) analysis further supported functional convergences, as we observed that the alkalinization-responsive gene sets regulated by the two regulators were enriched in highly similar physiological processes (Fig. 6D, Data S2). Volcano plots also illustrated that the majority of DEGs (both up- and down-regulated) in the *dal81Δ/Δ* mutant were similarly misregulated in the *stp2Δ/Δ* mutant under alkaline pH-inducing conditions (Fig. 6E). Importantly, among DEGs regulated by either Stp2 or Dal81, 77.9% showed overlapping expression between the two regulators (Fig. 6C), with the majority involved in the metabolism of carboxylic acids, organic acids, oxoacids and amino acids (Fig. 6D). Our RNA-seq data thus strongly suggest that Dal81 and Stp2 function similarly in the regulation of transcriptional response to pH alkalinization.

To determine whether Dal81 and Stp2 act together and directly participate in the induction of alkalinization-responsive genes, we further performed chromatin immunoprecipitation followed by sequencing (ChIP-seq) to identify the promoters bound by these two regulators. Both Dal81-13xMyc and Stp2-3xHA were expressed under the control of their endogenous promoters. After culturing the respective strains in YNB + 1% CAA (initial pH 4.5) for 6 h, cells were harvested, and cross-linked chromatin from Dal81-13xMyc, Stp2-3xHA, or untagged control strains was precipitated using anti-Myc or anti-HA antibodies. Our analysis identified 2,624 and 2,599 specific binding sites for Dal81 and Stp2, respectively, with 85% of these sites localized to promoter regions (Fig. 7A). Of significance, HOMER motif analysis revealed a shared core DNA-binding motif recognized by both Dal81 and Stp2 (Fig. 7B). To infer direct target genes, we integrated ChIP-seq data with differential gene expression profiles[31], yielding 240 target genes for Dal81 and 211 for Stp2 (Fig. S13A, B, Data S3 and Data S4). Among these, 99 genes were identified as common targets of both regulators (Fig. 7C, Data S4). Functional classification of these shared targets showed that 42 genes were synergistically activated, 56 genes were coordinately repressed, and 1 was antagonistically regulated by both regulators (Fig. S13C, Data S4). Strikingly, Gene ontology (GO) enrichment analysis linked the biological processes of these common targets to the metabolism of organic acids, oxoacids, carboxylic acids, and amino acids (Fig. 7D, Data S5). Key examples of these target genes include the putative NAD-specific glutamate dehydrogenase-encoding gene, *GDH2* (Fig. S14A), the putative proline oxidase-encoding gene, *PUT1* (Fig. S14B), and the phosphoenolpyruvate carboxykinase-encoding gene, *PCK1* (Fig. S14C), which are involved in multiple biological processes (Fig. S13C, Data S5) and have previously been implicated in pH alkalinization[27,32].

While both regulators shared target genes enriched in the metabolism of organic acids, oxoacids, carboxylic acids, and amino acids, Stp2 uniquely controlled a broader repertoire of genes within these pathways (Fig. S15, Data S6). For example, Stp2 specifically regulated genes involved in organic acid, carboxylic acid, and amino acid transport (Fig. S15, Data S6), underscoring its role in nutrient uptake and metabolic flux. In contrast, Dal81 exhibited specialization in carbohydrate metabolism, with exclusive regulation of genes

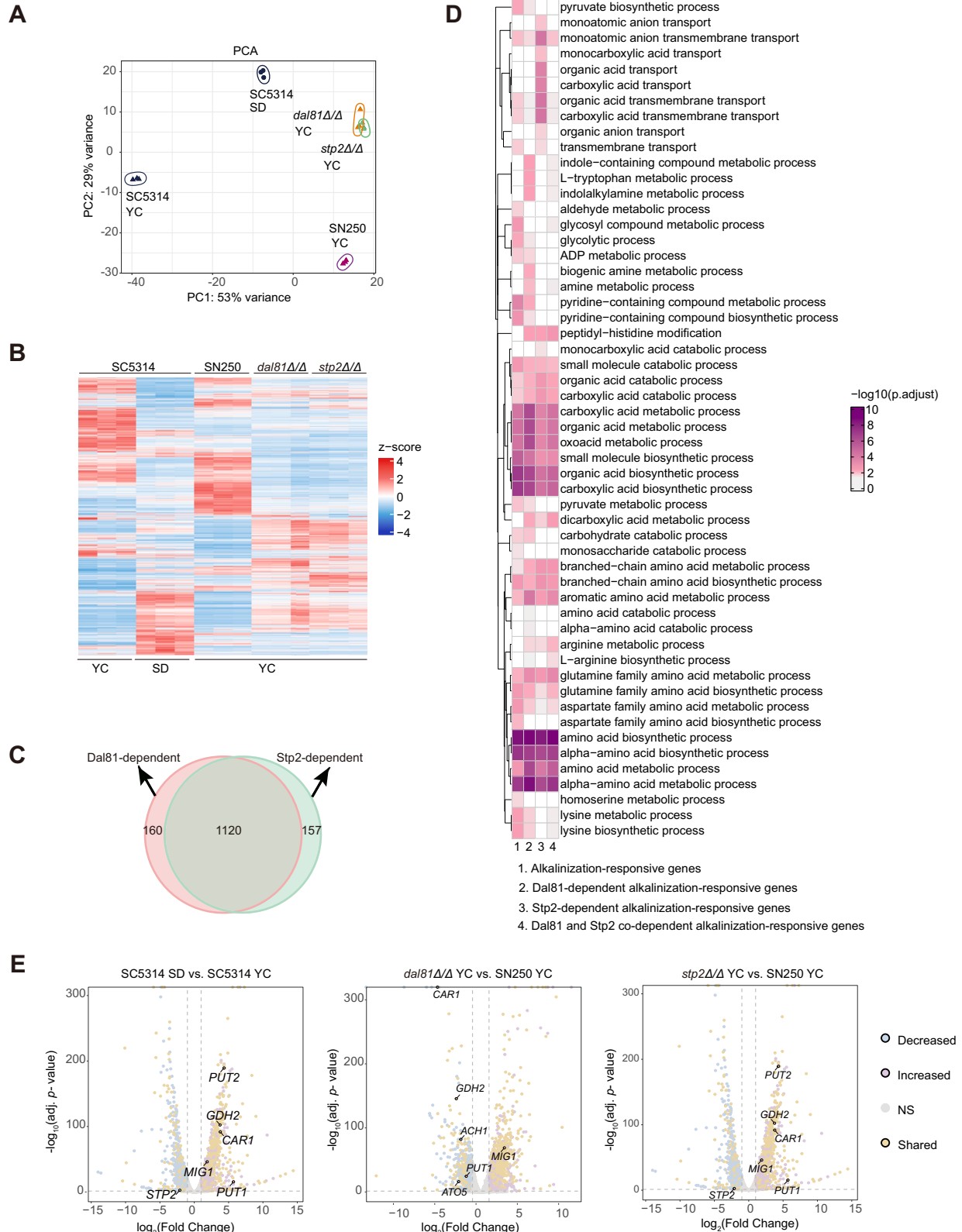

1. Alkalinization-responsive genes
2. Dal81-dependent alkalinization-responsive genes
3. Stp2-dependent alkalinization-responsive genes
4. Dal81 and Stp2 co-dependent alkalinization-responsive genes

involved in sugar (*e.g.*, hexoses and glucose) catabolism (Fig. S15, Data S6). These findings suggest that Dal81 and Stp2 coordinate shared metabolic processes (e.g., amino acid and organic acid metabolism) while governing distinct, non-overlapping pathways (Stp2 in transport; Dal81 in sugar metabolism), highlighting their functional complementarity in shaping cellular metabolism and environmental adaptation.

The direct binding of Dal81 and Stp2 to downstream target genes such as *GDH2*, *PUT1* and *PCK1* was further validated by ChIP-qPCR (Fig. 7E, F, Fig. S14D). Interestingly, deletion of *DAL81* resulted in the loss of Stp2 enrichment at the promoters of these common targets, accompanied by downregulated gene expression (Fig. 7H–J). In line with this observation, knocking out *STP2* yielded the same results. Additionally, reduced binding of Stp2 to target genes *GDH2*, *PUT1*, and

**Fig. 6 | The transcriptional profiles regulated by Dal81 and Stp2 are highly similar. A** Principal component analysis (PCA) of *C. albicans* gene expression at 6 h after transferring to SD or YC medium. SD: synthetic dextrose minimal medium; YC: YNB + 1% CAA, initial pH 4.5. **B** Heat map of gene expression in the indicated strain under different conditions. Each vertical line represents a sample, with gene expression displayed horizontally. Expression values were mean-normalized per gene and shown as z scores relative to the mean. **C** Venn diagram of alkalinization-responsive gene sets controlled by Dal81 and Stp2, based on RNA-Seq analysis of knockout mutants versus SN250. **D** Functional enrichment analysis of alkalinization-responsive genes specifically regulated by Dal81, Stp2, or commonly

by both. *P* values were calculated via one-sided Fisher's exact test with Benjamini-Hochberg correction; terms with FDR-adjusted $p < 0.05$ are shown as −log10 (*p*. adjust). GO terms are clustered by Lin's semantic similarity. **E** Volcano plots of differentially expressed genes (DEGs). SC5314 in SD vs. YC medium for 6 h, and *dal81Δ/Δ* or *stp2Δ/Δ* vs. SN250 in YC medium for 6 h. Vertical dotted lines indicate 2-fold change, and horizontal dotted lines mark adjusted $p = 0.05$. Statistical significance was determined using DESeq2 (two-sided Wald test with Benjamini-Hochberg correction). Shared DEGs across conditions are highlighted in yellow. Labeled genes were previously reported, with their deletion mutants showing alkalinization defects.

*PCK1* was observed in the *DAL81m21/dal81Δ* strains, where the Dal81–Stp2 interaction was disrupted by mutations, compared to the control strain *DAL81/dal81Δ* (Fig. 7G). RT-qPCR analysis of these three target genes (Fig. 7K–M) revealed a downregulated expression trend for *GDH2* and *PCK1* in the residue mutant strains (Fig. 7K and L). These data strongly support that Dal81 forms a complex with Stp2 and co-regulates the expression of most of their common target genes.

### Dal81 and Stp2 are essential for regulating the commensalism and pathogenicity of *C. albicans*

*C. albicans* typically survives as a commensal of the mammalian microbiome and also acts as an opportunistic pathogen causing disseminated infection[33]. Notably, pH levels in mammalian mucosal niches vary widely, ranging from acidic to alkaline, and fungi like *C. albicans* must adapt to this pH spectrum to survive, indicating that pH alkalinization, co-regulated by Dal81 and Stp2, may contribute to the commensalism and/or virulence of *C. albicans*. To test this speculation, we first evaluated the intestinal colonization capacity of the *dal81Δ/Δ* or *stp2Δ/Δ* knockout strain using the intestinal commensalism mouse model of *C. albicans*[34]. Knockout of either *DAL81* or *STP2* led to a significant decrease in intestinal colonization (Fig. 8A). Most significantly, simultaneous deletion of both *DAL81* and *STP2* resulted in the complete absence of *C. albicans* colonization within the intestine (Fig. 8B), strongly validating their roles as crucial regulators of *C. albicans* gastrointestinal colonization. Given that successful gut colonization requires interactions with both the host and the microbes, we further evaluated the specific contributions of Dal81 and Stp2 using competitive infection assays. Immunosuppressed ICR mice were gavaged with 1:1 mixtures of wild-type *C. albicans* and either the *dal81Δ/Δ* or *stp2Δ/Δ* mutant, and the abundance of each strain in fecal pellets was monitored by qPCR over 8 days. Throughout the time course, both the *dal81Δ/Δ* and *stp2Δ/Δ* mutants showed a consistent competitive disadvantage relative to the wild-type strain (Fig. 8C), indicating that Dal81 and Stp2 promote commensalism through mechanisms that are likely independent of the host immune response, as the assay was performed in immunosuppressed mice.

On the other hand, the reduced survival of *dal81Δ/Δ* and *stp2Δ/Δ* mutants in macrophages raised the question of whether these strains would be compromised during disseminated infection. Therefore, we tested the *dal81Δ/Δ* and *stp2Δ/Δ* strains in the standard mouse tail-vein injection model of disseminated hematogenous candidiasis[35]. Both mutants exhibited significant virulence attenuation compared with the wild-type and *DAL81* complemented strains (Fig. 8D), as we found that mice infected with the wild-type strain had a median survival time of 4 days, whereas survival times for those infected with the mutant strains remained undefined due to prolonged survival, indicating that Dal81 and Stp2 are required for virulence in this infection model. Statistical analysis confirmed that the survival curve for the *dal81Δ/Δ* and *stp2Δ/Δ* strains were significantly different from those of the wild-type or complemented cells ($P < 0.0001$). Strikingly, the *dal81Δ/Δstp2Δ/Δ* double knockout strain displayed complete virulence attenuation relative to the wild-type (Fig. 8E), underscoring the indispensable and synergistic roles of Dal81 and Stp2 in fungal pathogenesis. To further explore how the Dal81-Stp2 interaction and pH alkalinization

contribute to *C. albicans* virulence, we evaluated the pathogenic potential of the *DAL81m21/dal81Δ* mutant, defective in Dal81-Stp2 interaction, in the systemic infection model. Notably, this mutant was significantly attenuated in virulence compared to the wild-type control, as we observed that mice infected with the *DAL81/dal81Δ* control strain had an average survival time of 15 days, whereas survival in the *DAL81m21/dal81Δ* group remained undefined due to limited mortality over the experimental period (Fig. 8F). These findings identify specific Dal81-Stp2 interaction sites as critical determinants of both pH alkalinization and *C. albicans* virulence.

In summary, our animal model experiments fully demonstrate that Dal81 and Stp2 coordinately regulate environmental alkalinization and play a pivotal role in modulating both commensalism and pathogenicity of *C. albicans*, linking their molecular interactions to in vivo fitness and virulence.

## Discussion

Microorganisms encounter various stresses during their inhabitance, and adaptation serves as a crucial strategy for them to enhance the utilization of non-preferred substrates and increase the tolerance to stress. Fungi, in particular, have the ability to thrive in a diverse range of habitats and are considered to be among the most extremotolerance microorganisms due to their morphological plasticity and diverse lifestyles. During its long-term life history in humans, *C. albicans*, a common fungal commensal and opportunistic pathogen, has evolved sophisticated strategies to rapidly sense, adapt to and even change the fluctuating microenvironments within the host. Meanwhile, the ability to adapt to or modify the environmental pH has been known to be crucial for the survival and growth of this fungus in various host niches[36]. Literatures have shown that *C. albicans* is able to actively alkalinize the host environment[29] and the transcription factor Stp2 was identified to be the master regulator controlling the process of pH alkalinization[15]. Here, we provide compelling evidence that the transcription factor Dal81 also plays a critical role in pH alkalinization in *C. albicans*. Notably, our findings demonstrate that this important environmental pH-modifying behavior requires the coordinated activity of Dal81 and Stp2. Deletion of either transcription factor significantly impairs the organism's ability to elevate extracellular pH. To illustrate the proposed mechanism, we developed a schematic model (Fig. S17) depicting the coordination between Dal81 and Stp2 in regulating gene transcription induced by environmental alkalinization. This model highlights the cooperative interaction of the two factors within the nucleus, where they orchestrate the expression of downstream genes required for the survival and propagation of *C. albicans* in the host.

Dal81 is a fungus-specific protein characterized by a Gal4-like zinc finger DNA binding domain[37] and its regulatory role in gene expression has been addressed in multiple fungi. For example, Dal81 in *S. cerevisiae* was found to be important for nitrogen metabolism by regulating the transcriptional activation of a significantly large number of genes associated with the catabolism of nitrogen sources such as urea, allantoin, arginine, and γ-aminobutyrate (GABA)[38–41]. The lack of *DAL81* hinders the maximal induction of these genes[42] such as the GABA-inducible genes involved in GABA utilization (*UGA1*, *UGA2*, and

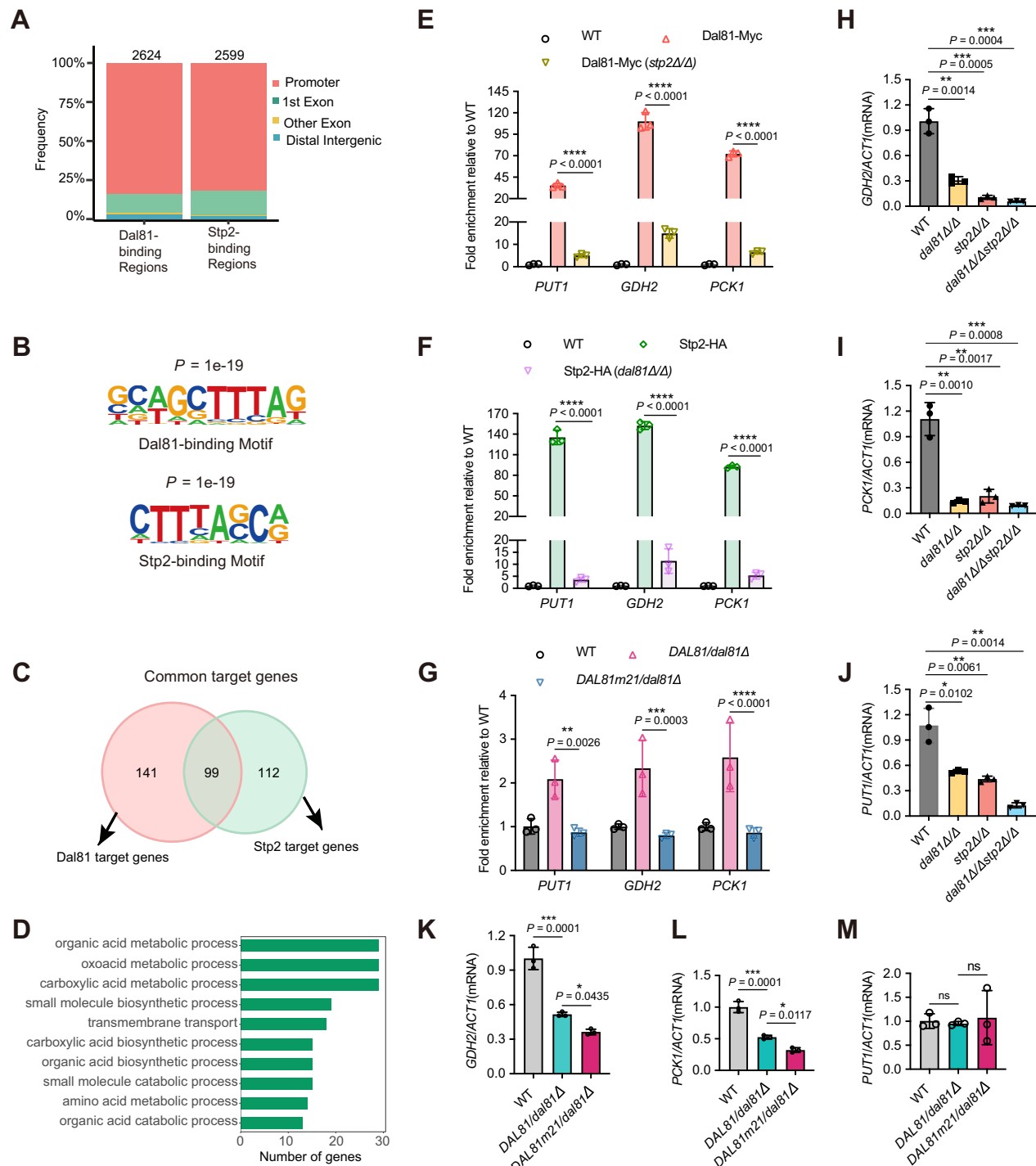

**Fig. 7 | Dal81 and Stp2 coordinate to activate transcription of alkalinization-responsive genes. A** Genome-wide distribution of candidate Dal81 or Stp2 binding regions from ChIP-seq assays in untagged wild-type, Stp2-HA, and Dal81-Myc strains grown in alkalinization-inducing medium (YNB + 1% CAA, initial pH 4.5). **B** De novo motif analysis using HOMER identified consensus binding motifs for Dal81 and Stp2 at their binding sites. **C** Venn diagram showing overlap between Dal81 and Stp2 binding sites. **D** GO enrichment analysis of common targets genes shared by Dal81 and Stp2. **E** ChIP-qPCR validation for Dal81 binding to *PUT1*, *GDH2,* and *PCK1* in Dal81-Myc or Dal81-Myc (*stp2Δ/Δ*) strains, shown as fold enrichment of the signal from immunoprecipitation relative to the input DNA. WT served as the negative

control. **F** ChIP-qPCR validation for Stp2 binding to *PUT1*, *GDH2* and *PCK1* in Stp2-HA or Stp2-HA (*dal81Δ/Δ*) strains. **G** ChIP-qPCR validation for Stp2 binding to *PUT1*, *GDH2* and *PCK1* in Dal81-Myc or Dal81m21-Myc strains. **H**–**M** RT-qPCR analysis of *PUT1*, *GDH2* and *PCK1* mRNA abundance in different strains under alkalinization conditions. Values for each gene were normalized against *ACT1*. Data are means ± SD of three biological replicates. Statistical significance via unpaired two-tailed Student's *t*-test (**E, F, H**–**J**) and one-way ANOVA with Tukey's multiple comparisons test (**K**–**M**). * $P < 0.05$, ** $P < 0.01$, ***$P < 0.001$, **** $P < 0.0001$; ns: not significant. Source data for (**E**–**M**) are provided as a Source Data file.

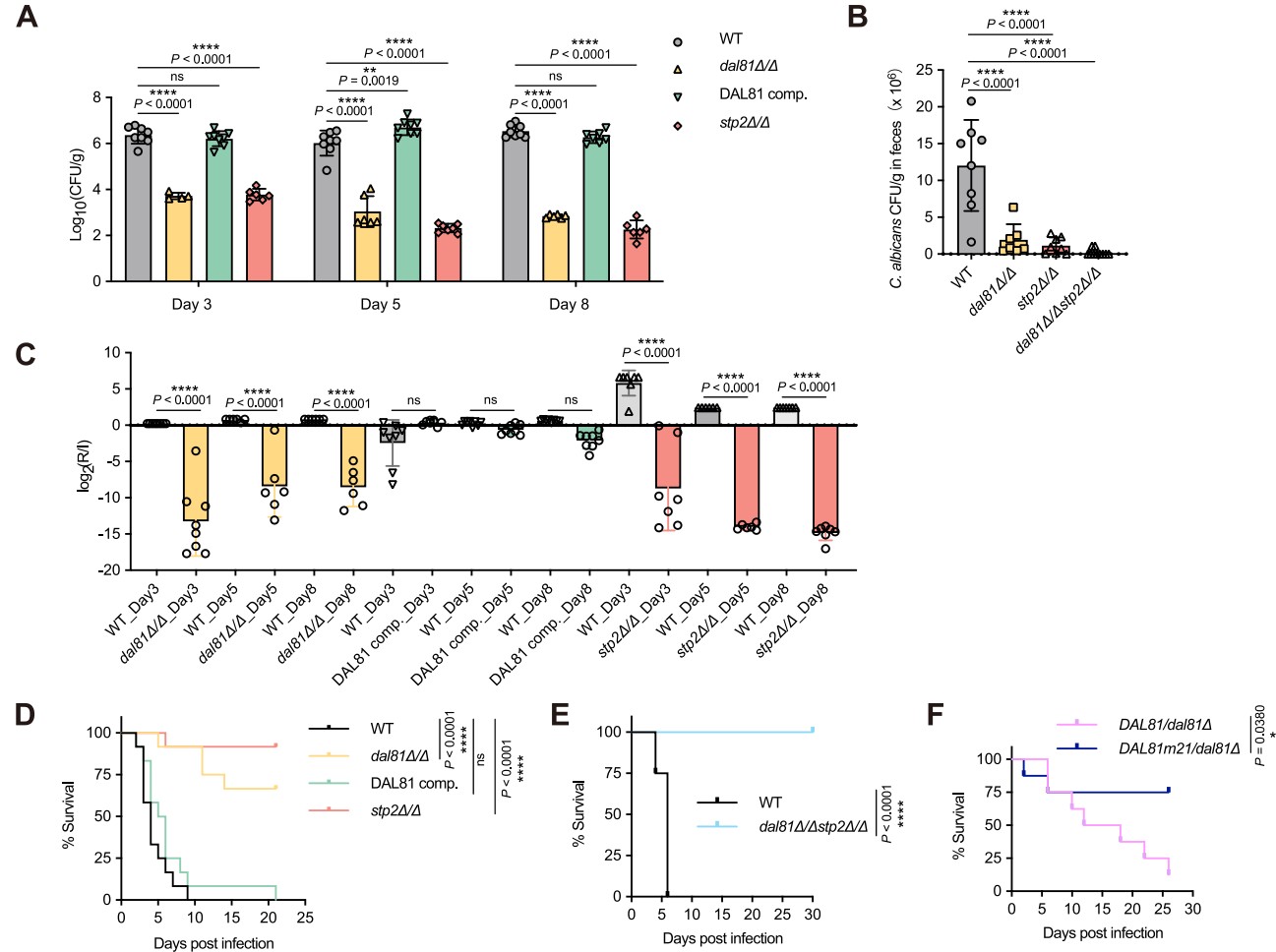

**Fig. 8 | Dal81 and Stp2 are key regulators of *C. albicans* commensalism and pathogenesis. A** *C. albicans* *dal81Δ/Δ* and *stp2Δ/Δ* mutant strains showed defective gastrointestinal colonization compared to wild-type (WT) or DAL81comp. strains. Female C57BL/6 mice (*n* = 8) were orally gavaged with $10^8$ CFU of indicated strains and colonization was measured in fecal pellets on days 3, 5 and 8 post-inoculations via CFU/g quantification. **B** The *dal81Δ/Δstp2Δ/Δ* double mutant exhibited complete colonization failure. Mice (*n* = 8) were gavaged as in (**A**), with colonization assessed on day 3. **C** Dal81 and Stp2 promote gut commensalism in immunosuppressed ICR mice (*n* = 8/group), inoculated with 1:1 mixtures of WT and *dal81Δ/Δ*, DAL81comp., or *stp2Δ/Δ* (1 × $10^8$ CFU total). Strain abundance in inoculum (I) and after recovery from fecal pellets (R) was quantified by strain-specific qPCR. **D** Deletion of *DAL81* or

*STP2* attenuated *C. albicans* virulence, with Wild-type *DAL81* complementation restoring function. C57BL/6 mice (*n* = 12) were tail-vein infected with 2 × $10^5$ CFU of indicated strains and survival was monitored at the specified timepoints. **E** The *dal81Δ/Δstp2Δ/Δ* double mutant showed complete virulence loss. Mice (*n* = 8) were infected with 1 × $10^6$ CFU as in (**D**). **F** Survival of mice (*n* = 8) after intravenous (i.v.) challenge with the *DAL81m21/dal81Δ* mutant. Experiments were repeated twice with similar results. Statistical significance via two-way ANOVA with Tukey's test (**A, C**), and one-way ANOVA with Tukey's test (**B**). Survival data (**D, E**) were evaluated by Kaplan-Meier and log-rank (Mantel–Cox) test. ns, not significant. Source data are provided as a Source Data file.

*UGA4*[38,43] and the allophanate-inducible genes involved in the utilization of urea (*DUR1-2* and *DUR3*), allantoate (*DAL7*), and arginine (*CAR2*)[44–46]. Interestingly, Dal81 is also required for the activation of *AGP1* encoding an amino acid permease that mediates the uptake of a wide range of amino acids from the external environment[47,48]. Mechanistically, Dal81 is activated by the Ssy1-Ptr3-Ssy5 (SPS) sensor system[42,47,48] and acts to facilitate the binding of a list of other transcription factors to the promoters of their downstream target genes. For example, the activation of Dal81 was found to significantly promote the binding of Stp1 and Stp2 to the *AGP1* promoter[43,49–51] as well as the recruitment of Uga3 and Gall11 to the promoters of *UGA1* and *UGA4*[50]. Interestingly, a divergent functional role for the transcriptional regulator Dal81 has been characterized in *Candida parapsilosis*. In this organism, the induction of genes involved in GABA metabolism occurs independent of Dal81 and its deletion does not impair the ability of *C. parapsilosis* to acquire nitrogen from either GABA or allantoin[52], marking a key departure from its role in *C. albicans*. Instead, Dal81 serves to repress arginine synthesis under preferred nitrogen

conditions[52]. A further transcription network analysis suggests that Dal81 either initiates or assists Stp2 during the initiation of nitrogen mobilization from the vacuolar storage via Avt11[53]. In *C. albicans*, we found that Dal81 is required for regulation of gene expression involved in metabolism of organic acids, oxoacids, carboxylic acids, amino acids and carbohydrates. More importantly, the induction of these Dal81 targets enhances its crucial roles in environmental pH alkalinization, yeast-to-hyphal morphogenesis and cellular fitness, linking its role to *C. albicans* commensalism and pathogenicity.

Studies have demonstrated that the utilization of amino acids in *S. cerevisiae* and *C. albicans* requires the regulatory activities of two transcript factors, Stp1 and Stp2[51]. Stp2 regulates the expression of genes encoding amino acid permeases and catabolic machinery of amino acids[16,23], whereas Stp1 is considered as an important regulator in controlling the expression of genes associated with peptide and protein utilization[23]. Intriguingly, a recent study by Miramon et al. found that Stp2 acts as the predominant regulator of genes involved in amino acid and peptide utilization, with Stp1 paying a minor role,

supported by the finding that gene expression patterns in *stp1Δ/Δ* mutant cells were nearly identical to those in wild-type cells[54]. In our study, we found that Stp2 is a major regulator of the C. *albicans* response to amino acids, primarily controlling the metabolism and transport of organic acids, carboxylic acids, and amino acids. This observation aligns with previously published transcriptomic data from C. *albicans* cells lacking *STP2*[54]. However, we detected approximately four times more differentially expressed genes than reported in that prior study, a discrepancy that may be attributed to differences in strain background. Importantly, our results revealed that Stp2 and Dal81 form a complex that facilitates promoter binding and transcriptional activation of downstream target genes. This cooperative interaction likely explains why deletion of either regulators results in a failure of pH alkalinization. Thus, our work defines an uncharacterized regulatory mode operating in amino acid uptake and metabolism of C. *albicans*, which requires the coordination of these two transcription factors. However, we have also observed distinct roles of Stp2 and Dal81 in pH alkalinization. For instance, the *dal81Δ/Δstp2Δ/Δ* double mutant exhibited an additive defect in alkalinization in both YNB + 1% CAA and artificial saliva. Additionally, in GM-BCP medium, the *dal81Δ/Δstp2Δ/Δ* double mutant showed a more pronounced reduction in ammonia release compared to the *dal81Δ/Δ* and *stp2Δ/Δ* single mutants. Moreover, Dal81 is specifically required for effective alkalinization in vaginal simulating fluid. These observations suggest that beyond amino acid metabolism, other Dal81- or Stp2-dependent metabolic endpoints may also contribute to the exacerbated alkalinization defects observed in the *dal81Δ/Δstp2Δ/Δ* double knockout mutant. Notably, the differences in alkalinization phenotypes among the mutants across various culture media may stem from variations in the metabolic utilization of specific components in each medium. Specifically, YNB + 1% CAA primarily contains casamino acids, while artificial saliva includes both casamino acids and mucin. In contrast, vaginal simulating fluid is mainly composed of glucose, lactic acid, and acetic acid. Finally, GM-BCP medium contains glycerol and yeast extract, providing a rich mix of nucleotides, proteins, amino acids, sugars, and trace elements. Our RNA-seq and ChIP-seq data reveal that, in addition to shared target genes involved in amino acid metabolism, which are coregulated by Dal81 and Stp2, there are other amino acid metabolism-related genes specifically regulated by either Dal81 or Stp2. Moreover, genes associated with amino acid transport are uniquely controlled by Stp2. These findings help explain the enhanced alkalinization defects observed in the *dal81Δ/Δstp2Δ/Δ* double knockout mutant in YNB + 1% CAA and artificial saliva. Given that Dal81 regulates glucose metabolism (Fig. S15) and is pivotal for the utilization of lactic acid and acetic acid (Fig. S16), together with the previous observation that glucose suppresses environmental alkalinization, we speculate that the impaired utilization of these acidic compounds and glucose in the *dal81Δ/Δ* mutant likely underlies the notable alkalinization defects observed in vaginal simulating fluid. The combined disruption of sugar metabolism (via Dal81) and amino/organic acid metabolism (via Dal81 and Stp2) in the double mutant probably impairs multiple alkalinization pathways. This could account for the more pronounced reduction in ammonia release observed in the *dal81Δ/Δstp2Δ/Δ* strain on solid GM-BCP medium (Fig. 1H). Given that Dal81 regulates glucose metabolism (Fig. S15) and is pivotal for the utilization of lactic acid and acetic acid (Fig. S16), together with the previous observation that glucose suppresses environmental alkalinization, we speculate that the impaired utilization of these acidic compounds and glucose in the *dal81Δ/Δ* mutant likely underlies the notable alkalinization defects observed in vaginal simulating fluid. The combined disruption of sugar metabolism (via Dal81) and amino/organic acid metabolism (via Dal81 and Stp2) in the double mutant probably impairs multiple alkalinization pathways. This could account for the more pronounced reduction in ammonia release observed in the *dal81Δ/Δstp2Δ/Δ* strain on solid GM-BCP medium (Fig. 1H).

Notably, a previous study reported that mutant cells lacking *STP2* exhibited a marked delay in alkalinization when grown in artificial saliva: the pH reached only 5.3 after 5 h but increased to ~8.2 by 24 h[15]. However, in our study, using the SN250 genetic background as the wild-type control, we observed that after 56 h, the pH of the *stp2Δ/Δ* culture reached only 6.3, compared to 7.8 in the wild-type strain (Fig. 2C). We speculate that the inconsistency in results may be due to differences in strain background. Our *stp2Δ/Δ* mutant was generated using *HIS* and *LEU* as selection markers, whereas the *STP2*-deficient strain in the previous report was constructed in the SC5314 background using the SAT-flipper method. These findings suggest that C. *albicans* may employ distinct alkalinzation strategies tailored to specific niches, with Dal81-dependent alkalinization for colonization in the vaginal tract, Stp2-dependent alkalinization to facilitate colonization in the oral cavity, and a combined Dal81- and Stp2-dependent mechanism to support colonization in the gut.

In addition to its crucial role in alkalinization, we found that Dal81 is important for regulating hyphae formation both in vitro and in vivo. Notably, our analysis identified an enrichment of hyphae-specific genes (*HGC1*, *UME6*, *ALS3*, *HWP1*, *ECE1*, and *NRG1*) that are abnormally expressed in the *dal81Δ/Δ* mutant (Fig. S18). Among these genes, two regulators are particularly noteworthy, as Ume6 is recognized as the primary regulator of hyphal extension[55], especially in response to neutral pH[56], and Hgc1 controls the expression of *UME6*[57]. Although neutral or alkaline pH is one of several cues that induce the yeast-to-hypha transition in C. *albicans*, and the *dal81Δ/Δ* mutant does not exhibit an inherent defect in responding to neutral pH, it remains unclear whether the hyphal formation defect observed in *dal81Δ/Δ* mutant within macrophage phagosomes stems from impaired alkalinization on this environment. The hyphal defect could also be attributed to the stressful conditions within macrophages or other host-derived factors, which warrants further investigation in future studies.

In its commensal state, C. *albicans* is typically characterized by the absence of filamentation, epithelial invasion, and host cell damage[58]. Interestingly, recent studies have argued that virulence traits, long recognized primarily for their role in inducing host diseases, such as hyphae formation and production of the hyphal-specific toxin candidalysin, may play an unexpected active role in facilitating the establishment of C. *albicans* as a gut commensal[59–61]. Specifically, the study from Bennett and colleagues[60] revealed that producing hyphae actually enhances the ability of C. *albicans* to thrive as a commensal when colonizing mice with an intact, normal gut microbiota. The strain capable of switching to filamentous forms gaines a competitive advantage over those locked in the yeast state and importantly, this competitive benefit relies on hyphal-specific toxin candidalysin, which functions to disrupt the metabolic activity, growth patterns, glucose uptake. Based on these studies, it is plausible that pathogenicity is not an intrinsic characteristic of fungi themselves, instead, it arises as a consequence of disrupted or imbalanced interactions between the microbe and its host. Given that Dal81 plays a significant role in regulating both extracellular pH alkalinization and hyphal formation in C. *albicans*, these two factors likely contribute to the critical involvement of Dal81 in promoting commensalism and virulence. Supporting this idea, the *dal81Δ/Δ* mutant displayed notable deficiencies in gut colonization and reduced virulence in a systemic infection model. In similar, the *stp2Δ/Δ* mutant also showed impairments in gut colonization and diminished virulence. Remarkably, the double knockout strain (*dal81Δ/Δstp2Δ/Δ*) exhibited a total loss of gut colonization and a complete absence of virulence, pointing to an additive effect of Dal81 and Stp2 in driving both commensalism and pathogenicity. This combined impact may stem not only from the distinct sets of genes regulated by Dal81 and Stp2. Even among the genes they co-regulate, the contribution of each transcription factor may vary based on the biological context. For example, the expression of *GDH2* was further downregulated in the *dal81Δ/Δstp2Δ/Δ* mutant compared to the

*dal81Δ/Δ* single mutant, while *PUT1* displayed significantly different expression patterns across the single and double mutant strains, observations that suggest context-dependent differences in regulatory dominance between the two factors. It is important to note that our current evidence supporting the role of both Dal81- and Stp2-mediated regulation of pH alkalinization and hyphal formation in gut colonization of *C. albicans* remains incomplete. Further investigations, such as exploring the link between pH alkalinization and candidalysin production, are necessary to strengthen these connections. On the other hand, we cannot exclude the possibility that additional factors, such as the gut-specific morphological variants, may also contribute to Dal81- and Stp2-mediated regulation of gut colonization. A previous study by Pande et al. [62] highlighted that the "White-to-GUT" switch promotes *C. albicans* commensalism in the gastrointestinal (GI) tract, with a specialized, gastrointestinally induced transition (GUT) cell type, regulated by the transcription factor Wor1, actively promoting the maintenance of a commensal lifestyle in this fungus. Remarkably, GUT cells exhibit a reprogrammed cellular metabolism that is adapted to optimize nutrient utilization in the GI environment[62,63]. Future studies will focus on investigating whether the regulatory networks controlled by Dal81 and Stp2 influence the White-to-GUT switch and GUT-associated metabolic adaptations.

Our RNA-seq and ChIP-seq analyses revealed that Dal81 and Stp2 regulate the metabolism of organic acids, oxoacids, and carboxylic acids. Given that *C. albicans* cells depend on alternative carbohydrate sources for nutrition within the host environment, the reduced intestinal colonization and survival observed in the mutant strains may reflect an underlying fitness impairment. To investigate this, we evaluated whether deleting *DAL81* and *STP2* affects the ability to utilize alternative carbon sources in vitro, by assessing the growth of wild-type, *dal81Δ/Δ*, *stp2Δ/Δ*, Dal81 comp., and *dal81Δ/Δstp2Δ/Δ* strains in YEP medium supplemented with various alternative carbohydrates. The *dal81Δ/Δ* mutant exhibited significant growth defects when cultured with acetate, lactic acid, and citrate, along with milder impairments when grown on oleic acid, ethanol, and N-acetylglucosamine (GlcNAc). Similarly, the *stp2Δ/Δ* strain displayed pronounced growth deficiencies on lactic acid and citrate. Notably, the *dal81Δ/Δstp2Δ/Δ* double mutant showed far more severe growth impairments on acetate, citrate and GlcNAc compared to either single-knockout mutant. These findings suggest that Dal81- and Stp2 -dependent metabolic adaptation to alternative carbon sources may contribute to both commensalism and virulence in *C. albicans*, potentially explaining why the *dal81Δ/Δstp2Δ/Δ* strain exhibited a complete loss of gut colonization and full attenuation of virulence. Consistent with this, both the *dal81Δ/Δ* and *stp2Δ/Δ* mutants continued to show reduced intestinal colonization compared to the wild-type strain even in immunosuppressed mice (Fig. 8C), indicating that their colonization defects are likely driven by impaired alkalinization, defective hyphal formation, and compromised metabolic adaptation, rather than altered interactions with the host immune system. Furthermore, in addition to amino acid metabolism known to raise extracellular pH, previous studies have shown that the metabolism of carboxylic acids (*e.g.*, pyruvate, α-ketoglutarate, or lactate) and the alternative carbon source GlcNAc can also raise extracellular pH[19,64]. As such, beyond their roles as nutrients, defects in the metabolism of specific alternative carbohydrates in the *dal81Δ/Δ* and *stp2Δ/Δ* mutants may lead to a failure to alkalinize the extracellular environment within the host. Based on these observations, we hypothesize that the colonization and virulence defects in the *dal81Δ/Δ* and *stp2Δ/Δ* mutants arise from an overall fitness impairment driven by two key factors: impaired extracellular alkalinization and an inability to effectively utilize host-derived nutrients.

Targeting transcription factors essential for fungal survival and proliferation within the host may represent a promising approach for advancing antifungal drug discovery. In our study, Dal81 emerged as a critical regulator in the commensalism and pathogenicity of *C.* *albicans*, particularly notable for its specificity to the fungal kingdom, which positions it as an appealing target for the development of selective therapeutics. Importantly, we uncovered a physical interaction between transcription factors Dal81 and Stp2. Further insights from protein modeling and structural analysis, conducted using AlphaFold 3 and ChimeraX, revealed that Dal81 and Stp2 adopt a "key-and-lock" structural configuration, with 21 potential interaction sites identified at their binding interface. Mutagenesis of these sites resulted in reduced binding of Dal81 and Stp2 to their downstream target genes, coupled with diminished expression of select target genes. Functionally, these mutations impaired the ability to alkalinize the extracellular environmental and weakened *C. albicans* virulence. While these findings support the role of the Dal81–Stp2 interaction in coordinating gene regulation linked to pH alkalinization and fungal virulence, additional studies are required to pinpoint the specific residues critical for this interaction. Given that the coordinated regulation mode between Dal81 and Stp2 is necessary for alkalinization and plays a key role in modulating commensalism and pathogenicity of *C. albicans*, screening for small molecules that disrupt their interaction holds promise as a potential therapeutic strategy for fungal infections. In summary, our study enhances understanding of how *C. albicans* adapts to the diverse microenvironments within host, laying groundwork for the development of improved therapeutics to combat this pressing global health challenge.

## Methods
### Ethics statement
The animal experiments were conducted in strict compliance with the Regulations for the Care and Use of Laboratory Animals issued by the Ministry of Science and Technology of the People's Republic of China. All efforts were made to minimize animal suffering. The animal experiments performed in this study were approved by Institutional Animal Care and Use Committee (IACUC) at the Shanghai Institute of Immunity and Infection, Chinese Academy of Sciences (Permit Number: A2020025).

### Animals
Female C57BL/6 mice (6–8 weeks old, 18–20 g) and female ICR mice (6–8 weeks old, 23–25 g) were purchased from Beijing Vital River Laboratory Animal Technology Company (Beijing, China). All mice were housed in a pathogen-free animal facility under controlled conditions (temperature: 21 °C; relative humidity: 50–70%; 12-hour light/dark cycle) and had ad libitum access to food and water. Mice were given free access to food and water throughout the study. All experimental procedures were conducted in accordance with the protocol approved by the Institutional Animal Care and Use Committee (IACUC) at Shanghai Institute of Immunity and Infection, Chinese Academy of Sciences, China. The study did not include sex-based analysis.

### Cell isolation and culture
All cell lines used in this study, including RAW264.7 (ATCC TIB-71) and J774A.1 (ATCC TIB-67), were obtained from the American Type Culture Collection (ATCC). Cells were cultured in Dulbecco's Modified Eagle's Medium (DMEM) supplemented with 10% fetal bovine serum (FBS) and 1% penicillin-streptomycin. For experiments, cells were seeded at a density of $2 \times 10^5$ cells per 35-mm dish or $5 \times 10^4$ cells per well in 96-well plates, and used 16 h post-seeding. All cell cultures were incubated at 37 °C in a 5% $CO_2$ atmosphere and maintained under humidified conditions.

Bone marrow-derived macrophages (BMDMs) were prepared as described previously[65]. Briefly, bone marrow cells were isolated from the femurs and tibias of C57BL/6 mice (6–8 weeks old) and subjected to hypotonic lysis to remove erythrocytes. Cells were cultured in RPMI-1640 medium supplemented with 10% fetal bovine serum (FBS), 30% conditioned medium from L929 cells (as a source of macrophage

colony-stimulating factor), and 1% penicillin–streptomycin. Non-adherent cells were removed, and adherent cells were passaged every 3 days. After 6 days of incubation, BMDMs were seeded at a density of $2 \times 10^5$ cells per 35-mm dish or $5 \times 10^4$ cells per well in 96-well plates, and used for infection assays.

## Media and growth conditions

*C. albicans* strains were routinely grown in YPD (2% peptone, 1% yeast extract, 2% glucose) or YNB + 1% CAA medium (1.7% yeast nitrogen base without amino acids, 0.5% ammonium sulfate, 1% Casamino Acids) at 30 °C. For liquid cultures, the strains were grown in YPD at 30 °C with agitation on a rotary shaker. Transformants were selected on YPD medium supplemented with nourseothricin (200 µg/ml). Alkalinization assays in liquid medium were primarily conducted using YNB + 1% CAA medium or medium 199. In certain experiments, pH alkalinization assays were performed in artificial saliva (initial pH 4.3) and vaginal simulating fluid (initial pH 4.5), using formulations adapted from reference[15]. The artificial saliva formulation (per liter) contained: 1.7 g yeast nitrogen base (YNB) without amino acids and ammonium sulfate, 5.0 g casamino acids, 2.5 g mucin, 1.1 g KCl, 0.5 g NaCl, 14 mg choline chloride, 10 mg sodium citrate, and 1.0 mg ascorbate. The vaginal simulating fluid formulation (per liter) included: 2 g glucose, 2 g lactic acid, 1 g acetic acid, 0.4 g urea, 0.25 g mucin, 0.16 g glycerol, 0.018 g bovine serum albumin, 3.51 g NaCl, 1.4 g KOH, and 0.222 g $CaCl_2$. For solid-phase alkalinization assays, *C. albicans* strains were grown on GM-BCP or GM-BCG (1% yeast extract, 30 mM $CaCl_2$, 3% glycerol, 0.01% bromocresol purple or bromocresol green, 2% agar) as previously described[16,66] The pH of the medium was adjusted with HCl or NaOH. Artificial saliva medium and vaginal simulating fluid medium were prepared as previously described[15].

## Strain and plasmid construction

The strains, primers, and plasmids used in this study are listed in Supplementary Data 7, Supplementary Data 8, and Supplementary Table 1, respectively. All *C. albicans* strains used in this study were derived from the reference strain SC5314. Corresponding gene deletion, complementation, and overexpression strains were generated as previously described[67].The *C. albicans* mutant library, consisting of 674 knockout mutant strains, was sourced from AD Johnson[24]. The *C. albicans* mutant strains with disrupted Dal81-Stp2 interaction sites were generated via homologous recombination. Point mutations were introduced into *dal81Δ/Δ* by substituting specific residues with alanine (Ala), either through synthesis of mutated *DAL81* genes or site-directed mutagenesis using specific primers. PCR-amplified products were treated with DpnI to digest the original plasmid template. The final constructs included a 950 bp region upstream of the *DAL81* start codon, the full-length *DAL81* open reading frame (ORF) containing either 21 or 2 alanine substitutions (or wild-type *DAL81*), and a C-terminal 13xMyc tag. These fragments were assembled into the backbone of the *DAL81* addback plasmid to generate plasmids bCB557, bCB558, and bCB556, respectively. The resulting plasmids were digested with *KpnI* and *XhoI* and transformed into the *dal81Δ/Δ* strain expressing HA-tagged Stp2. Transformants were selected on SC-Arg medium and verified by colony PCR. To construct the *DAL81* gene deletion strain in the *C. albicans* SC5314 background, we used a modified version of the *SAT1*-flipper method as previously described[26]. Briefly, a *SAT1−FLP* cassette flanked by 88 bp of homology immediately to the 5′ and 3′ untranslated regions of the *DAL81* open reading frame (ORF) was amplified from the pSFS2A plasmid[26]. The PCR product was introduced into the *C. albicans* SC5314 strain by chemical transformation, and transformants were selected on YPD medium containing nourseothricin. Correct cassette integration was verified by PCR. To remove the *SAT1−FLP* cassette, cells were cultured in YPM medium (1% yeast extract, 2% peptone, 2% maltose) to induce *FLP* recombinase gene expression and excise the marker. The same PCR product was

then used to delete the second allele of *DAL81*, resulting in the generation of the homozygous deletion strain YCB1391 (*dal81Δ/Δ* M1). Primers used for PCR amplification and validation were synthesized by Genewiz (Jinweizhi Biotechnology Co., Ltd.).

## In vitro growth assay

For growth curves in liquid medium, cells from overnight cultures were diluted to an initial $OD_{600}$ of 0.15 into the indicated medium. At indicated time intervals, the $OD_{600}$ was measured. Data are presented from three biological replicates and plotted in Graphpad Prism. For agar plate assays, overnight yeast cultures were washed and adjusted to an $OD_{600}$ of 1.0 in sterile water. Serial 10-fold dilutions were prepared, and 5 µl aliquots of each dilution were spotted onto appropriate agar plates. To assess growth on alternative carbon sources, *C. albicans* cells were cultivated at 37 °C in YEP medium (1% yeast extract, 2% Facto-Peptone) supplemented with the indicated alternative carbohydrates. These included carboxylic acids (acetate, lactic acid, α-ketoglutarate, pyruvic acid), fatty acids (oleic acid), and other substrates such as ethanol, glycerol, citrate, mannitol, sorbitol, and N-acetylglucosamine (GlcNAc). Serial 10-fold dilutions were prepared, and 5 µl aliquots from each dilution were spotted onto agar plates containing the respective carbon sources.

## In vitro screen for genes required for pH alkalinization

The *C. albicans* mutant library was screened to identify mutants defective in pH alkalinization, following a previously described method[16] with minor modifications. Briefly, all mutant strains and the isogenic wild-type reference strain SN250 were grown in YPD at 30 °C for 2 days in 96-deep-well microplates. Subsequently, 5 µl of each culture was transferred into fresh unbuffered medium 199 (initial pH 4.5) in 96-well cell culture microplates. Plates were incubated at 37 °C for 24 h, and alkalinization was assessed by colorimetric change. Candidate mutants displaying reduced or absent alkalinization were further validated using both medium 199 and YNB + 1% CAA medium under the same conditions as described above.

## Alkalinization assay

Alkalinization assays were performed as previously described[16] with minor modifications. For liquid culture assays, *C. albicans* strains were grown overnight in YPD, harvested, washed with water, and diluted to a starting $OD_{600}$ of 0.2 in the YNB + 1% CAA medium. Cultures were incubated at 37 °C with shaking, and medium pH was measured at the indicated time intervals. For solid medium assays, 12-well plates were prepared with 2 ml of GM-BCP or GM-BCG per well. Strains were grown overnight in YPD, harvested, washed, and resuspended in water to an $OD_{600}$ of 1.0. A 5 µl aliquots of this suspension was spotted into each well (one strain per well). Plates were incubated at 37 °C, and alkalinization was assessed visually by the appearance of a purple (GM-BCP) or blue (GM-BCG) halo surrounding the fungal colonies, indicating an increase in pH.

## Hyphal induction assay

Wild-type and mutant *C. albicans* cells were cultured overnight in YPD medium at 30 °C, harvested, washed with sterile water, and diluted to achieve a final $OD_{600}$ of 0.1 in either YNB + 1% CAA medium or medium 199. The pH of the YNB + 1% CAA medium was adjusted to 4.5 with HCl. Medium 199 was either buffered to pH 4.5 using 0.1 M Tris-0.1 M MOPS, or to pH 7.5 using 0.15 M HEPES. Cultures were incubated at 37 °C with shaking. Samples were collected at 0 h, 5 h or 8 h and subjected to microscopic observation to assess hyphal morphology.

## Ammonia release assay

The release of ammonia by *C. albicans* cells during alkalinization was measured based on a previously described method[27], with some modification. Briefly, cells were grown overnight in YPD medium,

harvested, and washed with distilled water. A 2 μl aliquot of cell suspension ($OD_{600}$ = 1.0) was spotted onto each well of a 96-well microplate containing solid GM-BCP medium adjusted to pH 4.0. A second 96-well plate, serving as an acid trap, was prepared with each reservoir containing 10% citric acid and placed directly beneath the inoculated plate. The plates were incubated at 37 °C for 72 h. Following incubation, samples from the citric acid trap were collected for ammonia quantification. To measure ammonia levels, 20 μl of 10-fold diluted samples was mixed with 80 μl of Nessler's reagent and incubated at room temperature for 30 minutes. Absorbance at 400 nm ($OD_{400}$) was measured using a Thermo Scientific plate reader. The release of ammonia trapped in the citric acid was quantified using an ammonium chloride standard curve.

### Protein extraction and immunoblotting

Total protein was extracted as previously described[67]. Lysates corresponding to 1.5 $OD_{600}$ of cells were subjected to SDS-PAGE and analyzed by immunoblotting. Primary antibodies included anti- Myc (9E10, MBL, Cat.#M192-3) for Myc-tagged proteins and anti-HA antibody (3F10, Roche, Cat.#11867423001) for HA-tagged proteins, both used at 1:10000 and 1:1000 dilutions, respectively. Secondary antibody (goat anti-mouse, Protein tech) was used at a 1:8000 dilution. An antibody against α-tubulin (Novus Biologicals, Cat.#NB100-1639) was used as a loading control at a 1:2000 dilution.

### Macrophage cytotoxicity assay and *C. albicans* killing assay

The cytotoxicity of *C. albicans* toward RAW264.7 macrophages, as well as its survival within RAW264.7, J774A.1, or bone marrow-derived macrophages (BMDMs), was evaluated as previously described[15]. Briefly, macrophages were seeded at $5 \times 10^4$ cells per well in 96-well plates and incubated overnight at 37 °C with 5% $CO_2$. Cells were then infected with log-phase *C. albicans* at a multiplicity of infection (MOI) of 3. Macrophage damage was assessed 5 h post-infection by measuring lactate dehydrogenase (LDH) activity in the culture supernatant using the Cytotoxicity Detection Kit (Promega, Cat.#G1780), following the manufacturer's instructions. LDH release by infected macrophages was calculated relative to the maximum LDH release from lysed macrophages to determine the percentage of macrophage cytotoxicity. For the *C. albicans* killing assay, $1 \times 10^4$ log-phase *C. albicans* cells were resuspended in fresh cell culture medium and added to wells with or without macrophages at an MOI of 0.4, followed by six serial 1:5 dilutions. After incubation for 48 h at 37 °C with 5% $CO_2$, microcolonies were counted under an inverted microscope. *C. albicans* survival was calculated as the ratio of colony numbers in co-cultures to those in macrophage-free controls.

### Quantitation of phagosomal acidification

To assess phagosomal acidification, mid-log phase *C. albicans* cells were labeled with 0.1 mg/ml pHrodo (ThermoFisher, Cat.#P36011) as previously described[27]. The labeled fungal cells were then incubated with RAW264.7 or J774 A.1 macrophages at a MOI of 2. To monitor phagosomal pH, images were captured at 10-minute intervals over a period of 1.5 h using a spinning disk confocal microscope system (Olympus SpinSR10) at 40× magnification. One hundred fungal cell containing phagolysosomes were counted and evaluated for acidification. Values represent mean ± SD of three different experiments. Image analysis was performed using ImageJ Fiji (v2.3) and Imaris (v9.5.1) software.

### Fungal total RNA isolation and RT-qPCR

*C. albicans* strains were grown overnight in YPD, harvested, washed with water, and diluted to a starting $OD_{600}$ of 0.1 in SD or YNB + 1% CAA medium (initial pH 4.5). Cultures were incubated at 37 °C with shaking for 6 h. Cells were then collected, and total RNA was extracted according to published methods[68]. One microgram of total RNA,

treated with DNase I was used for cDNA synthesis using a PrimeScript RT reagent Kit (TaKaRa, Cat.#RR047A). Quantitative PCR was performed using TB Green Premix Ex Taq II (TaKaRa, Cat.#RR820A) on an ABI7900HT Fast Real-Time PCR System. The expression level of each gene was quantified and normalized against the level of *ACT1*. Primers used for RT-qPCR were synthesized by Genewiz (Jinweizhi Biotechnology Co., Ltd.) and are listed in Supplementary Data 8.

### Subcellular localization assay

*C. albicans* cells were grown overnight in YPD medium at 30 °C, collected by centrifugation, washed with sterile water, and resuspended in YNB + 1% CAA medium (initial pH 4.5) to an $OD_{600}$ of 0.2. Cells were then grown to the log phase. Fixation, cell wall digestion, and antibody hybridization were performed as previously described[69], with the modification that an anti-HA antibody (3F10, Roche, C Cat.#11867423001) was used at a 1:150 dilution, and detected with a Cy2-conjugated secondary antibody at a 1:400 dilution. Images were acquired using a 100x oil objective on an inverted fluorescence microscope (Olympus FV-1200). All images were processed using ImageJ Fiji (v2.3) software.

### Co-immunoprecipitation (Co-IP) analysis

*C. albicans* cells expressing Myc-tagged Dal81 or HA-tagged Stp2 were grown in YNB + 1% CAA medium (initial pH 4.5) until reaching log phase. Cells were harvested by centrifugation at 3,000 rpm for 5 min, washed thrice with ice-cold sterile water, and resuspended in 700 μl of lysis buffer (50 mM Tris-HCl, pH 7.5, 50 mM NaCl, 1 mM EDTA, 0.5% NP40, 5 mM $MgCl_2$, 10% glycerol) supplemented with a protease inhibitor cocktail (Roche, Cat.#04693132001). Cells lysis was performed using a FastPrep-24 (MP Biomedicals, USA) in the presence of one-third volume of glass beads. Lysates were centrifuged at 14,000 rpm for 5 min at 4 °C. Protein concentration in the supernatants was determined using the Bradford assay. For immunoprecipitation, 5 mg of total protein was incubated with 50 μl of EZview™ Red anti-c-Myc affinity gel (Sigma, Cat.#E6654) or EZview™ Red anti-HA affinity Gel (Sigma, Cat.#E6779) overnight at 4 °C with gentle rotation. Beads were then washed five times with wash buffer (50 mM Tris-HCl, pH 7.5;150 mM NaCl; 1 mM EDTA; 0.5% NP40; 5 mM $MgCl_2$; 10% glycerol; and protease inhibitor cocktail). Bound proteins were eluted by adding 60 μl of 2x SDS-PAGE loading buffer followed by boiling for 5 min. Eluted proteins were subjected to SDS-PAGE and analyzed by western blotting.

### Yeast two-hybrid (Y2H) assay

The Y2H assay was performed to investigate the interaction between *C. albicans* Dal81 and Stp2. The full-length of coding sequence of Dal81 was amplified from *C. albicans* SC5314 genomic DNA and cloned into the XmaI-BamHI sites of the pGADT7 vector, generating the prey plasmid (bCB335). Similarly, the full-length coding sequence of Stp2 was cloned into the BamHI-PstI sites of the pGBKT7 vector to create the bait plasmid (bCB336). The resulting plasmids were independently transformed into the *Saccharomyces cerevisiae* strains Y187 (prey) and Y2HGold (bait), respectively. Mating was carried out, and diploids were selected and screened on SD quadruple-dropout (SD/-Leu/-Trp/-His/-Ade) agar supplemented with X-α-Gal to assess protein-protein interactions. Blue colony formation indicated a positive interaction. The interaction between pGBKT7-53 and pGADT7-T served as a positive control.

### Protein model prediction

The *C. albicans* Dal81 and Stp2 protein sequences were obtained from Uniprot. AlphaFold 3[70] was used to model the Dal81-Stp2 protein complex. The most accurate model was selected base on AlphaFold's predicted Local Distance Difference Test (pLDDT) scores, which assessed the predicted structure's reliability at the amino acid level. The finalized structures were rendered using UCSF ChimeraX[30].

## RNA-Seq and data analysis

The wild-type (SN250 or SC5314), *dal81Δ/Δ* and *stp2Δ/Δ* strains were cultured overnight in YPD at 30 °C. Cells were harvested by centrifugation, washed with water, and resuspended at an $OD_{600}$ of 0.2 in either YNB + 1% CAA (initial pH of 4.5) medium or SD medium. Cultures were grown to the log phase, and total RNA was extracted as described above. RNA integrity was determined using an Agilent 2100 Bioanalyzer (Agilent Technologies). Triplicate RNA-seq libraries for each strain and condition were prepared and sequenced on an Illumina HiSeq 2000 genome analyzer (Novogene, Beijing, China). Raw reads were quality-checked using FastQC (https://github.com/s-andrews/FastQC) and MultiQC v1.13[71]. Adapters and low-quality bases were trimmed using Trimmomatic[72]. Clean reads were aligned to the *C. albicans* SC5314 genome (assembly 22) using STAR v2.7.6a[73]. The raw count tables were generated by featureCounts v2.0.1[74]. Differential expression analysis was performed in DESeq2 v1.20[75], with significance defined as *P* value (adj. *p* -value) <0.05 and |log2 fold change| >1. Heatmaps were plotted using the R package ComplexHeatmap v2.20.0[76], and Gene Ontology (GO) enrichment analysis were conducted with the clusterProfiler package[77].

## ChIP- Seq and data analysis

Strains were grown overnight in YPD at 30 °C, harvested by centrifugation and washed with water. Cells were then resuspended at an $OD_{600}$ of 0.2 in 100 ml of YNB + 1% CAA medium (initial pH 4.5) and grown to log phase. For chromatin immunoprecipitation, cells were cross-linked with 1% formaldehyde for 15 min with gentle agitation, and the reaction was quenched by the addition of glycine. Cells were pelleted, washed three times with ice-cold PBS, and stored at −80 °C in 2 ml screw-cap tubes. For lysis, frozen pellets were resuspended in 700 μl of lysis buffer (50 mM HEPES/KOH pH 7.5; 140 mM NaCl; 1 mM EDTA; 1% Triton X-100; 0.1% SDS; 0.1% Na-deoxycholate; protease inhibitor cocktail [EDTA-free, Roche]). and cells were mechanically disrupted using 500 μl of 0.5 mm $ZrO_2$ beads in a FastPrep-24 shaker (MP Biomedicals) for three cycles of 60 s at 6.0 m/s, with 5-minute cooling intervals on ice. After lysis, samples were rotated at 4 °C for 30 min, and the $ZrO_2$ beads were removed by puncturing the tube bottom and centrifuging for 3 min at 1000 rpm to collect the lysates. Chromatin was then sheared via ultrasonication to yield DNA fragments of approximately 100–500 bp. The sheared lysates were clarified by centrifugation at 10,000 g for 15 min at 4 °C, and the supernatants containing the solubilized chromatin were collected. A 50 μl aliquot of each lysate was reserved as Input DNA for downstream analysis.

To facilitate immunoprecipitation, 30 μl of magnetic protein A/G beads (Thermo, Cat.#88802) were washed twice with PBST and resuspended in 400 μl of the same buffer. Subsequently, either 5 μl of anti-Myc antibody (Abmart, Cat.#M20002) or 20 μl of anti-HA antibody (Roche, Cat.#11867423001) was added to the bead suspension and incubated for 6 h at 4 °C with rotation. Following antibody binding, the buffer was replaced with 900 μl of RIPA buffer, and 500 μl of solubilized chromatin was added. The mixture was incubated overnight at 4 °C with gentle agitation. The beads were then sequentially washed with 1 ml of each of the following buffers: buffer 1 (1xTE, 1% Triton X-100, 0.1% SDS, 0.1% Na-deoxycholate), buffer 2 (1xTE, 1% Triton X-100, 0.1% SDS; 0.1% Na-deoxycholate, 300 mM NaCl), buffer 3 (1xTE, 250 mM LiCl, 0.5% NP40, 0.5% Na-deoxycholate), buffer 4 (1xTE, 0.2% Triton X-100), and finally 1xTE buffer. To obtain chromatin-bound DNA, beads and Input DNA samples were incubated in 100 μl TE containing 4 μl of 5 M NaCl and 4 μl of 10% SDS for 4 h at 65 °C. RNA was removed by treatment with RNase A (100 μg/ml) at 37 °C for 1 h, followed by protein digestion using a mixture of 0.01 M EDTA, 0.04 M Tris-HCl (pH 6.5), and 100 μg/ml proteinase K at 55 °C for 3 h. The supernatant was transferred into a new tube, and the beads were washed once with 100 μl of TE. The wash was combined with the previous supernatant, and DNA was purified using the GeneJEET PCR purification kit (Thermo Scientific) according to the manufacturer's instructions.

The eluted DNA was used to construct the sequencing libraries using NEBNext® Ultra™ II DNA Library Prep Kit (Novogene, Beijing, China). Paired-end sequencing (2 ×150 bp) was performed on an Illumina HiSeq 2500 platform (Novogene, Beijing, China). Raw reads were trimmed with Trimmomatic v0.39[72], and quality was assessed using FastQC before and after adaptor trimming. Clean reads were aligned to the *C. albicans* SC5314 genome (assembly 22) using Bowtie2 v1.3.1[78] with default parameters. Resulting SAM files were converted to BAM format, and non-uniquely mapped reads were filtered out. Duplicates were marked and removed using Sambamba v0.7.1[79]. Samtools v1.6[80] was used to merge the BAM files from biological replicates and apply additional filtering. Normalized coverage files (bigWig) were generated using Deeptools 3.5.1[81] bamCoverage (RPKM normalization, duplicates ignored). ChIP-seq signal from replicates was combined, and peak calling was performed using Macs2[82].

## ChIP- qPCR assays

Precipitated and input DNA samples were prepared as described above and used as templates for ChIP-qPCR analysis using TB Green Premix Ex Taq II (TaKaRa, Cat.#RR820A) following the manufacturer's instructions. Primer sequences are listed in Table S2.

## In vivo animal assays

To assess the role of Dal81 and Stp2 in *C. albicans* commensalism within the gastrointestinal (GI) tract, we employed a previously described mouse colonization model[34,83]. Briefly, female C57BL/6 mice (6–8 weeks old) were administered penicillin (1.5 mg/ml) and streptomycin (2 mg/ml) in their drinking water starting three days prior to fungal inoculation. Mice were orally gavaged with 1 ×10⁸ CFUs of wild-type, *dal81Δ/Δ*, *stp2Δ/Δ*, *dal81Δ/Δstp2Δ/Δ*, or Dal81comp. strains of *C. albicans*. Fecal pellets were collected at various time points post-inoculation, homogenized in PBS, serially diluted, and plated on Sabouraud agar containing 50 μg/ml ampicillin and 15 μg/ml gentamycin. Plates were incubated at 30 °C, and fungal colonization was quantified by enumerating colony-forming units (CFUs). Statistical significance was determined using one-way or two-way ANOVA followed by Tukey's test. When required, female ICR mice (6–8 weeks old, average weight 23–25 g) were immunosuppressed with intraperitoneal (I.P.) injections of cyclophosphamide at a dose of 150 mg/kg, starting four days prior to fungal infection. Immunosuppression was maintained by administering two additional doses of cyclophosphamide (150 mg/kg, I.P.) on days 1 and 5 post-infection. To assess intestinal colonization by *C. albicans*, immunosuppressed mice were orally gavaged with a 1:1 mixture of the indicated strains (1 × 10⁸ CFUs total). Fecal pellets were collected at defined time points post-infection, and recovery and quantification of *C. albicans* cells were performed as previously described[24]. Statistical significance was determined using two-way ANOVA followed by Tukey's test. The virulence of the WT, *dal81Δ/Δ*, *stp2Δ/Δ dal81Δ/Δstp2Δ/Δ*, Dal81comp., *DAL81 /dal81Δ* or *DAL81m21 /dal81Δ* was assayed using an established murine systemic infection model as previously described[35,67]. Briefly, groups of female C57BL/6 mice (6-8 weeks old) were intravenously inoculated with 2 ×10⁵ CFUs of the respective *C. albicans* strains. Mice were monitored daily for clinical indicators including weight loss, disease symptoms, and survival. Survival data were analyzed using the Kaplan-Meier method, and statistical significance was determined by the log-rank (Mantel-Cox) test using GraphPad Prism software.

## Statistical analysis

All statistical analyses were performed using GraphPad Prism versions 8.2.1 and 10.3.1. Data are presented as mean ± standard deviation (SD). Specific statistical tests and sample sizes are detailed in the respective

figure legends. Statistical significance was defined as follows: *P < 0.05; **P < 0.01; ***P < 0.001; ****P < 0.0001; ns, non-significant.

## Reporting summary

Further information on research design is available in the Nature Portfolio Reporting Summary linked to this article.

## Data availability

All data necessary to evaluate the conclusions of this study are provided within the main text and/or the Supplementary Materials. The Dal81 ChIP-seq, Stp2 ChIP-seq, and RNA-seq datasets have been deposited in the National Center for Biotechnology Information Sequence Read Archive (NCBI SRA) under accession number PRJNA1190138. Source Data are available with this paper. Source data are provided with this paper.

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

## Acknowledgements

We are grateful to Suzanne Noble at UCSF for the gift of the *C. alblicans* homozygous gene deletion library. We also would like to thank members of the Changbin Chen lab for helpful discussions and advice. Special thanks to Miss Grace Morrissey for helping us proofread the manuscript. C.C. is supported by grants from the MOST Key R&D Program (2020YFA0907200; 2022YFC2303200); National Natural Science Foundation of China (32170195; 32200161); The Shanghai Science and Technology Innovation Action Plan 2023 "Basic Research Project" (23JC1404200); The Shanghai "Belt and Road" Joint Laboratory Project (22490750200); The Foundation of State Key Laboratory of Pathogen and Biosecurity (SKLPBS2236). X.H. is supported by grants from the MOST Key R&D Program (2022YFC2303504); National Natural Science Foundation of China (32070146); and Natural Science Foundation of Shanghai (20ZR1463800).

## Author contributions

X.H. and C.C. planned and designed the study; X.H., H.L. and C.C. planned and designed the revision study; X.H. conducted all of the experiments with the help from L.W., Y.Z., L.Z., S.L., K.L., W.X., Z.J., Y.P.Z., J.D., H.H., X.W., Y.W., T.J. and W.X.; G.C. performed all of the bioinformatics analysis. X.C. participated in the RNA-seq and ChIP-seq analysis; C.C. and X.H. wrote the manuscript; W.S. edited the English language. M.M. helped refine the working model. X.H., Z.H., Q.G., H.L. and C.C. discussed the experiments and results.

## Competing interests

The authors declare no competing interests.

## Additional information

¹Joint Laboratory for Biomedical Research and Pharmaceutical Innovation, The Unit of Pathogenic Fungal Infection & Host Immunity, Shanghai Institute of Immunity and Infection, Chinese Academy of Sciences, Shanghai 200031, China. ²Department of Respiratory and Critical Care Medicine, The First Affiliated Hospital of Guangxi Medical University, Nanning, Guangxi, China. ³University of Chinese Academy of Sciences, Beijing, China. ⁴USJ-Kong Hon Academy for Cellular Nutrition and Health, University of Saint Joseph, Macau, China. ⁵Reproductive Medicine Center, The Fourth Hospital of Shijiazhuang (Affiliated Obstetrics and Gynecology Hospital of Hebei Medical University), Shijiazhuang, Hebei, China. ⁶Nanjing Advanced Academy of Life and Health, Nanjing, China. ⁷Organ Transplantation Clinical Medical Center of Xiamen University, Department of General Surgery, Xiang'an Hospital of Xiamen University, School of Medicine, Xiamen University, Xiamen, China. ⁸These authors contributed equally: Xinhua Huang, Guangsheng Chen, Lei Wu, Yun Zou. ✉e-mail: xhhuang@siii.cas.cn; lihao6656@163.com; cbchen@ips.ac.cn

