## [Transparent Peer Review file · Nature Communications]

Coordinated regulation of pH alkalinization by two transcription factors promotes fungal commensalism and pathogenicity

Corresponding Author: Dr Changbin Chen

Version 0:

Reviewer comments:

Reviewer #1

(Remarks to the Author)

Summary

The objective of this manuscript by Huang, et al. is to identify the role of the transcription factor Dal81p in contributing to extracellular alkalinization in the fungal pathogen *Candida albicans* (Ca). The authors conducted a screen of a Ca deletion library and identified the known regulator of extracellular alkalinization, Stp2, along with the transcription factor Dal81 which was not previously associated with this phenotype. Using a series of approaches, including measuring culture pH, ammonia release, filamentation, and macrophage killing, the authors show that the *dal81* mutant is unable to alkalinize the extracellular environment (including in the phagosome), forms fewer hyphae in acidified phagosomal compartments and in unbuffered medium, and has poorer survival during macrophage challenge. These phenotypes were reversed when an ectopic copy of DAL81 was re-introduced in the mutant. The *dal81* mutant shows no obvious growth defects otherwise, further potentially linking observed phenotypes to an inability to modulate the pH. The authors then constructed a *dal81/stp2* double mutant to show that it has more severe alkalinization defects in standard culture and biological fluid simulants. Interestingly, an epistasis study revealed that both STP2 and DAL1 are required for maximal extracellular alkalinization, as overexpression of either in the absence of the reciprocal transcription factor could not rescue the observed alkalinization defect of the single mutants. Moreover, deletion of STP2 results in a compensatory increase in DAL81 expression, but not vice versa. Yeast two hybrid, fluorescence co-localization, and reciprocal co-immunoprecipitation experiments revealed that Stp2 and Dal81 likely physically interact, which may explain the epistasis results. Transcriptional profiling studies revealed that both transcription factors regulate remarkably similar target genes, including those involved in oxoacid, organic acid, carboxylic acid, and amino acid metabolism. Follow-up Chip-Seq studies revealed that both transcription factors directly bind approximately 30% of the shared target genes, while each also directly regulates diverse genes. Lastly, the authors show that like STP2, DAL81 is required for maximal colonization of the murine GI tract and contributes to virulence during disseminated candidiasis.

Collectively, this is a very nicely performed series of experiments to demonstrate that while Stp2 is a master regulator of alkalinization, Dal81 plays both redundant and unique roles in contributing to this phenotype, which is an important virulence determinant. The plethora of molecular and omics techniques are leveraged in a meaningful and focused way, to nicely highlight both the physical interaction of Dal81 and Stp2 and their coordinated regulatory functions. I have several comments and suggestions that should be considered to further improve the manuscript and strengthen the findings. Please see those comments below.

Major comments

1. Fig S2B: While the number of colonies between WT and *stp2*, *dal81* mutants is similar, the size of those colonies are clearly different. The mutants are comparatively smaller, suggesting reduced growth or loss of fitness. The authors should comment on this in the Results.
2. While the ammonia release observed in the *dal81* mutant in Fig. 1E may be statistically significant compared to WT, I question its biological meaningfulness. If the ammonia release is not driving failed alkalinization, then what other Dal81-dependent metabolic endpoints may be responsible? This should be noted in the Results and followed up in the Discussion.

3. Please similarly address point #2 in line 264.
4. In Figure 1H, what is the comparator to determine percentage of survival? This should be explicitly described in the legend or the Methods.
5. The colony morphologies in Fig. 2F are quite distinct in WT, *dal81*, *stp2*, and double mutants. The authors should note this and provide an explanation as to why.
6. The epistasis data presented in Fig. 3 is very nice. Does *Dal1p* bind to the *STP2* promoter and observed by Chip-Seq? Might this explain the observations here?
7. The western blot data in Fig. 3B and especially 3D should be quantified by densitometry and shown.
8. Replicate, unprocessed blots from Figs. 3B,D and 4C,D should be provided in the supplementary materials.
9. The co-localization data presented in Fig. 4 is not abundantly clear. While I can certainly observe punctate staining in the nuclei of WT and revertant, I also see punctate staining that does not overlap with nuclear staining. The authors should provide a quantitative readout of comparative colocalization amongst the strains.
10. It is peculiar that the authors did not use the *stp2/dal81* double mutant in the in vivo colonization and virulence studies. It would be informative to know whether a more severe colonization defect could be observed at the day 3 timepoint. It's possible that in vivo *Stp2* and *Dal81* partially compensate one another.
11. Can the authors speculate whether the colonization and virulence defects observed by the loss of *Stp2* and/or *Dal81* are due to alkalization defects or due to their inability to use alternative carbohydrate sources available in the host? As these transcription factors control the metabolism of several organic acids, it is conceivable that loss of "virulence" is instead due to an overall fitness defect.
12. The authors may consider performing an experiment in immunosuppressed mice. If reduced colonization is still observed, then it is likely that loss of *Stp2* or *Dal81* is independent of immune interaction and phagosomal escape and instead due to an inherent fitness cost. At minimum, this needs to be addressed in the Discussion.
13. I could not locate the Chip-Seq data in the supplementary data set. This should be provided.

Minor comments

1. Line 60: The term "normal abiotic factors" is inherently context-dependent. What is normal? Perhaps better to state "deviation from homeostatic conditions".
2. Line 86: *C. albicans* is not the most common human fungal pathogen. The dermatophytes (e.g., *Trichophyton*) are more common infecting microbes than the *Candida* species.
3. Line 162: Please define MM in the text at least once, in the figure legend, or both.
4. Line 171: This sentence reads awkwardly. Please revise.
5. Line 209. Reference #28 indicates that hyphal growth initiates inside of acidic phagosomes and expanding hyphae rupture the phagosome driving proton leakage to result in alkalization. The authors should be careful in statements (e.g., line 213 but throughout) whether alkalization drives hyphal growth or is simply a consequence of the yeast-to-hypha transition. There should be more nuanced interpretation and could provide material for the Discussion.
6. Line 415-417: Details regarding Chip-Seq should be moved to the Methods.
7. Line 656: More detail regarding the macrophage cytotoxicity and fungal killing assays is required.
8. Line X: A more nuanced discussion of why extracellular alkalization phenotypes differ so drastically in YNB+1% CAA, artificial saliva, and vaginal simulant fluid should be discussed more thoughtfully. What are the compositions of those media with respect to amino, carboxylic, etc. acids?
9. Fig. 2: I find the labeling of this figure unnecessarily confusing. Please remove the 1,2,3,4,5 labels and instead label with the appropriate strain name in each panel.
10. Fig. 7: The authors should continue using the same color schema for the strains here as they did throughout the remainder of the study.

Reviewer #2

(Remarks to the Author)

Adaptation to changing environment is key to the survival of microorganisms. *Candida albicans*, a commensal and opportunistic pathogen of humans, is exposed to environmental variations, including changes in pH across the different niches it colonizes. To circumvent low pH, eg upon phagocytosis, *C. albicans* can trigger environmental alkalization. A large body of research has shown that this process is under the control of the master regulator *Stp2*. In this report, Huang et al. reveal the involvement of another transcription factor, namely *Dal81*, in the control of environmental alkalization and propose that *Stp2* and *Dal81* physically interact to achieve this control. In order to support this conclusion, the authors provide evidence that 1) *C. albicans stp2* and *dal81* knock-out mutants have similar phenotypes relative to environmental

alkalinization and that simultaneous inactivation of STP2 and DAL81 is additive/synergistic; 2) loss of DAL81 results in reduced levels of Stp2 while loss of STP2 does not have any effect of Dal81 protein levels; 3) Dal81 and Stp2 physically interact as evidenced by 2-hybrid and co-immunoprecipitation; 4) Dal81 and Stp2 share transcriptional targets using transcript profiling and CHIP-Seq and respectively influence the binding of the other transcription factor at some of these targets. Furthermore, they show that STP2 and DAL81 are similarly necessary for *C. albicans* colonization of the mouse GI tract and for *C. albicans* virulence in a mouse model of disseminated infection.

This is an important manuscript that is likely to attract interest from the molecular mycology community and beyond, as they change the current paradigm of pH adaptation in fungal pathogens. The experiments are carefully conducted and presented, comprehensive, using cutting-edge approaches. The manuscript is overall well-written even though it would benefit from proof reading by a native English speaker. Yet there are several aspects that deserve clarification.

Major comments:

- 1) The authors use HA- or myc-tagged versions of Spt2 and Dal81 to assess chromatin binding and regulation of gene expression by the two TFs. They do not provide any evidence that these modified forms behave like the wild-type forms. Data should be provided that address this issue as the functionality of the tagged TFs will influence the results of the experiments shown in several figures.
- 2) The authors state in the abstract "our results demonstrate that Dal81 physically interacts with Stp2 to co-regulate the expression of downstream target genes related to metabolism of organic acid, oxoacid, carboxylic acid and amino acid. This coordinated regulation mode is indispensable for the alkalization process and plays a pivotal role in modulating commensalism and pathogenicity of *C. albicans*." While I agree with the authors that their data demonstrate that Dal81 physically interacts with Stp2 AND co-regulate the expression of downstream target genes, I don't think that they provide direct evidence that the Stp2-Dal81 interaction is necessary to co-regulate the expression of downstream target genes. Indeed, what is shown is that the absence of one of the two partners impairs 1) binding to target promoters of the other partner and 2) gene expression of the target genes. That gene expression of the target genes is maintained to some extent when one of the two partners is present and further impaired when the two partners are lacking (Fig 6G-I) indicates that the interaction is somewhat dispensable for the expression of downstream target genes. Furthermore, the reduction in Stp2 binding to target promoters when Dal81 is missing could also result from reduced levels of Stp2 as shown in Fig. 3B. The additivity effect of knocking out the two genes for a number of investigated phenotypes is also indicative that each transcription factor can function in the absence of the other, thus questioning the conclusion that "This coordinated regulation mode is indispensable for the alkalization process". Finally, as the authors have only evaluated the impact of knocking out STP2 or DAL81 on commensalism and virulence, the conclusion that "this coordinated regulation mode plays a pivotal role in modulating commensalism and pathogenicity of *C. albicans*" is not well supported. Ideally, one would like to have evidence that the two factors interact at the chromatin. One would also like to have data in a context whereby interaction is abolished while the two proteins remain present. Have the authors used alphaFold-related tools to explore the interaction between Stp2 and Dal81? Could this identify key residues for the interaction? What would be the consequence on chromatin binding and gene expression of affecting such residues? While I acknowledge these experiments might be beyond the scope of this study, it seems important that the conclusions made by the authors are softened and that the discussion addresses these questions.
- 3) In some places, the authors state that the *stp2* and *dal81* knock out mutants have similar behaviors. These conclusions should be revisited. For instance, they state l.154-155 that "Similar to the *stp2* Δ/Δ mutant, the *dal81* Δ/Δ mutant failed to release significant amounts of ammonia (Fig. 1E)" while the figure shows a slight (but significant) reduction for the *dal81* mutant compared to wt and a more drastic reduction for the *stp2* mutant. Thus I don't think one can state that the *stp2* and *dal81* have similar behavior. On another instance, the authors state l. 257-260 that "Compared to that of the two single mutants, the alkalization defects were more pronounced in *C. albicans* deficient for both STP2 and DAL81 when cells were grown on YNB+1% CAA at initial pH 4.5 (Fig. 2B), artificial saliva (AS) (Fig. 2C), and vaginal simulating fluid (VSF) (Fig. 2D)." while Fig. 2C shows that the *dal81* mutant behaves almost like wild-type and the *stp2* mutant behaves almost like the double mutant. Fig. 2D shows that the *stp2* mutant behaves like wild-type while the *dal81* mutant behaves like the double mutant. These are important observations that could be discussed as they inform the conclusions of the authors (see major comment above) and provide insights on one of the questions raised in the introduction (see l.113-118). Overall, the conclusions should be fine-tuned to better reflect the data and possibly put in a better perspective. For instance, the observation that the *stp2* mutant has an alkalization defect in artificial saliva medium (Fig. 2C) is inconsistent with previously published results in Vylkova and Lorenz (PMID: 24626429).
- 4) The discussion could be strengthened according to some of the comments above but also by discussing how consistent the authors' description of the Spt2 regulatory network is with those presented in previous reports. Also, the authors may want to discuss more thoroughly the importance of Dal81 for alkalization, hyphae formation and virulence.

Minor comments

- English, while overall good, should be improved. There are a number of sentences that do not work (l.225-227, It is the alkalization defect, instead of a general morphological defect, contributing to the inability of mutant cells lacking DAL81 to switch from yeast to hyphal forms; l.298-300, It has been reported that the truncated version of Stp2 lacks the amino terminal nuclear exclusion domain (the first 99 amino acids) and micks the proteolytically processed form(21); l.438-434, For example, GDH2 encoding the putative NAD-specific glutamate dehydrogenase and PUT1 encoding the putative proline oxidase, both of which are related to metabolism of organic acid, oxoacid, carboxylic acid and amino acid (fig. S9B) and previously reported to be involved in pH alkalization(26), as well as PCK1 encoding the phosphoenolpyruvate carboxykinase gene, which plays a role in the small molecule biosynthetic process and may be involved in pH alkalization(31).)
- The references in the text lack a space for separation from the preceding word.
- l.1: the authors have used alkalization and alkalinization; they should decide for one or the other; I would recommend alkalinization
- l.36: I am not sure that screening 674 knock-out mutants can be qualified high throughput

- l.72: "fungal pathogens were found to be able to actively modify, either acidify or alkaline, the pH"; pH cannot be acidified or alkalized, it is decreased or raised.
- l. 87: Infection at mucosal surfaces are generally referred to as superficial infections, invasive infection corresponding to those where there is invasion of tissue.
- l. 92-93: I don't think there are cases of morbidity or mortality
- Inconsistent use of starting pH of YNB-CAA medium: line 143: pH 4.0, line 216: pH 4.5, line 273: pH 4.3.
- Results from Fig. 1C are described significant (l.146) but it is showing only 1 out of 3 replicates. Instead, the authors should show mean + SD and statistical significance of all three replicates. This panel would also benefit to have data for the *stp2* knock-out mutant.
- l.151-154 is not exact. Fig. S2 clearly shows that the *dal81* mutant has decreased growth compared to the WT.
- l. 212-213: That *Dal81* regulates pH alkalization in a way similar to that of *Stp2* is not demonstrated at this stage of the ms.
- l.225 : revise sentence according to the experiment shown in Fig S4B: "Strains were grown overnight in YPD, then transferred to medium 199 buffered at pH 4.5 or 7.5."
- l.248-249 and Fig. 2A: please mention at which time point the RNA for RT-qPCR were obtained. This is also lacking in the methods part.
- Fig. 2A: the red bar for *STP2* should be cut as for *DAL81*
- Fig. 2E: This is unlikely YNB+CAA medium. It might be liquid GM-BCP medium. Should be clarified.
- l. 262-264: "that the mutant lacking both *STP2* and *DAL81* displayed the strongest defects in pH alkalization on the solid GM-BCP medium (Fig. 2F)" is "due to its lower release rate of volatile ammonia (Fig. 1E)." has not been demonstrated; it is likely due.
- Fig. 3A-C: same comment as before: mention at which time point the RNA for RT-qPCR and proteins for western blot were obtained.
- l. 363 and l. 369: what do the author mean by in vivo co-IP.
- Fig. 4CD: in panel C, indicate IP-HA; in panel D, indicate IP-myc; Panel C would benefit from a control with untagged *Stp2*-HA while panel D would benefit from a control with untagged *Dal81*-myc; this is especially important as a protein is detected with the myc antibody upon HA IP when *Dal81* is not tagged. Why only one band is immunoprecipitated for *Stp2*-HA ?
- Fig. 5B: Instead of showing these Venn diagrams, the authors may consider showing the one from Fig.S8A together with the GO enrichment analysis of Fig.S8B. It would highlight that most of the identified genes are regulated by both *Dal81* and *Stp2*.
- Fig. 5C: Here the authors compare the *dal81* mutant and *stp2* mutant to SN250. In the figure legend they mention that the mutants were compared to the WT (SC5314). It is confusing and should be clarified.
- Fig. 6B reads "Sef1 binding Motif". It should read "Stp2-binding Motif".
- Fig. 6H should read PCK1/ACT1
- l.439: *Stp2*-Myc should read *Stp2*-HA
- Fig. 7ABC: the 3 panels could be merged in a single panel.
- Fig. S1B: Why SC-Medium? This wasn't use elsewhere. Why not performing a spot dilution assay on YNB-CAA?
- Fig. S2: The figure legend title is "...*dal81*d mutant cells have no obvious growth defect." This is not reflecting data in Fig.S2B and C. Fig. S2C shows only one replicate. Showing the mean + SD of all 3 replicates would be better.
- Fig. S5B: In the figure legend the authors mention that they used YPD medium for the hyphae formation assay. Why not buffered Medium 199 like in Fig. S4? Why using different media?

Reviewer #3

(Remarks to the Author)

The manuscript by Huang et al., titled "Coordinated Regulation of pH Alkalization by Two Transcription Factors Promotes Fungal Commensalism and Pathogenicity," identifies *Dal81* as a key co-regulator involved in amino acid utilization, environmental pH neutralization, colonization, and pathogenicity in *Candida albicans*. The authors clearly demonstrate the physical interaction between *Stp2* and *Dal81*, which enhances our understanding of amino acid catabolism and its role in evading the immune response by inhibiting phagosomal maturation. The results are presented logically and clearly, with appropriate controls in place. The conclusions drawn are well supported by the experimental evidence, making this a highly relevant study that highlights its own significance.

However, since *Dal81* is an uncharacterized transcription factor, a more in-depth discussion regarding its role is necessary, particularly concerning the phenotypes observed on simulated media (saliva vs. vaginal fluid; Fig. 2) and the differential gene regulation between *Stp2* and *Dal81* (Fig. 6).

Specific comments:

The background strain SN250 is commonly used for its convenience in generating deletion mutants, allowing for the selection of transformants through auxotrophy complementation. However, it is important to note that amino acid utilization and auxotrophic markers can have confounding effects on the results. Therefore, the use of prototrophic strains is highly recommended for these studies. We are not suggesting the creation of a new set of strains, but we encourage consideration of this point for future research.

In the library screen, two hits were identified: *DAL81* and *GRR1*. Were there any other notable hits to mention, even if the phenotypes are mild, partial, or delayed? We believe this would enhance the significance of this work.

The experiments involving macrophages are quite intriguing. Did the authors assess phagosome maturation at later time points? Investigating the temporal dynamics of phagosome activation would enhance the relevance of this study. While RAW264.7 macrophages have been utilized in *C. albicans* survival assays, we recommend considering other cell lines, such as J774A.1 or THP-1-derived macrophages, as well as primary murine macrophages (bone marrow-derived). The RAW264.7 macrophages lack a crucial component of the NLRP3 inflammasome complex, which plays an essential role in the interaction between *C. albicans* and these immune cells. Therefore, we strongly suggest verifying the key results using a more suitable cell line.

The phenotypes of the *stp2* and *dal81* mutants display an additive effect in amino acid metabolism; however, they appear to have different roles in AS (*stp2*) and in VSF (*dal81*). A more detailed discussion is necessary to explore these differences. Do the authors have any transcriptional data that could enhance the discussion of these results? Additionally, can the authors speculate on the implications of these differences?

Demonstrating the nuclear localization of Stp2 is quite challenging, particularly with immunohistochemistry (IHC). Others have had slightly better success using fluorescently tagged versions of this protein. The authors should consider adopting a similar approach to convincingly demonstrate that the deletion of DAL81 does not impact the nuclear translocation of Stp2. Alternatively, the processing of Stp2 could serve as evidence for this claim.

The heat map presented in Figure 5 summarizes the transcriptional responses of different strains under two conditions. However, it lacks annotations that describe the affected genes. Including general labels, such as gene ontology, would make the heat map more informative and justify its inclusion in the figure. Additionally, the volcano plots should feature labeling or highlighting of specific genes for clarity. Furthermore, incorporating a principal component analysis would be a valuable enhancement.

Minor comments:

Please use conventional nomenclature for the complemented strain to avoid confusion (DAL81 AB change to +DAL81 or DAL81 comp.).

Version 1:

Reviewer comments:

Reviewer #1

(Remarks to the Author)

Summary

The authors have done a very commendable job in responding to initial comments. The additional experimentation performed fills a number of knowledge gaps in the original manuscript. I have only minor comments that I feel can be handled editorially to further improve the manuscript. Overall, this is a very nice study that adds to our understanding of *C. albicans* pathogenesis, metabolism, and extracellular alkalinization.

Minor comments

Prior Q1 and new Fig. S3. The authors should more specifically point out that the growth defect of *dal81* and *stp2* mutants is mild in YPD and much more severe in casamino acids. This is consistent with their data showing that these mutants fail to alkalinize and utilize amino acids.

Line 284: The authors should remove the statement "of course, other Dal81 or Stp2-dependent...". The phrase "of course" should be removed entirely as it may not be clear to the reader. Secondly, this belongs better in the Discussion to speculate on what those phenotypes may be.

Fig 8B. The y-axis labeling seems incorrect. Please adjust.

Prior Q11. I appreciate that the authors have profiled growth of their mutants in a variety of host relevant carbon sources. However, the question originally posed is still not entirely answered. The point was to delineate whether *in vivo* phenotypes were due to lack of alkalinization or inability to grow/utilize host sourced nutrients. Perhaps it's both? I'm not advocating that the authors do additional experimentation, but a more nuanced explanation is warranted.

Prior Q12. The explanation regarding alkalinization and morphogenesis in the gut environment may be misguided. *C. albicans* does not readily adopt hyphal morphology in the gut and instead transitions to other morphotypes (e.g., GUT cells, PMID: 23892606). Again, a more thoughtful explanation/discussion of the results would further strengthen the conclusions.

Reviewer #2

(Remarks to the Author)

In this revision, Huang, Chen, Wu, Zou et al. have comprehensively and satisfactorily addressed the criticisms raised by 3 reviewers on the first version of their manuscript. They have provided additional experiments that elegantly demonstrate the importance of physical interaction between Spt2 and Dal81 for controlling alkalization in *C. albicans*. As such the study will be of importance for those interested in fungal pathogenicity and physiology.

I have only minor comments :

1) In the section « The physical interaction between Dal81 and Stp2 is required for pH alkalization and virulence of *C. albicans* », they provide information about the impact of mutating Dal81 residues involved in the interaction with Spt2 on *C. albicans* virulence. This comes before their section where they describe the role of Dal81 in commensalism and virulence using knock-outs. This is a bit awkward and I would recommend putting all data on commensalism and virulence in a single section.

2) There is still room for improvement of the text. Here is a list of suggested changes, with in particular the need to use the past tense all along the result section rather than alternating between past and present. I have stopped making suggestion at some point.

l.48 – A mutant lacking...

l.82-83 : to actively modify, either lowering or raising, the pH...

l.88 : that utilizes...

l.122 : pH alkalization regulation

l. 124 on both vaginal...

l.142 : 37°C

l.144 : defective in pH alkalization

l.149 : and a highly filamentous...

l.165 : defect...was also observed...

l.181 and elsewhere : alkalize

l.195 : appears to be

l.233 : we asked

l.239 : formed normal

l.244 : was indistinguishable

l.248-250 : check sentence

l.252 : exhibited

l.262 : suggested

l. 269 : were

l.284 : as well as other Dal81- or Stp2-

l.344 : of Stp2

Reviewer #3

(Remarks to the Author)

I am pleased that the authors have made substantial improvements to the manuscript and have addressed my major concerns: the strain background and the macrophage model.

They state that the *dal81* mutant was generated in the SC5314 background, a prototrophic reference strain broadly used in the field, which would address concerns about strain-dependent phenotypes. However, the data supporting this claim are not shown in the manuscript. I strongly encourage the authors to include representative validation and phenotypic confirmation using the SC5314-derived strain to substantiate this key point.

The addition of the J774A.1 murine macrophage line and the use of bone marrow-derived macrophages are appreciated and strengthen the relevance of the findings. The results appear consistent across models and enhance the overall robustness of the conclusions.

That said, the assessment of macrophage killing relies exclusively on LDH release, an endpoint assay that offers limited temporal resolution. The use of viability dyes such as propidium iodide (PI) or DRAQ5 could have provided dynamic, real-time insights into host cell death, adding mechanistic depth to the macrophage interaction data. Moreover, there is no direct readout addressing the susceptibility of *dal81* Δ/Δ to macrophage killing. One can only presume that, since this mutant causes less macrophage damage, it is more susceptible to immune-mediated clearance. Although not essential for publication, these points represent missed opportunities to enrich the study.

Similarly, while the updated images showing Stp2 nuclear localization are somewhat improved, the signal-to-noise ratio remains limited. Immunofluorescence may not be the optimal approach for confidently assessing nuclear translocation in this context. The authors support efficient Stp2 activation by showing proteolytic processing, and we can reasonably assume that this is followed by nuclear translocation; however, the evidence provided is suboptimal. For future work, I would recommend using fluorescent protein tags expressed from the native locus to provide more robust and interpretable localization data.

The authors have also made commendable improvements to the presentation and interpretation of the transcriptomic data. I appreciate the inclusion of Gene Ontology enrichment categories, which offer functional context to the expression patterns shown in the heat map. The addition of a PCA plot is also welcome, as it effectively visualizes variance across samples and reinforces the reproducibility of the RNA-seq data. Furthermore, the updated volcano plots with labeled pH alkalization–

associated genes are more informative and accessible to readers. These changes significantly enhance the clarity and impact of the transcriptomic analyses and fully address my prior concerns regarding Figure 5 (now revised as Figure 6).

The expanded section addressing the role of Dal81 and Stp2 in *C. albicans* commensalism is a valuable addition. The inclusion of competitive colonization assays in immunosuppressed mice adds functional depth and strengthens the conclusion that both transcription factors are required for efficient gastrointestinal persistence. The observation that the *dal81Δ/Δ stp2Δ/Δ* double mutant completely fails to colonize the gut is particularly striking and supports a model in which pH modulation is central to the commensal lifestyle. I also appreciate the authors' effort to consider microbial-intrinsic mechanisms of commensal fitness that may operate independently of host immunity. To further improve clarity, the authors might briefly elaborate on whether the colonization defect is due to impaired survival, altered nutrient utilization, or impaired niche competition, issues that could guide future work.

Finally, I recommend that the manuscript undergo careful proofreading to correct minor typographical errors and improve overall clarity and language flow.

Overall, the manuscript has improved considerably and contributes meaningful insights into *Candida albicans*–host interactions. With the inclusion of the SC5314-based data as claimed, I would support the publication.

Point-by point responses to referees' comments:

Referees' comments:

Reviewer #1

1. Major Comments

Q1. Fig S2B: While the number of colonies between WT and *stp2*, *dal81* mutants is similar, the size of those colonies are clearly different. The mutants are comparatively smaller, suggesting reduced growth or loss of fitness. The authors should comment on this in the Results.

Answer: We thank the reviewer for this insightful observation. Indeed, although the number of colonies formed by the wild-type and mutant strains appears comparable, the mutant colonies are visibly smaller. This difference is likely due to reduced growth of the mutant strains, as observed in both YPD liquid medium and YNB supplemented with 1% casamino acids (CAA) (**revised Fig. S3A and C**). The slower growth likely contributes to the smaller colony size observed on YPD plates (**revised Fig. S3B**). However, compared to their growth phenotypes, the alkalization defects of these mutants were significantly more pronounced, which we think is due to the inability to efficiently catabolize amino acids. We rephrased our descriptions in the revised manuscript. (**Lines 159-164 on page 6**)

Q2. While the ammonia release observed in the *dal81* mutant in Fig. 1E may be statistically significant compared to WT, I question its biological meaningfulness. If the ammonia release is not driving failed alkalization, then what other Dal81-dependent metabolic endpoints may be responsible? This should be noted in the Results and followed up in the Discussion.

Answer: We appreciate the reviewer for this thoughtful comment. Upon revisiting the original data, we noted that ammonia release was measured in both undiluted and 10-fold diluted samples in this study. In the original version of Fig. 1E (using undiluted samples), the difference in ammonia release between the *dal81Δ/Δ* mutant and the wild type was statistically significant but not visually distinct, likely due to measurement sensitivity being compromised by high sample concentration. Following reanalysis with 10-fold diluted samples, we observed that the *dal81Δ/Δ* strain, similar to *stp2Δ/Δ* mutant, released significantly less ammonia than the wild type (**revised Fig. 1E**). This suggests that reduced ammonia release may be a primary factor contributing to the failed alkalization phenotype of the *dal81Δ/Δ* mutant. This revised analysis, by optimizing sample dilution, eliminates the interference of high concentration and clarifies the metabolic phenotype difference between the mutant and wild type, thereby strengthening

the core conclusions of our study. (Lines 168-171 on page 6)

Q3. Please similarly address point #2 in line 264.

Answer: We thank the reviewer for this comment and the opportunity to clarify this aspect. We observed that the *dal81Δ/Δstp2Δ/Δ* double mutant exhibited a more pronounced reduction in ammonia release compared to the *stp2Δ/Δ* single mutant, yet showed no difference when compared to the *dal81Δ/Δ* mutant (revised Fig. 1E). These results suggest that the two transcription factors may have additive effects on ammonia release, though we could not rule out the possibility that other Dal81 or Stp2-dependent metabolic endpoints may contribute to this phenotype. (Lines 280 – 285 on pages 10 and 11)

The GM-BCP medium contains glycerol and yeast extract, providing a complex mixture of nucleotides, proteins, amino acids, sugars, and trace elements. Our RNA-seq and ChIP-seq analyses reveal that Dal81 and Stp2 jointly regulate the metabolism of carboxylic acids, organic acids, oxoacids, and amino acids (revised Fig. 7D). In addition, Dal81 independently regulates sugar metabolism, including pathways involving carbohydrates, hexoses, and glucose (revised Fig. S15), while Stp2 specifically controls genes involved in the metabolism and transport of carboxylic acids and related compounds. Given that high glucose levels are known to suppress pH alkalization (Vylkova et al., 2011), the combined disruption of sugar metabolism (via Dal81) and amino/ organic acids metabolism (via Dal81 and Stp2) in the double mutant likely impairs multiple alkalization pathways. This could explain the severe defect in pH alkalization observed in the *dal81Δ/Δstp2Δ/Δ* strain on solid GM-BCP medium.

We have incorporated this interpretation into the Results (Lines 168-171 on page 6) and expanded the Discussion to address the broader biological significance of Dal81 and Stp2-dependent metabolic regulation beyond ammonia release, highlighting their interconnected

roles in shaping cellular pH homeostasis (Lines 770-772 on page 28, Lines 778-779 and Lines 786-794 on page 29).

Reference

Vylkova, S., Carman, A. J., Danhof, H. A., Collette, J. R., Zhou, H., & Lorenz, M. C. (2011). The fungal pathogen *Candida albicans* autoinduces hyphal morphogenesis by raising extracellular pH. *MBio*, 2(3), 10-1128.

Q4. In Figure 1H, what is the comparator to determine percentage of survival? This should be explicitly described in the legend or the Methods.

Answer: We appreciate the reviewer for identifying this oversight and apologize for ambiguity in our initial description. To address this, we clarify that the survival of *C. albicans* in macrophages was assessed using an end-point dilution assay, following the methodology outlined in a previously published study (Vylkova et al., 2011), which we adapted for our work. Specifically, fungal survival was quantified as the ratio of colony-forming units (CFUs) recovered from macrophage-containing wells to those recovered from macrophage-free control wells, expressed as a percentage. This clarification has been integrated into the revised Methods section (Lines 995–1001 on Page 34), with the relevant reference now properly cited to ensure transparency and reproducibility (Line 988 on page 34).

Reference

Vylkova, S., & Lorenz, M. C. (2014). Modulation of phagosomal pH by *Candida albicans* promotes hyphal morphogenesis and requires Stp2p, a regulator of amino acid transport. *PLoS pathogens*, 10(3), e1003995.

Q5. The colony morphologies in Fig. 2F are quite distinct in WT, *dal81*, *stp2*, and double mutants. The authors should note this and provide an explanation as to why.

Answer: We appreciate the reviewer for highlighting this point and apologize for the lack of clarity in our initial presentation. Upon repeating the experiment, we observed distinct colony morphologies on GM-BCP medium. As shown in Fig. 2F, the wild-type and *Dal81*-complemented strains formed wrinkled colonies, a hallmark of robust hyphal development, whereas the *stp2Δ/Δ* and *dal81Δ/Δstp2Δ/Δ* double mutant strains produced smooth colonies, indicative of a yeast-dominant state with no detectable hyphal growth. The *dal81Δ/Δ* mutant displayed an intermediate phenotype, characterized by limited colony wrinkling. (revised Fig. 2F). We have updated the figure accordingly and now include a description of these morphological differences in the main text (Lines 291-297 on page 11). To further investigate

the basis of these differences, we examined the cellular morphology of each strain under the microscope. Consistent with the colony appearances, we observed that wild-type and complemented strains displayed elongated, branching hyphae, while *dal81Δ/Δ* cells showed defective filamentation. In contrast, *stp2Δ/Δ* and the double mutant remained entirely in the yeast form, with no evidence of hyphal differentiation (**revised Fig. 2G**). These findings suggest that impaired hyphal development contributes to the altered colony morphology and highlight the coordinated roles of Dal81 and Stp2 in regulating both alkalization and morphogenesis in *C. albicans*. (**Lines 291-304 on page 11**)

Q6. The epistasis data presented in Fig. 3 is very nice. Does Dal1p bind to the STP2 promoter and observed by Chip-Seq? Might this explain the observations here?

Answer: We thank the reviewer for this insightful suggestion. After carefully reviewing our ChIP-seq data, we found no evidence that transcription factor Dal81 binds to the promoter region of *STP2*, nor does *STP2* bind to the promoter of *DAL81*. Thus, these two transcription factors do not appear to directly regulate each other at the transcriptional level. As shown in Figure 3B, our data suggest that Dal81 influences the stability of Stp2 through protein–protein interaction, as supported by our complex formation assays (**revised Fig. 4C and D, Fig. 5A**). Therefore, the relationship between Dal81 and Stp2 is not epistatic in a classic transcriptional hierarchy but appears to depend on their physical association. Consistent with this model, overexpression of *DAL81* in the *stp2Δ/Δ* strain or overexpression of Stp2 in the *dal81Δ/Δ* background does not rescue the alkalization defects observed in the knockout strains. This supports the conclusion that both factors are required in coordination, and that one cannot compensate for the loss of the other through overexpression alone. (**Lines 326-353 on page 13**)

Q7. The western blot data in Fig. 3B and especially 3D should be quantified by densitometry and shown.

Answer: We thank the reviewer for this helpful suggestion. In response, we have performed densitometric quantification of the Western blot bands shown in Figures 3B and 3D. The quantified data have now been included in the revised figures (**revised Fig. 3B and Fig. 3D**), providing a clearer comparison of protein expression levels.

Q8. Replicate, unprocessed blots from Figs. 3B, D and 4C, D should be provided in the supplementary materials.

Answer: We thank the reviewer for this important suggestion. We have now included the replicate, unprocessed blots corresponding to Figures 3B, 3D, 4C, and 4D in the supplementary materials (**Supplementary Information, Pages 15-18**).

Q9. The co-localization data presented in Fig. 4 is not abundantly clear. While I can certainly observe punctate staining in the nuclei of WT and revertant, I also see punctate staining that does not overlap with nuclear staining. The authors should provide a quantitative readout of comparative colocalization amongst the strains.

Answer: We greatly appreciate the reviewer's insightful comment. In response, we re-performed the subcellular localization experiments using confocal microscopy to enhance image clarity and resolution. The revised images presented in **Figure 4A** offer a clearer depiction of Stp2 localization across strains. Additionally, we quantitatively analyzed the distribution of Stp2 in the wild-type, *dal81Δ/Δ*, and *STP2*^{OE}* mutant strains. Our results show that Stp2 mainly localizes to both the cytoplasm and nucleus in all four strains and Dal81 appears to have no effect on the nuclear localization or functional activation of Stp2 (**revised Fig. 4A**). This quantitative comparison has been incorporated into the revised figure and provides a more robust assessment of Stp2's subcellular distribution. (**Lines 386-396 on page 15**)

Q10. It is peculiar that the authors did not use the *stp2/dal81* double mutant in the in vivo colonization and virulence studies. It would be informative to know whether a more severe colonization defect could be observed at the day 3 timepoint. It's possible that in vivo Stp2 and Dal81 partially compensate one another.

Answer: We thank the reviewer for this valuable suggestion. Initially, we did not include the *dal81Δ/Δstp2Δ/Δ* double mutant in our colonization and virulence studies, as both the *dal81Δ/Δ* and *stp2Δ/Δ* single mutants exhibited markedly reduced intestinal colonization in the later stages of oral gavage infection. In response to the reviewer's comment, we conducted additional experiments to assess the in vivo colonization and virulence of the *dal81Δ/Δstp2Δ/Δ* double mutant. Our results show that the double mutant is completely incapable of colonizing the intestinal tract (**revised Fig. 8B**) and exhibits full attenuation of virulence (**revised Fig. 8E**). These findings suggest that Dal81 and Stp2 may partially compensate for each other in vivo, and that both are essential for effective colonization and pathogenicity. (**Lines 689-692 on page 26**)

Q11. Can the authors speculate whether the colonization and virulence defects observed by the loss of Stp2 and/or Dal81 are due to alkalization defects or due to their inability to use alternative carbohydrate sources available in the host? As these transcription factors control the metabolism of several organic acids, it is conceivable that loss of "virulence" is instead due to an overall fitness defect.

Answer: We thank the reviewer for this thoughtful and important comment. The host environment provides a variety of carbon sources, including glucose, carboxylic acids (such as lactate and α -ketoglutarate), amino acids, and N-acetylglucosamine (GlcNAc). To determine whether the colonization and virulence defects observed in the *dal81Δ/Δ*, *stp2Δ/Δ*, and

dal81Δ/Δstp2Δ/Δ mutants are due to a general fitness defect, we assessed their growth on eleven host-relevant alternative carbon sources, each used as the sole carbon source. Our results showed no significant growth defects in the mutant strains compared to the wild-type on most of these carbon sources, with the exception of lactate, citrate, acetate and GlcNAc (**revised Fig. S16**). These findings suggest that while the mutants exhibit specific metabolic deficiencies, their colonization and virulence defects are unlikely to be explained solely by a general fitness impairment. Instead, it is likely that the inability to induce environmental alkalization, along with specific defects in utilizing select host-relevant carbon sources, contributes to their attenuated colonization and virulence phenotypes. (**Lines 836-850 on page 30**)

Q12. The authors may consider performing an experiment in immunosuppressed mice. If reduced colonization is still observed, then it is likely that loss of Stp2 or Dal81 is independent of immune interaction and phagosomal escape and instead due to an inherent fitness cost. At minimum, this needs to be addressed in the Discussion.

Answer: We sincerely thank the reviewer for this insightful comment. Following the recommendation, we assessed the colonization capacity of the mutant strains in immunosuppressed mice. Our results showed that both the *dal81Δ/Δ* and *stp2Δ/Δ* mutants continued to exhibit reduced intestinal colonization compared to the wild-type strain (**revised Fig. 8C**), indicating that their colonization defects are not primarily due to altered immune interactions. Rather, these findings support the idea that Dal81 and Stp2 contribute to commensalism through intrinsic mechanisms. It is well established that neutral/alkaline pH promotes the yeast-to-hyphae transition (References 1 – 3), a morphological change that is critical for *C. albicans* colonization in the gut (References 4 – 5). Our data also show that Dal81 and Stp2 are required for the utilization of several host-relevant carbon sources such as lactate, citrate, acetate, and GlcNAc (**revised Fig. S16**). Therefore, we propose that Dal81 and Stp2 support fungal commensalism by coordinating environmental alkalization and nutrient utilization, thereby promoting both hyphal development and metabolic adaptation within the host. We have addressed this point in the revised Discussion to provide a more comprehensive interpretation of the mutants' commensalism phenotype (**Lines 820-854 on page 30**).

Reference

1. Davis, D. A. (2009). How human pathogenic fungi sense and adapt to pH: the link to virulence. *Current opinion in microbiology*, 12(4), 365-370.
2. Vylkova, S., Carman, A. J., Danhof, H. A., Collette, J. R., Zhou, H., & Lorenz, M. C. (2011). The fungal pathogen *Candida albicans* autoinduces hyphal morphogenesis by raising extracellular pH. *MBio*, 2(3), 10-1128.
3. Vylkova, S. (2017). Environmental pH modulation by pathogenic fungi as a strategy to

conquer the host. *PLoS pathogens*, 13(2), e1006149.

4. Du Toit, A. (2024). Hyphae promote *Candida albicans* fitness and commensalism in the gut. *Nature Reviews Microbiology*, 22(5), 258-258.
5. Liang, S. H., Sircaik, S., Dainis, J., Kakade, P., Penumutchu, S., McDonough, L. D., ... & Bennett, R. J. (2024). The hyphal-specific toxin candidalysin promotes fungal gut commensalism. *Nature*, 627(8004), 620-627.

Q13. I could not locate the Chip-Seq data in the supplementary data set. This should be provided.

Answer: We thank the reviewer for this helpful comment. The ChIP-Seq data have now been included in the revised supplementary materials. Specifically, these data are presented in **revised Fig. S14A-C** and **Supplementary Data Files 3 to 6**.

2. Minor Comments

Q1. Line 60: The term “normal abiotic factors” is inherently context-dependent. What is normal? Perhaps better to state “deviation from homeostatic conditions”.

Answer: We thank the reviewer for the helpful suggestion. In line with the recommendation, we have rephrased the term “normal abiotic factors” with “deviation from homeostatic conditions” to improve clarity and precision in the revised manuscript (**Line 70 on page 4**).

Q2. Line 86: *C. albicans* is not the most common human fungal pathogen. The dermatophytes (e.g., *Trichophyton*) are more common infecting microbes than the *Candida* species.

Answer: We appreciate the reviewer’s correction and apologize for the outdated and inaccurate description. The original statement was based on a 2007 publication (Noble and Johnson, 2007) and had not been updated. In the revised manuscript, we have replaced “the most common human fungal pathogen” with “a common opportunistic fungal pathogen” to reflect current understanding (**Line 96 on page 4**).

Reference

1. Noble, S. M., & Johnson, A. D. (2007). Genetics of *Candida albicans*, a diploid human fungal pathogen. *Annu. Rev. Genet.*, 41(1), 193-211.

Q3. Line 162: Please define MM in the text at least once, in the figure legend, or both.

Answer: In accordance with the reviewer's suggestion, we have added the full definition of "MM" (metsulfuron methyl) in both the main text (**Line 174 on page 6**) and the **figure legend of Fig. S4**.

Q4. Line 171: This sentence reads awkwardly. Please revise.

Answer: We thank the reviewer for the helpful feedback. In response, we have revised the sentence to improve clarity and readability (**Lines 186-188 on page 7**).

Q5. Line 209. Reference #28 indicates that hyphal growth initiates inside of acidic phagosomes and expanding hyphae rupture the phagosome driving proton leakage to result in alkalization. The authors should be careful in statements (e.g., line 213 but throughout) whether alkalization drives hyphal growth or is simply a consequence of the yeast-to-hypha transition. There should be more nuanced interpretation and could provide material for the Discussion.

Answer: We greatly appreciate the reviewer's careful reading and insightful suggestion. We acknowledge that reference #28 was incorrectly cited and have now removed it. We have also corrected the related statement for accuracy (**Lines 230-232 on page 9, Lines 246-247 on page 10**). In addition, we have expanded the Discussion section to address the nuanced relationship between alkalization and hyphal formation—considering whether alkalization drives hyphal growth or is instead a consequence of the yeast-to-hypha transition (**Lines 812-819 on pages 29-30**).

Q6. Line 415-417: Details regarding Chip-Seq should be moved to the Methods.

Answer: We thank the reviewer for this helpful suggestion. In response, we have integrated the detailed description of the ChIP-Seq protocol into the Methods section for clarity and consistency (**Lines 1118-1119 on page 36**).

Q7. Line 656: More detail regarding the macrophage cytotoxicity and fungal killing assays is required.

Answer: We thank the reviewer for this valuable comment. In response, we have added detailed descriptions of the macrophage cytotoxicity and fungal killing assays in the Methods section to improve clarity and reproducibility (**Line 985-994 on page 34**).

-Q8. Line X: A more nuanced discussion of why extracellular alkalization phenotypes differ so drastically in YNB+1% CAA, artificial saliva, and vaginal simulant fluid should be

discussed more thoughtfully. What are the compositions of those media with respect to amino, carboxylic, etc. acids?

Answer: We sincerely appreciate the reviewer's insightful comment. In response, we have expanded and refined the discussion on why extracellular alkalization phenotypes differ so markedly across YNB + 1% CAA, artificial saliva (AS), and vaginal simulant fluid (VSF) (**Lines 766-789 on pages 28-29**). Our results demonstrate that transcription factors Dal81 and Stp2 contribute differentially to alkalization depending on the medium. Specifically, Stp2 plays a dominant role in alkalization in YNB + 1% CAA, Dal81 is essential in VSF, and Stp2 is also important in AS. These differences are likely attributable to the distinct compositions and metabolic demands of each medium. YNB + 1% CAA contains primarily casamino acids. AS includes casamino acids and mucin. VSF contains glucose, lactic acid, and acetic acid. Our growth assays on different carbon sources revealed that Dal81 is critical for utilizing lactic acid, acetate, and citrate, as the *dal81Δ/Δ* mutant exhibits marked growth defects when these organic acids are provided as sole carbon sources (**revised Fig. S16**). Utilization of these acidic substrates likely drives alkalization in VSF through metabolic consumption. Conversely, Stp2 regulates genes related to amino acid transport and metabolism (**revised Fig. 5D and Fig. S15**), underscoring its importance in media rich in amino acids and peptides such as YNB + 1% CAA and AS. The distinct metabolic requirements of each medium likely account for the strain-specific alkalization patterns observed. This expanded interpretation has been incorporated into the revised manuscript, as noted above. We thank the reviewer again for highlighting the importance of this mechanistic distinction.

Q9. Fig. 2: I find the labeling of this figure unnecessarily confusing. Please remove the 1,2,3,4,5 labels and instead label with the appropriate strain name in each panel.

Answer: We apologize for the confusion. In response to the reviewer's feedback, we have removed the numerical labels (1, 2, 3, 4, 5) and replaced them with the corresponding strain names in each panel (**revised Fig. 2B-D**).

Q10. Fig. 7: The authors should continue using the same color schema for the strains here as they did throughout the remainder of the study.

Answer: We appreciate the reviewer's helpful suggestion. In the **revised Fig. 8**, we have standardized the color scheme for the strains to match that used throughout the rest of the study.

Reviewer #2

1. Major Comments

Q1. The authors use HA- or myc-tagged versions of Spt2 and Dal81 to assess chromatin binding and regulation of gene expression by the two TFs. They do not provide any evidence that these modified forms behave like the wild-type forms. Data should be provided that address this issue as the functionality of the tagged TFs will influence the results of the experiments shown in several figures.

Answer: We appreciate the reviewer's thoughtful comment. To address this concern, we compared the growth and alkalization of the untagged wild-type strain with the HA- and Myc-tagged Stp2 and Dal81 strains. The results demonstrate that the tagged strains exhibit similar growth and alkalization capacities in YNB + 1% CAA, indicating that HA and Myc epitope tags do not interfere with the functional activity of Dal81 and Stp2 (**revised Fig. S10**). (**Lines 427-431 on page 16**)

Q2. The authors state in the abstract "our results demonstrate that Dal81 physically interacts with Stp2 to co-regulate the expression of downstream target genes related to metabolism of organic acid, oxoacid, carboxylic acid and amino acid. This coordinated regulation mode is indispensable for the alkalization process and plays a pivotal role in modulating commensalism and pathogenicity of *C. albicans*." While I agree with the authors that their data demonstrate that Dal81 physically interacts with Stp2 AND co-regulate the expression of downstream target genes, I don't think that they provide direct evidence that the Stp2-Dal81 interaction is necessary to co-regulate the expression of downstream target genes. Indeed, what is shown is that the absence of one of the two partners impairs 1) binding to target promoters of the other partner and 2) gene expression of the target genes. That gene expression of the target genes is maintained to some extent when one of the two partners is present and further impaired when the two partners are lacking (Fig 6G-I) indicates that the interaction is somewhat dispensable for the expression of downstream target genes.

Answer: We sincerely thank the reviewer for their careful analysis and valuable suggestions, which have significantly helped us improve the clarity and strength of our conclusions. We agree that our previous interpretation may have overstated the indispensability of the Dal81–Stp2 interaction for downstream gene expression. While our data demonstrate physical interaction between Dal81 and Stp2, as well as co-regulation of several downstream target genes, we acknowledge that these results do not unambiguously prove that the physical interaction is essential for co-regulation of all target genes. To clarify this point, we reanalyzed the expression patterns of three representative target genes (*GDH2*, *PCK1*, and *PUT1*) in *dal81Δ/Δ*, *stp2Δ/Δ*, and *dal81Δ/Δstp2Δ/Δ* strains (**Revised Fig. 7H–J**). We observed the following: 1) *GDH2* expression was further reduced in the *dal81Δ/Δstp2Δ/Δ* mutant compared to *dal81Δ/Δ*, but not compared to *stp2Δ/Δ*. 2) *PCK1* expression was similarly reduced in both single and double knockouts. 3) *PUT1* expression levels were significantly different across all

three strains. These results suggest that while the Dal81–Stp2 interaction enhances regulation of these genes, it is not strictly required in all cases. We have revised the abstract (**Line 52 and Line 60 on page 3**) and discussion to reflect this more nuanced interpretation (**Lines 831-835 on page 20**).

Furthermore, the reduction in Stp2 binding to target promoters when Dal81 is missing could also result from reduced levels of Stp2 as shown in Fig. 3B.

Answer: Regarding the reviewer’s concern that reduced Stp2 binding in the absence of Dal81 could result from lower Stp2 protein levels (Fig. 3B), we performed ChIP–qPCR assays using overexpressed Stp2-HA in both WT and *dal81Δ/Δ* backgrounds. The results confirmed that deletion of *DAL81* abolishes Stp2 binding to the promoters of *GDH2*, *PCK1*, and *PUT1* (**Revised Fig. S14D**), independent of the total Stp2 protein abundance.

The additivity effect of knocking out the two genes for a number of investigated phenotypes is also indicative that each transcription factor can function in the absence of the other, thus questioning the conclusion that “This coordinated regulation mode is indispensable for the alkalization process”.

Answer: To address the reviewer’s comment on the additive phenotype in double mutants, we have expanded our analysis using RNA-seq and ChIP-seq data (**Revised Fig. 7D, Fig. S15, and Data S6**). Our findings indicate: 1) Dal81 and Stp2 co-regulate genes enriched in organic acid, oxoacid, carboxylic acid, and amino acid metabolism. 2) Dal81 specifically regulates sugar metabolism genes (e.g., glucose, hexose). 3) Stp2 specifically regulates genes involved in transport, such as organic acids transport and amino acids transport. Thus, the additive phenotypes observed in some conditions likely reflect the presence of both shared and distinct regulatory targets for Dal81 and Stp2, which we have now clarified in the revised discussion (**Lines 767-793 on pages 28-29, Lines 827-835 on page 30**).

Finally, as the authors have only evaluated the impact of knocking out STP2 or DAL81 on commensalism and virulence, the conclusion that “this coordinated regulation mode plays a pivotal role in modulating commensalism and pathogenicity of *C. albicans*” is not well supported. Ideally, one would like to have evidence that the two factors interact at the chromatin. One would also like to have data in a context whereby interaction is abolished while the two proteins remain present. Have the authors used alphafold-related tools to explore the interaction between Stp2 and Dal81? Could this identify key residues for the interaction? What would be the consequence on chromatin binding and gene expression of affecting such residues? While I acknowledge these experiments might be beyond the scope of this study, it seems important that the conclusions made by the authors are softened and that the discussion addresses these

questions.

Answer: We appreciate the reviewer's final point about the functional validation of Dal81 and Stp2 interaction and its relevance to chromatin binding, commensalism, and virulence. To further investigate this, we used AlphaFold to predict 21 potential interface residues between Dal81 and Stp2 (**Revised Fig. 5A, B**). We then created the following mutant strains:

1. *DAL81m2/dal81Δ*, with alanine substitutions at the two highest-scoring interaction residues. This mutation had minimal effect on protein levels or alkalization capacity on medium 199.
2. *DAL81m21/dal81Δ*, in which all 21 predicted residues were substituted. Co-IP assays showed a complete loss of Dal81 and Stp2 interaction (**Revised Fig. 5G**), accompanied by: 1) Reduced binding of both TFs to the promoters of *GDH2*, *PCK1*, and *PUT1* (**Revised Fig. 7G**). 2) Downregulated expression of *GDH2* and *PCK1* (**Revised Fig. 7K, L**). 3) Loss of alkalization ability (**Revised Fig. 5I**). 4) Attenuated virulence in a bloodstream infection model (**Revised Fig. 5J**).

These results provide more direct evidence that physical interaction between Dal81 and Stp2 is functionally relevant to gene regulation, alkalization, and virulence. We have accordingly softened our claims and expanded the discussion to acknowledge these limitations and suggest future directions (**Lines 53-54 on page 3, Line 136 on page 5, Lines 859-870 on page 31**). Once again, we thank the reviewer for their thoughtful and constructive feedback.

Q3. In some places, the authors state that the *stp2* and *dal81* knock out mutants have similar behaviors. These conclusions should be revisited. For instance, they state 1.154-155 that "Similar to the *stp2Δ/Δ* mutant, the *dal81Δ/Δ* mutant failed to release significant amounts of ammonia (Fig. 1E)" while the figure shows a slight (but significant) reduction for the *dal81* mutant compared to wt and a more drastic reduction for the *stp2* mutant. Thus I don't think one can state that the *stp2* and *dal81* have similar behavior.

Answer: We thank the reviewer for this insightful and constructive feedback. We have carefully re-examined our data and revised our manuscript accordingly to more accurately reflect the observed phenotypes. In our initial experiments, ammonia release was measured using both undiluted and 10-fold diluted samples collected from cells spotted at $OD_{600} = 1$ on GM-BCP plates. In the original Fig. 1E, data from undiluted samples at $OD_{600} = 1$ were used for plotting. Under these conditions, the difference between *dal81Δ/Δ* and wild-type was not readily apparent, likely due to assay saturation or reduced sensitivity at high concentrations. Upon reanalyzing the data using 10-fold diluted samples from the same $OD_{600} = 1$ condition, we observed that the *dal81Δ/Δ* strain released significantly less ammonia than WT. Importantly, there was no significant difference in ammonia release between *dal81Δ/Δ* and *stp2Δ/Δ* under these conditions (**revised Fig. 1E**). (**Lines 168-171 on page 6**)

On another instance, the authors state l. 257-260 that “Compared to that of the two single mutants, the alkalization defects were more pronounced in *C. albicans* deficient for both STP2 and DAL81 when cells were grown on YNB+1% CAA at initial pH 4.5 (Fig. 2B), artificial saliva (AS) (Fig. 2C), and vaginal simulating fluid (VSF) (Fig. 2D).” while Fig. 2C shows that the *dal81* mutant behaves almost like wild-type and the *stp2* mutant behaves almost like the double mutant. Fig. 2D shows that the *stp2* mutant behaves like wild-type while the *dal81* mutant behaves like the double mutant.

Answer: We appreciate the reviewer pointing out the need to better differentiate the phenotypes of the single mutants across different environmental conditions. In AS medium (**Revised Fig. 2C**), the *dal81Δ/Δ* mutant behaves similarly to WT, while the *stp2Δ/Δ* mutant displays a strong alkalization defect comparable to the double mutant. Conversely, in VSF medium (**Revised Fig. 2D**), *stp2Δ/Δ* resembles WT, while *dal81Δ/Δ* mimics the alkalization defect seen in the double mutant. These distinct phenotypes suggest that Dal81 and Stp2 regulate alkalization in a context-dependent manner, with each factor playing a more dominant role depending on the environmental condition. We have corrected and clarified this in the main text (**Lines 276-290 on Pages 10-11**) and further expanded upon it in the Discussion (**Lines 766-793 on pages 28-29**), where we now highlight the condition-specific regulatory roles of Dal81 and Stp2 in governing extracellular pH modulation.

These are important observations that could be discussed as they inform the conclusions of the authors (see major comment above) and provide insights on one of the questions raised in the introduction (see l.113-118). Overall, the conclusions should be fine-tuned to better reflect the data and possibly put in a better perspective. For instance, the observation that the *stp2* mutant has an alkalization defect in artificial saliva medium (Fig. 2C) is inconsistent with previously published results in Vylkova and Lorenz (PMID: 24626429).

Answer: In the cited study, *stp2Δ/Δ* cells showed delayed alkalization in AS medium, with the pH reaching only 5.3 at 5 hours but rising to ~8.2 by 24 hours. In our study, conducted in SN250 genetic background as the WT control, we observed that at 56 hours, the pH of the *stp2Δ/Δ* culture reached only 6.3, compared to 7.8 for WT (**Revised Fig. 2C**). These findings support a clear alkalization defect in the *stp2Δ/Δ* strain in our hands. We speculate that differences in strain background and gene disruption strategies may account for the apparent discrepancy. Specifically, our *stp2Δ/Δ* strain was constructed in the SN250 background using *HIS* and *LEU* markers, whereas the *stp2Δ/Δ* strain in the Vylkova and Lorenz study was constructed in the SC5314 background using the SAT-flipper method. We now address this point explicitly in the Discussion section (**Lines 794-805 on page 29**).

Q4. The discussion could be strengthened according to some of the comments above but also

by discussing how consistent the authors' description of the Spt2 regulatory network is with those presented in previous reports. Also, the authors may want to discuss more thoroughly the importance of Dal81 for alkalization, hyphae formation and virulence.

Answer: We thank the reviewer for the valuable comments. In response, we have strengthened the discussion section in several ways: 1) We have expanded on the effects of *DAL81* and *STP2* deletions on alkalization across different environmental conditions (**Lines 766-793 on pages 28-29**). 2) We now include a discussion of the discrepancies with previous reports, and suggest possible reasons for the observed differences, such as strain background and marker differences (**Lines 794-802 on page 29**). 3) We address the consistency of our findings on the Stp2 regulatory network with previous studies, highlighting the shared regulatory patterns (**Lines 747-766 on page 28**). Additionally, we provide a more thorough discussion of the importance of Dal81 in regulating alkalization (**Lines 744-805 on pages 28-29**), hyphal formation (**Lines 806-819 on pages 29-30**), and virulence (**Lines 820-854 on pages 30**). These additions aim to better integrate our findings with the existing literature and provide a clearer perspective on the distinct and overlapping roles of Dal81 and Stp2 in *C. albicans* physiology and pathogenicity.

2. Minor Comments

Q1. English, while overall good, should be improved. There are a number of sentences that do not work (1.225-227, It is the alkalization defect, instead of a general morphological defect, contributing to the inability of mutant cells lacking DAL81 to switch from yeast to hyphal forms; 1.298-300, It has been reported that the truncated version of Stp2 lacks the amino terminal nuclear exclusion domain (the first 99 amino acids) and mimics the proteolytically processed form(21); 1.438-434, For example, GDH2 encoding the putative NAD-specific glutamate dehydrogenase and PUT1 encoding the putative proline oxidase, both of which are related to metabolism of organic acid, oxoacid, carboxylic acid and amino acid (fig. S9B) and previously reported to be involved in pH alkalization(26), as well as PCK1 encoding the phosphoenolpyruvate carboxykinase gene, which plays a role in the small molecule biosynthetic process and may be involved in pH alkalization(31).)

Answer: We thank the reviewer for this valuable comment. We have carefully revised the manuscript to improve the clarity and correctness of the English throughout. In particular, we have corrected the specific sentences mentioned (**Lines 246-247 on page 10, Lines 343-344 on page 13, Lines 578-583 on page 22**) and avoiding potential language errors.

Q2. The references in the text lack a space for separation from the preceding word.

Answer: We have formatted all references according to the guidelines of *Nature*

Communications, using superscript numbering without a space before the citation, as per the journal's style. If any specific instance appears inconsistent, we will be happy to recheck and correct it accordingly.

Q3. 1.1: the authors have used alkalization and alkalization; they should decide for one or the other; I would recommend alkalization.

Answer: We thank the reviewer for the careful reading and helpful suggestion. In response, we have revised the manuscript to consistently use the term *alkalinization* throughout, in accordance with the reviewer's recommendation.

Q4. 1.36: I am not sure that screening 674 knock-out mutants can be qualified high throughput.

Answer: We thank the reviewer for this thoughtful observation. As the *C. albicans* genome comprises over 6,000 genes, our library of 674 gene knockouts represents a substantial but partial subset and does not qualify as high-throughput screening. In response, we have revised the terminology in the manuscript to "large-scale genetic screening" (**Line 46 on page 3, Line 131 on page 5**) to more accurately reflect the scope of our study.

Q5. 1.72: "fungal pathogens were found to be able to actively modify, either acidify or alkaline, the pH"; pH cannot be acidified or alkalized, it is decreased or raised.

Answer: We thank the reviewer for this valuable comment. We have revised the sentence accordingly to improve accuracy and clarity. Specifically, we replaced "acidify or alkaline" with "decrease or raise the pH" in the manuscript (**Line 83 on page 4**).

Q6. - 1. 87: Infection at mucosal surfaces are generally referred to as superficial infections, invasive infection corresponding to those where there is invasion of tissue.

Answer: We thank the reviewer for the helpful comment. We have revised the manuscript to accurately reflect the terminology by changing "invasive opportunistic infections" to "superficial infections" (**Line 97 on page 4**).

Q7. - 1. 92-93: I don't think there are cases of morbidity or mortality

Answer: We thank the reviewer for this valuable comment and agree with the clarification. As stated in the WHO Fungal Priority Pathogens List (FPPL) released in 2022, the exact morbidity and mortality rates caused by this fungus are currently unknown. However, the report notes that this fungus is associated with a serious risk of mortality and/or morbidity. We have revised our

statement in the manuscript to reflect this point more accurately. The sentence now reads:
"Due to the difficulty of eradication and treatment, and significant risk of morbidity and mortality, this fungus was recently classified in the critical priority fungal pathogen group by the World Health Organization (WHO)" (Lines 102-103 on page 4).

Reference

1. World Health Organization. (2022). *WHO fungal priority pathogens list to guide research, development and public health action*. World Health Organization.

Q8. - Inconsistent use of starting pH of YNB-CAA medium: line 143: pH 4.0, line 216: pH 4.5, line 273: pH 4.3.

Answer: We sincerely thank the reviewer for this helpful observation. We initially used a starting pH of 4.0, as described in Reference #1, to measure pH changes. In subsequent experiments, we primarily used a starting pH of 4.5, following the protocol in Reference #2 from the same research group. Both references report consistent results despite the slight difference in initial pH. Therefore, we believe that maintaining the starting pH within the range of 4.0 to 4.5 is appropriate and likely has only a minor impact on the overall conclusions.

References

1. Vylkova, S., Carman, A. J., Danhof, H. A., Collette, J. R., Zhou, H., & Lorenz, M. C. (2011). The fungal pathogen *Candida albicans* autoinduces hyphal morphogenesis by raising extracellular pH. *MBio*, 2(3), 10-1128.
2. Vylkova, S., & Lorenz, M. C. (2014). Modulation of phagosomal pH by *Candida albicans* promotes hyphal morphogenesis and requires Stp2p, a regulator of amino acid transport. *PLoS pathogens*, 10(3), e1003995.

Q9. Results from Fig. 1C are described significant (1.146) but it is showing only 1 out of 3 replicates. Instead, the authors should show mean + SD and statistical significance of all three replicates. This panel would also benefit to have data for the *stp2* knock-out mutant.

Answer: We greatly appreciate the reviewer's careful reading of our manuscript and thoughtful suggestions. In response, we have revised Figure 1C to present the mean \pm standard deviation (SD) and include statistical analysis based on all three biological replicates. Additionally, we have incorporated the data for the *stp2 Δ* mutant into the **revised Fig. 1C** as requested.

Q10. 1.151-154 is not exact. Fig. S2 clearly shows that the *dal81* mutant has decreased growth compared to the WT.

Answer: We greatly appreciate the reviewer's careful reading of our manuscript and insightful comment. As correctly pointed out, the initial Figure S2 (**revised Fig. S3**) clearly shows that the *dal81Δ/Δ* mutant exhibits a growth defect compared to the wild-type. We have revised the corresponding statement in the manuscript to accurately reflect this observation (**Lines 159-161 on page 6**).

Q11. 1. 212-213: That Dal81 regulates pH alkalization in a way similar to that of Stp2 is not demonstrated at this stage of the ms.

Answer: We agree that the statement was not sufficiently supported at this stage of the manuscript. Accordingly, we have removed the phrase "in a way similar to that of Stp2" to avoid overinterpretation (**Line 233 on page 9**).

Q12. 1.225 : revise sentence according to the experiment shown in Fig S4B: "Strains were grown overnight in YPD, then transferred to medium 199 buffered at pH 4.5 or 7.5."

Answer: We greatly appreciate the reviewer's careful reading and helpful suggestion. In accordance with the experimental conditions shown in initial Fig. S4B (**revised Fig. S6B**), we have revised the sentence as follows: "Cells of indicated strains were grown overnight in YPD, then transferred to medium 199 buffered at pH 4.5 or 7.5" (**Lines 244–246 on page 9**).

Q13. 1.248-249 and Fig. 2A: please mention at which time point the RNA for RT-qPCR were obtained. This is also lacking in the methods part.

Answer: We thank the reviewer for pointing out this important omission. The RNA samples for RT-qPCR were collected 6 hours after *C. albicans* cells were transferred to SD or YNB + CAA medium. We have now included this information in both the figure legend (**Line 309 on page 12**) and the Methods section (**Line 1013 on page 35**) to ensure clarity and reproducibility.

Q14. Fig. 2A: the red bar for STP2 should be cut as for DAL81

Answer: We thank the reviewer for the valuable comment. In the **revised Figure 2A**, the red bar for Y has been appropriately truncated for consistency.

Q15. Fig. 2E: This is unlikely YNB+CAA medium. It might be liquid GM-BCP medium. Should be clarified.

Answer: We thank the reviewer for the insightful comment and apologize for the lack of clarity.

In Figure 2E, we used YNB+CAA medium supplemented with bromocresol purple as a pH indicator. This clarification has been added to the figure legend (**Line 318 on page 13**).

Q16. 1. 262-264: “that the mutant lacking both STP2 and DAL81 displayed the strongest defects in pH alkalization on the solid GM-BCP medium (Fig. 2F)” is “ due to its lower release rate of volatile ammonia (Fig. 1E).” has not been demonstrated; it is likely due.

Answer: We thank the reviewer for the valuable comment. We agree that the causality has not been definitively demonstrated. Accordingly, we have revised the statement in the manuscript to indicate that the stronger defect in pH alkalization observed in the double mutant is likely due to its reduced release of volatile ammonia (**Line 284 on page 11**).

Q17. Fig. 3A-C: same comment as before: mention at which time point the RNA for RT-qPCR and proteins for western blot were obtained.

Answer: We apologize for the oversight regarding the omission of sample collection time points. We have now added the specific time points for RNA extraction used in RT-qPCR and protein collection for Western blotting in the legends of Figure 3 (**Lines 358 and 361 on page 14**).

Q18. 1. 363 and 1. 369: what do the author mean by in vivo co-IP.

Answer: We apologize for the lack of clarity. Our initial intention was to distinguish Co-IP from yeast two-hybrid assays by referring to Co-IP as “in vivo,” given that it detects protein-protein interactions within cells under more physiological conditions. However, we acknowledge that “in vivo Co-IP” is not a standard term. To avoid confusion, we have revised the term to “Co-IP” in the manuscript (**Lines 425 and 436 on page 16**).

Q19. Fig. 4CD: in panel C, indicate IP-HA; in panel D, indicate IP-myc; Panel C would benefit from a control with untagged Stp2-HA while panel D would benefit from a control with untagged Dal81-myc; this is especially important as a protein is detected with the myc antibody upon HA IP when Dal81 is not tagged. Why only one band is immunoprecipitated for Stp2-HA ?

Answer: We sincerely appreciate the reviewer’s insightful comments. In response, we have re-performed the co-IP experiments using untagged Stp2-HA as a control for IP-HA (**revised Fig. 4C**) and untagged Dal81-Myc as a control for IP-Myc (**revised Fig. 4D**). These controls have been included in the revised figure to ensure the specificity of the interactions observed. Consistent with our previous findings, the results confirm that Dal81 physically interacts with

Stp2. Regarding the presence of only one immunoprecipitated band for Stp2-HA, we believe this is due to the fact that Dal81 interacts specifically with the processed form of Stp2, which translocates to the nucleus following proteolytic cleavage in the cytoplasm.

Q20. Fig. 5B: Instead of showing these Venn diagrams, the authors may consider showing the one from Fig.S8A together with the GO enrichment analysis of Fig.S8B. It would highlight that most of the identified genes are regulated by both Dal81 and Stp2.

Answer: We thank the reviewer for this insightful suggestion. In response, we have moved the original Fig. 5B to the supplementary information (**revised Fig. S12**). We have incorporated the previous Fig. S8A into the main manuscript as the **revised Fig. 6C**, and included the GO enrichment analysis from the previous Fig. S8B as the **revised Fig. 6D**. These changes better illustrate the overlap in gene regulation by Dal81 and Stp2, as recommended.

Q21. Fig. 5C: Here the authors compare the *dal81* mutant and *stp2* mutant to SN250. In the figure legend they mention that the mutants were compared to the WT (SC5314). It is confusing and should be clarified.

Answer: We appreciate the reviewer's observation. We apologize for the confusion. The *dal81Δ/Δ* and *stp2Δ/Δ* mutants were indeed compared to SN250, not SC5314. We have corrected the figure legend accordingly in the updated Fig. 6E (**Lines 552–554 on page 22**) to reflect this accurately.

Q22. Fig. 6B reads “Sef1 binding Motif”. It should read “Stp2-binding Motif”.

Answer: We thank the reviewer for the careful reading and helpful suggestion. The label has been corrected to “Stp2-binding Motif” in the **revised Fig. 7B**.

Q23. Fig. 6H should read PCK1/ACT1

Answer: We thank the reviewer for the attentive reading and helpful correction. The label has been updated to *PCK1/ACT1* in the **revised Fig. 7I**.

Q24. 1.439: Stp2-Myc should read Stp2-HA

Answer: We thank the reviewer for the careful observation. This has been corrected in the revised manuscript, where “Stp2-Myc” has been replaced with “Stp2-HA” (**Line 599 on page 23**).

Q25. Fig. 7ABC: the 3 panels could be merged in a single panel.

Answer: We thank the reviewer for the helpful suggestion. In the revised manuscript, the three panels originally shown in Fig. 7A–C have been consolidated into a single panel (**revised Fig. 8A**)

Q26. Fig. S1B: Why SC-Medium? This wasn't use elsewhere. Why not performing a spot dilution assay on YNB-CAA?

Answer: We thank the reviewer for this valuable observation. In response, we repeated the experiment using YNB + 1% CAA. The results confirm that the *grr1Δ/Δ* mutant exhibits a significant growth defect under this condition (**revised Fig. S2B**).

Q27. Fig. S2: The figure legend title is „...dal81d mutant cells have no obvious growth defect. “ This is not reflecting data in Fig.S2B and C. Fig. S2C shows only one replicate. Showing the mean + SD of all 3 replicates would be better.

Answer: We thank the reviewer for this thoughtful feedback. We agree that the *dal81Δ/Δ* mutant displays a slight growth defect, and we have revised the figure legend title accordingly in the **revised Fig. S3**. Furthermore, we repeated the experiment shown in the initial Fig. S2C using a BioTek Synergy 2 Multi-Mode Microplate Reader to automatically record OD values every 15 minutes. The mean ± SD from three biological replicates is now presented in the **revised Fig. S3C**, as recommended.

Q28. Fig. S5B: In the figure legend the authors mention that they used YPD medium for the hyphae formation assay. Why not buffered Medium 199 like in Fig. S4? Why using different media?

Answer: We apologize for the typographical error in the original submission. The hyphae formation assay shown in Fig. S5B was in fact performed using medium 199, consistent with the conditions in Fig. S4B. We thank the reviewer for pointing out this discrepancy, and the figure legend has been corrected in the **revised Fig. S7**.

Reviewer #3

1. Specific Comments

Q1. The background strain SN250 is commonly used for its convenience in generating deletion mutants, allowing for the selection of transformants through auxotrophy complementation.

However, it is important to note that amino acid utilization and auxotrophic markers can have confounding effects on the results. Therefore, the use of prototrophic strains is highly recommended for these studies. We are not suggesting the creation of a new set of strains, but we encourage consideration of this point for future research.

Answer: We greatly appreciate the reviewer for this valuable and constructive suggestion. To address this, we successfully generated a *dal81Δ/Δ* mutant strain in the prototrophic SC5314 background using the SAT-flipper method. As demonstrated in **Review Fig. 1**, the alkalinization phenotype of this strain is consistent with that of the *dal81Δ/Δ* mutant in the SN250 background, we recognize the significance of this consideration and will continue to utilize prototrophic strains such as SC5314 in future studies.

Review Fig. 1. *dal81Δ/Δ* M1 mutant cells display impaired alkalinization on medium 199. *C. albicans* SN250, *dal81Δ/Δ*, the complemented strain (DAL81 comp.), SC5314, and the newly generated *DAL81* deletion mutant in the SC5314 background (*dal81Δ/Δ* M1) were grown overnight in YPD, harvested by centrifugation, washed with water, and resuspended in medium 199 (initial pH 4.0) at an OD₆₀₀ of 1.0. Cultures were then serially diluted (1:5) in 96-well plates. Alkalinization was assessed after 24 hours of incubation at 37 °C and visualized by a color shift in the medium from yellow to orange-red.

Q2. In the library screen, two hits were identified: DAL81 and GRR1. Were there any other notable hits to mention, even if the phenotypes are mild, partial, or delayed? We believe this would enhance the significance of this work.

Answer: We thank the reviewer for this insightful comment. During our library screening, mutants unable to grow in medium 199 (initial pH 4.0) were eliminated from subsequent analysis. From the remaining strains, we identified 23 deletion mutants with defects in environmental alkalinization (**revised Fig. S1**). Among these, the *dal81Δ/Δ* and

*grr1*Δ/Δ mutants displayed the most significant deficiencies.

Q3. The experiments involving macrophages are quite intriguing. Did the authors assess phagosome maturation at later time points? Investigating the temporal dynamics of phagosome activation would enhance the relevance of this study. While RAW264.7 macrophages have been utilized in *C. albicans* survival assays, we recommend considering other cell lines, such as J774A.1 or THP-1-derived macrophages, as well as primary murine macrophages (bone marrow-derived). The RAW264.7 macrophages lack a crucial component of the NLRP3 inflammasome complex, which plays an essential role in the interaction between *C. albicans* and these immune cells. Therefore, we strongly suggest verifying the key results using a more suitable cell line.

Answer: We extend our gratitude to the reviewer for these insightful and constructive suggestions. In response, we evaluated phagosome maturation in J774A.1 macrophages for up to 150 minutes after infection with *C. albicans*. This time point was chosen because by 150 minutes, most wild-type *C. albicans* cells have developed elongated hyphae, which typically cause macrophage rupture and increased cell death (**revised Fig. 5B**). The *C. albicans* cells were labeled with pHrodo; however, since fluorescence fades following cell division, we restricted our analysis to the 150-minute window to ensure accurate tracking. Images were taken every 10 minutes to monitor the temporal dynamics of phagosome activation during this period (**revised Fig. 5**). As recommended by the reviewer, we also conducted *C. albicans* survival assays using both J774A.1 macrophages and bone marrow-derived macrophages (BMDMs). The results from these additional models matched those observed with RAW264.7 macrophages (**revised Fig. 1H–J**), thus reinforcing our conclusions.

Q4. The phenotypes of the *stp2* and *dal81* mutants display an additive effect in amino acid metabolism; however, they appear to have different roles in AS (*stp2*) and in VSF (*dal81*). A more detailed discussion is necessary to explore these differences. Do the authors have any transcriptional data that could enhance the discussion of these results? Additionally, can the authors speculate on the implications of these differences?

Answer: We thank the reviewer for this insightful comment. This question was also raised by Reviewer #2, and we have addressed it in the revised Discussion section. Specifically, we incorporated transcriptomic data alongside growth assays on various alternative carbon sources to explore the distinct roles of Dal81 and Stp2 in regulating alkalization under different extracellular conditions (**Lines 766-793 on pages 28-29**). Given that nutrient composition and pH vary substantially across different host niches, we propose that Dal81 and Stp2 mediate metabolic responses tailored to these microenvironments. For instance, in the vaginal tract, Dal81 appears to play a more dominant role in regulating pH alkalization. We further

speculate on the physiological and pathogenic implications of these functional distinctions between Dal81 and Stp2 in the revised manuscript (**Line 802-805 on page 29**).

Q5. Demonstrating the nuclear localization of Stp2 is quite challenging, particularly with immunohistochemistry (IHC). Others have had slightly better success using fluorescently tagged versions of this protein. The authors should consider adopting a similar approach to convincingly demonstrate that the deletion of DAL81 does not impact the nuclear translocation of Stp2. Alternatively, the processing of Stp2 could serve as evidence for this claim.

Answer: We thank the reviewer for this constructive suggestion. We agree that directly demonstrating the nuclear localization of Stp2, particularly through immunohistochemistry (IHC), is technically challenging. In our study, we used Western blot analysis to assess the activation and processing of Stp2 as an indirect indicator of its nuclear translocation. Specifically, when cultured in YNB + 1% CAA, the *dal81Δ/Δ* mutant exhibited both the full-length and cleaved (active) forms of Stp2, comparable to those observed in the wild-type strain (**revised Fig. 3B**). This suggests that Stp2 undergoes proper processing and activation even in the absence of Dal81, implying that its nuclear translocation is not impaired. We have clarified this point in the revised manuscript (**Lines 392-396 on page 15**).

Q6. The heat map presented in Figure 5 summarizes the transcriptional responses of different strains under two conditions. However, it lacks annotations that describe the affected genes. Including general labels, such as gene ontology, would make the heat map more informative and justify its inclusion in the figure. Additionally, the volcano plots should feature labeling or highlighting of specific genes for clarity. Furthermore, incorporating a principal component analysis would be a valuable enhancement.

Answer: We sincerely thank the reviewer for these insightful comments. In the revised manuscript, we have enhanced the clarity and informativeness of how the transcriptomic data is presented, according to the reviewer's advices. The heat map (**revised Fig. 6B**) clearly summarizes the transcriptional responses of different strains under two conditions and highlights the similarities in gene expression patterns between the *dal81Δ/Δ* and *stp2Δ/Δ* mutant strains. For this reason, we have kept this panel in the figure. To offer functional context, we added several groups of Gene Ontology (GO) enrichment analyses in the **revised Fig. 6D**. In line with the reviewer's suggestion, we also included a Principal Component Analysis (PCA) plot in the revised manuscript (**revised Fig. 6A**), which effectively shows the variance across the transcriptomic datasets and makes the data easier to interpret. Additionally, the volcano plots (**revised Fig. 6E**) have been improved by labeling genes known to be associated with pH alkalinization, thus enhancing both their clarity and relevance.

2. Minor Comments

Please use conventional nomenclature for the complemented strain to avoid confusion (DAL81 AB change to +DAL81 or DAL81 comp.).

Answer: We appreciate the reviewer for this valuable comment. As recommended, we have substituted the term “DAL81 AB” with the conventional nomenclature “DAL81 comp.” across the entire manuscript and all figures to avoid potential confusion.

Response to the reviewers

1. Reviewer #1

Minor comments (Remarks to the Author):

Q1. *Prior Q1 and new Fig. S3. The authors should more specifically point out that the growth defect of *dal81* and *stp2* mutants is mild in YPD and much more severe in casamino acids. This is consistent with their data showing that these mutants fail to alkalinize and utilize amino acids.*

Answer: We would like to thank the reviewer for this insightful comment. We have accepted the reviewer's suggestion and revised the relevant description in the revised manuscript accordingly. "In comparison to the wild-type strain, both the *dal81* Δ/Δ and *stp2* Δ/Δ mutants exhibited only mild growth defects in YPD medium but showed significantly impaired growth in YNB medium supplemented with 1% casamino acids (CAA)". **(Lines 153-156 on Page 5, Fig. S3)**

Q2. *Line 284: The authors should remove the statement "of course, other Dal81 or Stp2-dependent...". The phrase "of course" should be removed entirely as it may not be clear to the reader. Secondly, this belongs better in the Discussion to speculate on what those phenotypes may be.*

Answer: We would like to express our gratitude to the reviewer for the valuable suggestions. In response to the reviewer's feedback, we have replaced the phrase "of course" with "and additional" in the revised manuscript **(Lines 250 on Page 8)**. Furthermore, the relevant speculative content about the additive alkalinization defect of the *dal81* Δ/Δ *stp2* Δ/Δ double mutant has been integrated into the Discussion section, where we elaborate on this possibility from the following perspectives: 1) Beyond amino acid metabolism, other Dal81- or Stp2-dependent metabolic endpoints may also contribute to the exacerbated alkalinization defects observed in the *dal81* Δ/Δ *stp2* Δ/Δ double knockout mutant; 2) The differences in alkalinization phenotypes among the mutants across various culture media may stem from variations in the metabolic utilization of specific components in each medium. Specifically, YNB + 1% CAA primarily contains casamino acids, while artificial saliva includes both casamino acids and

mucin. In contrast, vaginal simulating fluid is mainly composed of glucose, lactic acid, and acetic acid. Finally, GM-BCP medium contains glycerol and yeast extract, providing a rich mix of nucleotides, proteins, amino acids, sugars, and trace elements; 3) Our RNA-seq and ChIP-seq data reveal that, in addition to shared target genes involved in amino acid metabolism, which are coregulated by Dal81 and Stp2, there are other amino acid metabolism-related genes specifically regulated by either Dal81 or Stp2. Moreover, genes associated with amino acid transport are uniquely controlled by Stp2. These findings help explain the enhanced alkalization defects observed in the *dal81Δ/Δstp2Δ/Δ* double knockout mutant in YNB + 1% CAA and artificial saliva; 4) Given that Dal81 regulates glucose metabolism (Fig. S15) and is pivotal for the utilization of lactic acid and acetic acid (Fig. S16), together with the previous observation that glucose suppresses environmental alkalization, we speculate that the impaired utilization of these acidic compounds and glucose in the *dal81Δ/Δ* mutant likely underlies the notable alkalization defects observed in vaginal simulating fluid. The combined disruption of sugar metabolism (via Dal81) and amino/organic acid metabolism (via Dal81 and Stp2) in the double mutant probably impairs multiple alkalization pathways. This could account for the more pronounced reduction in ammonia release observed in the *dal81Δ/Δstp2Δ/Δ* strain on solid GM-BCP medium (Fig. 1H). **(Lines 593-626 on Pages 17-18)**

On the other hand, we also provided detailed information to explain the observation that the *dal81Δ/Δstp2Δ/Δ* double knockout strain exhibited a total loss of gut colonization and a complete absence of virulence, with the following key points included. 1) Based on recent studies, it is plausible that pathogenicity is not an intrinsic characteristic of fungi themselves, instead, it arises as a consequence of disrupted or imbalanced interactions between the microbe and its host. Given that Dal81 plays a significant role in regulating both extracellular pH alkalization and hyphal formation in *C. albicans*, these two factors likely contribute to the critical involvement of Dal81 in promoting commensalism and virulence; 2) The double knockout strain (*dal81Δ/Δstp2Δ/Δ*) exhibited a total loss of gut colonization and a complete absence of virulence, pointing to an additive effect of Dal81 and Stp2 in driving both commensalism and pathogenicity. This combined impact may stem not only from the distinct sets of genes regulated by Dal81 and Stp2. Even among the genes they co-regulate, the

contribution of each transcription factor may vary based on the biological context; 3) It is important to note that our current evidence supporting the role of both Dal81- and Stp2-mediated regulation of pH alkalization and hyphal formation in gut colonization of *C. albicans* remains incomplete. Further investigations, such as exploring the link between pH alkalization and candidalysin production, are necessary to strengthen these connections; 4) On the other hand, we cannot exclude the possibility that additional factors, such as the gut-specific morphological variants, may also contribute to Dal81- and Stp2-mediated regulation of gut colonization. Future studies will focus on investigating whether the regulatory networks controlled by Dal81 and Stp2 influence the White-to-GUT switch and GUT-associated metabolic adaptations. **(Lines 653–696 on Pages 18-19)**

Q3. Fig 8B. The y-axis labeling seems incorrect. Please adjust.

Answer: We sincerely thank the reviewer for identifying this issue and apologize for the error due to inattention. Upon careful checking of our raw data, we can confirm that the values themselves remain accurate and unchanged. However, the y-axis in Fig. 8B was incorrectly labeled as “Log₁₀ (CFU/g)”, identical to the label used in Fig. 8A. We apologized for inadvertently applying the y-axis labeling method (“Log₁₀ (CFU/g)”) from Fig. 8A directly to Fig. 8B without considering the specific context of this figure.

In our animal model experiments, we observed that the *dal81Δ/Δstp2Δ/Δ* double mutant was completely cleared following oral gavage, with virtually no colonies detected (“0 CFU”) on agar plates cultured from the fecal samples. This finding made the labeling method used for Fig. 8A unsuitable for Fig. 8B. We have therefore adopted a revised labeling method, “*C. albicans* CFU/g in feces (×10⁶)”, which has been validated in relevant published literatures (see below). This correction has been implemented in the revised manuscript. **(Revised Fig. 8B)**

Reference:

Savage, H. P., Bays, D. J., Tiffany, C. R., Gonzalez, M. A., Bejarano, E. J., Carvalho, T. P., ... & Bäumlér, A. J. (2024). Epithelial hypoxia maintains colonization resistance against *Candida albicans*. *Cell Host & Microbe*, 32(7), 1103-1113.

Q4. *Prior Q11. I appreciate that the authors have profiled growth of their mutants in a variety of host relevant carbon sources. However, the question originally posed is still not entirely answered. The point was to delineate whether in vivo phenotypes were due to lack of alkalinization or inability to grow/utilize host sourced nutrients. Perhaps it's both? I'm not advocating that the authors do additional experimentation, but a more nuanced explanation is warranted.*

Answer: We appreciate the reviewer for highlighting this important point and apologize for the lack of clarity in our initial response. In the revised Discussion section, we propose that the colonization and virulence defects in the *dal81Δ/Δ* and *stp2Δ/Δ* mutants arise from an overall fitness impairment driven by two key factors: impaired extracellular alkalinization and an inability to effectively utilize host-derived nutrients. This notion was raised by the following observations: 1) Dal81- and Stp2 -dependent metabolic adaptation to alternative carbon sources may contribute to both commensalism and virulence in *C. albicans*; 2) The colonization defects are likely driven by impaired alkalinization, defective hyphal formation, and compromised metabolic adaptation, rather than altered interactions with the host immune system; 3) Beyond their roles as nutrients, defects in the metabolism of specific alternative carbohydrates in the *dal81Δ/Δ* and *stp2Δ/Δ* mutants may lead to a failure to alkalinize the extracellular environment within the host.

We have now strengthened and clarified this interpretation in the revised Discussion section to more accurately reflect this nuanced understanding. **(Lines 711–728 on Page 20)**

Q5. *Prior Q12. The explanation regarding alkalinization and morphogenesis in the gut environment may be misguided. C. albicans does not readily adopt hyphal morphology in the gut and instead transitions to other morphotypes (e.g., GUT cells, PMID: 23892606). Again, a more thoughtful explanation/discussion of the results would further strengthen the conclusions.*

Answer: We thank the reviewer for the insightful suggestions and have revised the relevant aspects of the Discussion section accordingly, as outlined below. First, our results showed that both the *dal81Δ/Δ* and *stp2Δ/Δ* mutants continued to show reduced intestinal colonization compared to the wild-type strain even in immunosuppressed mice (**Fig. 8C**), indicating that

their colonization defects are likely driven by impaired alkalization, defective hyphal formation, and compromised metabolic adaptation, rather than altered interactions with the host immune system (**Lines 717–719 on Page 20**). Second, although the commensal state of *C. albicans* is typically characterized by the absence of filamentation, epithelial invasion, and host cell damage, recent studies have argued that virulence traits, long recognized primarily for their role in inducing host diseases, such as hyphae formation and production of the hyphal-specific toxin candidalysin, may play an unexpected active role in facilitating the establishment of *C. albicans* as a gut commensal. Specifically, the study from Bennett and colleagues (*Nature*, 2024) revealed that producing hyphae actually enhances the ability of *C. albicans* to thrive as a commensal when colonizing mice with an intact, normal gut microbiota. The strain capable of switching to filamentous forms gained a competitive advantage over those locked in the yeast state and importantly, this competitive benefit relies on hyphal-specific toxin candidalysin, which functions to disrupt the metabolic activity, growth patterns, glucose uptake. It is plausible that pathogenicity is not an intrinsic characteristic of fungi themselves, instead, it arises as a consequence of disrupted or imbalanced interactions between the microbe and its host. Given that Dal81 plays a significant role in regulating both extracellular pH alkalization and hyphal formation in *C. albicans*, these two factors likely contribute to the critical involvement of Dal81 in promoting commensalism and virulence (**Lines 653-669 on Pages 18-19**). Third, it is important to note that our current evidence supporting the role of both Dal81- and Stp2-mediated regulation of pH alkalization and hyphal formation in gut colonization of *C. albicans* remains incomplete. Further investigations, such as exploring the link between pH alkalization and candidalysin production, are necessary to strengthen these connections. On the other hand, we cannot exclude the possibility that additional factors, such as the gut-specific morphological variants, may also contribute to Dal81- and Stp2-mediated regulation of gut colonization. A previous study by Pande *et al.* highlighted that the “White-to-GUT” switch promotes *C. albicans* commensalism in the gastrointestinal (GI) tract, with a specialized, gastrointestinally induced transition (GUT) cell type, regulated by the transcription factor Wor1, actively promoting the maintenance of a commensal lifestyle in this fungus. Remarkably, GUT cells exhibit a reprogrammed cellular metabolism that is adapted to optimize nutrient utilization in the GI environment. Future studies will focus on investigating whether the regulatory

networks controlled by Dal81 and Stp2 influence the White-to-GUT switch and GUT-associated metabolic adaptations **(Lines 682-696 on Page 19)**.

References:

1. Jacobsen, I. D. (2023). The role of host and fungal factors in the commensal-to-pathogen transition of *Candida albicans*. *Curr Clin Microbiol Rep* 10: 55–65.
2. Fróis-Martins, R., Lagler, J., & LeibundGut-Landmann, S. (2024). *Candida albicans* Virulence Traits in Commensalism and Disease. *Current Clinical Microbiology Reports*, 11(4), 231-240.
3. Liang, S. H., Sircaik, S., Dainis, J., Kakade, P., Penumutchu, S., McDonough, L. D., ... & Bennett, R. J. (2024). The hyphal-specific toxin candidalysin promotes fungal gut commensalism. *Nature*, 627(8004), 620-627.
4. Du Toit, A. (2024). Hyphae promote *Candida albicans* fitness and commensalism in the gut. *Nature Reviews Microbiology*, 22(5), 258-258.
5. Pande, K., Chen, C., & Noble, S. M. (2013). Passage through the mammalian gut triggers a phenotypic switch that promotes *Candida albicans* commensalism. *Nature genetics*, 45(9), 1088-1091.

2. Reviewer #2

Minor comments (Remarks to the Author):

Q1. 1) *In the section « The physical interaction between Dal81 and Stp2 is required for pH alkalization and virulence of C. albicans », they provide information about the impact of mutating Dal81 residues involved in the interaction with Spt2 on C. albicans virulence. This comes before their section where they describe the role of Dal81 in commensalism and virulence using knock-outs. This is a bit awkward and I would recommend putting all data on commensalism and virulence in a single section.*

Answer: We appreciate the reviewer's constructive suggestion. We have reorganized the manuscript accordingly. All data relevant to *C. albicans* commensalism and virulence are now merged into a single consolidated section. Specifically, we have moved the data pertaining to the virulence evaluation of the *DAL81m21/dal81Δ* mutant (defective in Dal81-Stp2 interaction) to this unified section. Consequently, the original Figure 5J has been updated and relocated to the revised **Figure 8F**.

Q2. 2) *There is still room for improvement of the text. Here is a list of suggested changes, with in particular the need to use the past tense all along the result section rather than alternating between past and present. I have stopped making suggestion at some point.*

l.48 – A mutant lacking...

l.82-83 : to actively modify, either lowering or raising, the pH...

l.88 : that utilizes...

l.122 : pH alkalization regulation

l. 124 on both vaginal...

l.142 : 37°C

l.144 : defective in pH alkalization

l.149 : and a highly filamentous...

l.165 : defect...was also observed...

l.181 and elsewhere : alkalinize

l.195 : appears to be

l.233 : we asked

l.239 : formed normal

l.244 : was indistinguishable

l.248-250 : check sentence

l.252 : exhibited

l.262 : suggested

l. 269 : were

l.284 : as well as other Dal81- or Stp2-

l.344 : of Stp2

Answer: We thank the reviewer for these valuable suggestions, which will greatly enhance the clarity and readability of our work. We have carefully revised the manuscript, including thorough proofreading of the English language to minimize potential linguistic errors, especially those noted by the reviewer.

3. Reviewer #3

Minor comments (Remarks to the Author):

Q1. They state that the *dal81* mutant was generated in the SC5314 background, a prototrophic reference strain broadly used in the field, which would address concerns about strain-dependent phenotypes. However, the data supporting this claim are not shown in the manuscript. I strongly encourage the authors to include representative validation and phenotypic confirmation using the SC5314-derived strain to substantiate this key point.

Answer: We greatly appreciate the reviewer for this valuable and constructive suggestion. We have validated the newly generated *DAL81* deletion mutant in the SC5314 background using colony PCR and RT-qPCR, confirming the successful deletion of the *DAL81* gene (**Lines 137–144 on Page 5, Revised Fig.1C-D**). Additionally, we performed phenotypic assays with this SC5314-derived mutant (*dal81Δ/Δ* M1) and confirmed its defect in environmental alkalization on both medium 199 (**Revised Fig.1E**) and GM-GCP medium (**Review Fig. 1**). The alkalization phenotype of the *dal81Δ/Δ* M1 strain was consistent with that of the previously constructed SN250-derived *dal81Δ/Δ* mutant. These data have now been included in the revised manuscript to substantiate this key point.

Review Fig. 1. The *DAL81*-deficient (*dal81Δ/Δ* M1) mutant displayed alkalization defect on GM-GCP agar plate. Cells were spotted onto GM-GCP solid medium (initial pH 4.0) and incubated at 37 °C for 3 days.

Q2. That said, the assessment of macrophage killing relies exclusively on LDH release, an endpoint assay that offers limited temporal resolution. The use of viability dyes such as propidium iodide (PI) or DRAQ5 could have provided dynamic, real-time insights into host cell death, adding mechanistic depth to the macrophage interaction data. Moreover, there is no

direct readout addressing the susceptibility of *dal81Δ/Δ* to macrophage killing. One can only presume that, since this mutant causes less macrophage damage, it is more susceptible to immune-mediated clearance. Although not essential for publication, these points represent missed opportunities to enrich the study.

Answer: We appreciate the reviewer's insightful comments. In response, we performed propidium iodide (PI) staining on J774A.1 macrophages infected with *C. albicans* at 2 hours post-infection (hpi). We observed greater macrophage death in the wild-type (WT) group compared to the *dal81Δ/Δ* mutant group (**Review Fig. 2**). These results further support our LDH release data and demonstrate that the *dal81Δ/Δ* mutant had an attenuated ability to damage macrophages.

Regarding the reviewer's point about the susceptibility of *dal81Δ/Δ* to macrophage killing, we assessed this by quantifying viable fungal cells via CFU counting following macrophage infection. As shown in **Revised Figures 1K-M**, the *dal81Δ/Δ* mutant exhibited reduced survival across all three macrophage models tested, indicating that this mutant is indeed more sensitive to macrophage-mediated killing than the wild type. While we have not yet investigated the specific mechanisms underlying this increased susceptibility, we agree that this represents an important direction for our future study.

Review Fig. 2. Deletion of *DAL81* significantly reduced J774A.1 macrophage cell death. (A) J774A.1 macrophages were incubated with *C. albicans* at a multiplicity of infection (MOI) of 3. Dying cells were stained with propidium iodide (PI) (red) at a concentration of 200 ng/mL.

Images were acquired at 2 hpi, with representative images shown. Scale bars = 40 μm . Experiments were repeated three times. The percentage of PI-positive J774A.1 macrophages was calculated at each time point, with at least 100 cells assessed per sample. Data are presented as means \pm SD of three biologically independent samples. Statistical significance was assessed via unpaired two-tailed Student's *t* test.

Q3. *Similarly, while the updated images showing Stp2 nuclear localization are somewhat improved, the signal-to-noise ratio remains limited. Immunofluorescence may not be the optimal approach for confidently assessing nuclear translocation in this context. The authors support efficient Stp2 activation by showing proteolytic processing, and we can reasonably assume that this is followed by nuclear translocation; however, the evidence provided is suboptimal. For future work, I would recommend using fluorescent protein tags expressed from the native locus to provide more robust and interpretable localization data.*

Answer: We appreciate the reviewer's feedback regarding the limitations of our immunofluorescence-based assessment of Stp2 nuclear localization, as well as the valuable suggestion for future work. We are planning to incorporate these ideas in our follow-up studies to further validate and refine our understanding of Stp2 nuclear localization, and we thank the reviewer for this constructive input. In fact, one of our students is currently working to address these issues.

Q4. *To further improve clarity, the authors might briefly elaborate on whether the colonization defect is due to impaired survival, altered nutrient utilization, or impaired niche competition, issues that could guide future work.*

Answer: We sincerely thank the reviewer for these invaluable suggestions, which will greatly enhance our understanding of how the Dal81-Stp2 interaction influences the gastrointestinal colonization and pathogenicity of *C. albicans*. These ideas will certainly be included in our list of future research priorities.

Q5. *Finally, I recommend that the manuscript undergo careful proofreading to correct minor typographical errors and improve overall clarity and language flow.*

Answer: We are grateful to the reviewer for these constructive suggestions. We have implemented key edits in the revised manuscript, including careful proofreading to correct minor typographical errors.

Q6. *Overall, the manuscript has improved considerably and contributes meaningful insights into Candida albicans–host interactions. With the inclusion of the SC5314-based data as claimed, I would support the publication.*

Answer: We sincerely appreciate the reviewer’s positive feedback on our work. As noted in Q1, we have generated a new *DAL81* deletion mutant in the SC5314 background and comprehensively evaluated the role of Dal81 in pH alkalization of *C. albicans*. All these data have been incorporated in the revised manuscript (**Lines 137–147 on Page 5, Revised Fig.1C-E**).